## OPEN

# NANOG prion-like assembly mediates DNA bridging to facilitate chromatin reorganization and activation of pluripotency

Kyoung-Jae Choi [1,7], My Diem Quan[1,7], Chuangye Qi[2,7], Joo-Hyung Lee[2], Phoebe S. Tsoi[1], Mahla Zahabiyon[3], Aleksandar Bajic[3], Liya Hu [4], B. V. Venkataram Prasad [4], Shih-Chu Jeff Liao[5], Wenbo Li [2,6 ✉], Allan Chris M. Ferreon [1 ✉] and Josephine C. Ferreon[1 ✉]

Human NANOG expression resets stem cells to ground-state pluripotency. Here we identify the unique features of human NANOG that relate to its dose-sensitive function as a master transcription factor. NANOG is largely disordered, with a C-terminal prion-like domain that phase-transitions to gel-like condensates. Full-length NANOG readily forms higher-order oligomers at low nanomolar concentrations, orders of magnitude lower than typical amyloids. Using single-molecule Förster resonance energy transfer and fluorescence cross-correlation techniques, we show that NANOG oligomerization is essential for bridging DNA elements in vitro. Using chromatin immunoprecipitation sequencing and Hi-C 3.0 in cells, we validate that NANOG prion-like domain assembly is essential for specific DNA recognition and distant chromatin interactions. Our results provide a physical basis for the indispensable role of NANOG in shaping the pluripotent genome. NANOG's unique ability to form prion-like assemblies could provide a cooperative and concerted DNA bridging mechanism that is essential for chromatin reorganization and dose-sensitive activation of ground-state pluripotency.

NANOG gates access to unrestricted self-regeneration and germline development[1–3]. NANOG levels are tightly regulated in cells, with high levels correlating with reprogramming and self-renewal and low levels leading to spontaneous differentiation[4,5]. This striking dose sensitivity is linked to the *Nanog* gene's monoallelic to biallelic expression switch as the cell transitions towards ground-state pluripotency[6]. Furthermore, a landmark study[7] has shown that human stem cells in vitro could achieve ground-state pluripotency, similar to mouse embryonic stem cells (ESCs), by inducing additional NANOG expression.

The pluripotent genome is shaped by the master transcription factors (TFs) NANOG and OCT4[8]. Recent studies, including ours[9], have proposed that chromatin reorganization could occur through liquid–liquid phase separation (LLPS)[10–14]. We have demonstrated that KLF4 condensates facilitate NANOG expression, which explains the key role of KLF4 in induced cellular reprogramming[9]. LLPS is enhanced by the disordered prion-like domains (PrDs) present in many RNA- and DNA-binding proteins[15,16]. We thus characterized which unique features of NANOG can explain its key role in activating pluripotency.

### Results

**NANOG aggregation via its C-terminal PrD.** Early studies pinpointed NANOG's ability to dimerize[17]. Self-association is essential for NANOG pluripotency function, regardless of species origin and sequence conservation[17–20]. Previous studies reported that mouse NANOG dimerizes (~3 μM)[17,21] through its tryptophan repeat

(WR) domain (Fig. 1a). By means of pulldown experiments, human NANOG was also reported to self-associate via its WR domain[19]. Interestingly, the chicken NANOG C-terminal domain (CTD) was found to lack the conserved WR sequence (Extended Data Fig. 1a), but could still form higher-order helical structures (30–50 MDa at 30–70 μM) through critical tyrosine residues[18].

Human NANOG is aggregation-prone[22] and there is currently limited biophysical characterization of the protein and its subdomains. To delineate regions that contribute to aggregation, we determined each domain's relative solubility. The NANOG N-terminal domain (NTD) and DNA-binding domain (DBD) were highly soluble, whereas the WR-containing CTD constructs have limited solubility (Fig. 1b). The NANOG NTD is intrinsically disordered, exhibiting a random-coil circular dichroism (CD) spectral signature (Extended Data Fig. 1b) and narrow $^1$H peak dispersion in its two-dimensional (2D) $^{15}$N heteronuclear single quantum coherence (HSQC) NMR spectra (Fig. 1c), consistent with the NANOG computational disorder prediction (Extended Data Fig. 1c). The NANOG DBD has been well characterized in the literature as a folded domain with nanomolar to micromolar DNA-binding affinities[23]. In stark contrast to the NTD and DBD, the NANOG CTD is highly aggregation-prone. At 10 μM, signals in the 1D $^1$H NMR spectra were hardly detectable (Fig. 1d).

To probe the NANOG CTD structure, we utilized single-molecule Förster/fluorescence resonance energy transfer (smFRET), a technique that is ideal for aggregation-prone systems as experiments are performed in picomolar concentrations[24,25]. The NANOG CTD

[1]Department of Pharmacology and Chemical Biology, Baylor College of Medicine, Houston, TX, USA. [2]Department of Biochemistry and Molecular Biology, McGovern Medical School, University of Texas Health Science Center, Houston, TX, USA. [3]Department of Molecular and Human Genetics, Baylor College of Medicine and Jan and Dan Duncan Neurological Research Institute at Texas Children's Hospital, Houston, TX, USA. [4]Verna and Marrs McLean Department of Biochemistry and Molecular Biology, Baylor College of Medicine, Houston, TX, USA. [5]ISS, Inc., Champaign, IL, USA. [6]Graduate School of Biomedical Sciences, University of Texas MD Anderson Cancer Center and UTHealth, Houston, TX, USA. [7]These authors contributed equally: Kyoung-Jae Choi, My Diem Quan, Chuangye Qi. ✉e-mail: Wenbo.Li@uth.tmc.edu; Allan.Ferreon@bcm.edu; Josephine.Ferreon@bcm.edu

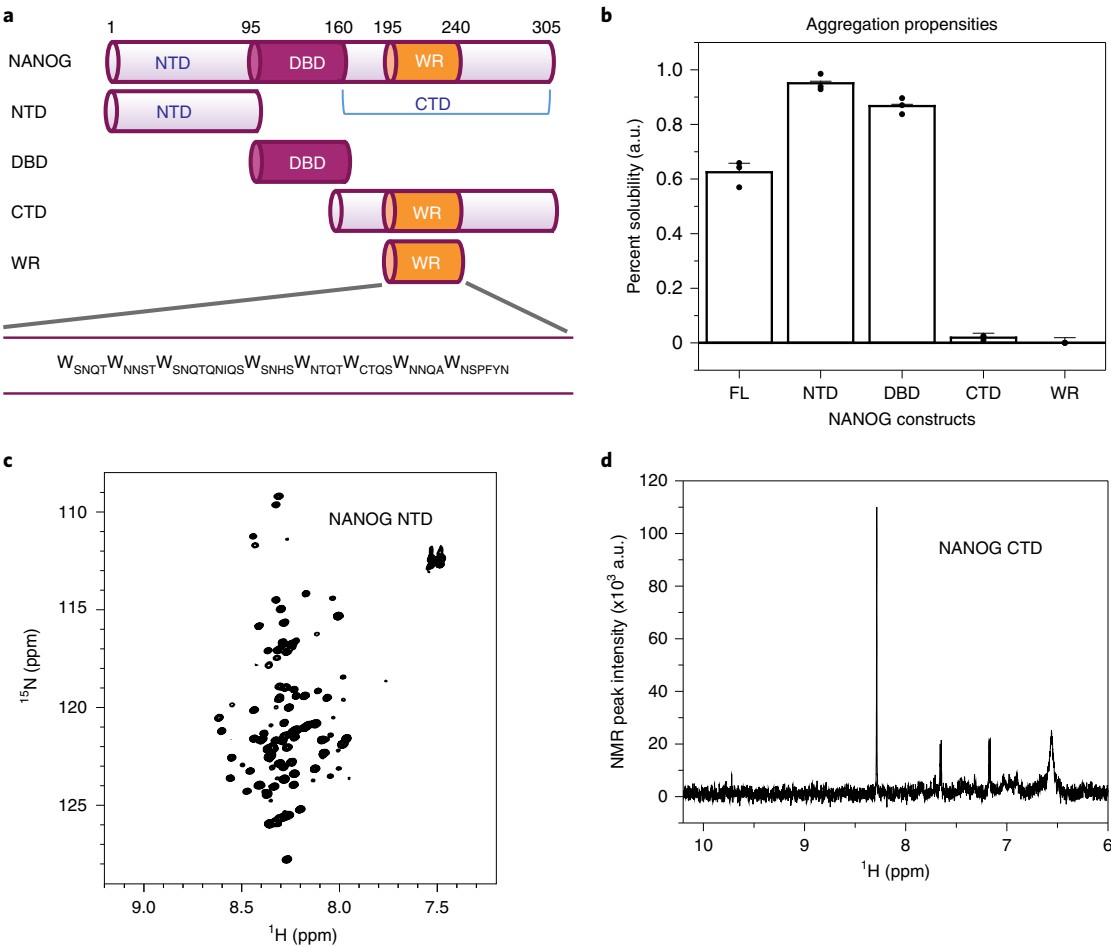

**Fig. 1 | The WR domain limits NANOG solubility. a**, NANOG constructs. FL, full-length; NTD, N-terminal domain; DBD, DNA-binding domain; CTD, C-terminal domain; WR, tryptophan repeats domain. **b**, Solubility of the NANOG constructs, determined from the fraction of protein remaining in the supernatant after centrifugation. Data are presented as mean ± s.d. ($n = 3$ independent replicates; 6 μM total concentration for all constructs, except for 100 μM NTD). **c**, 2D NMR $^{15}$N-HSQC spectra of the $^{15}$N NANOG NTD (500 μM), showing limited peak dispersion in the $^1$H dimension. **d**, 1D $^1$H NMR spectra of the NANOG CTD at 10 μM (256 scans; ~10 min), showing the absence of strong NMR backbone amide and tryptophan side-chain peaks. au: arbitrary unit.

was mutated to introduce Cys residues at positions 183 and 243, flanking the WR domain. Guanidine hydrochloride (GdnHCl) protein denaturation experiments were performed using the NANOG CTD doubly labelled with Alexa Fluor 488 and Alexa Fluor 594 (AF488/AF594). We observed shifts in FRET efficiencies towards lower values, as expected for denaturant-induced protein expansion (Fig. 2a). The observed FRET changes followed a non-cooperative transition (Fig. 2b), indicating that the CTD did not have a persistent structure or lacked a stable compact hydrophobic core[25,26]. However, we acquired CD spectra at a protein concentration of 2 μM and observed β-sheet secondary structure formation (Fig. 2c), indicating that the CTD undergoes a disorder-to-order transition. We speculated that this structural transition is linked to amyloid formation because the WR domain sequence is strikingly similar to amyloid prions or prion-like domains, which are rich in polar residues (Ser, Asn and Gln) and hydrophobic residues[15,27,28]. Indeed, thioflavin T (ThT) fluorescence assays demonstrated rapid protein aggregation with concentration-dependent kinetics (Fig. 2d). At low micromolar concentrations and in the absence of crowding agents, NANOG CTD easily phase-transitions to gel-like condensates (Fig. 2e) that are ThT-positive and migrate as high-molecular-weight ($M_w$) complexes in semi-denaturing detergent agarose gel electrophoresis (SDD-AGE; Fig. 2f). Scanning electron micrographs

of NANOG CTD gel revealed networks of fibril-like structures (Fig. 2g). To investigate whether Trp residues play a major role in aggregation, we mutated three (W468A) or four (W1357A) alternating Trp residues in the WR repeat sequence to Ala. This resulted in a reduction of β-sheet structures to more random coil-like structures (Fig. 2h). Solubility issues with the WR domain or peptides prevented experimental high-resolution structural studies. Consequently, we performed computational modelling of the two WR repeat sequences most homologous to published X-ray structures of peptide prions: yeast Sup35[28] and human prion protein (Fig. 2i). Both WR structures showed steric zipping of β-strands but were modelled in different orientations, suggesting that the WR domain does not adopt a unique structure. Heterogeneous orientations of WR assembly may allow the spatial flexibility for NANOG domains to interact with DNA and other proteins.

**NANOG oligomerizes in vitro and in cells.** We next investigated whether oligomerization translates to the full-length protein. We tested whether endogenous NANOG could spontaneously assemble in H9 human ESCs as well as nucleofected NANOG in HEK 293T mammalian cells. In the presence of the chemical crosslinker disuccinimidyl sulfoxide (DSSO), NANOG readily crosslinked to form dimers and other high-$M_w$ complexes in both cell types (Fig. 3a,b).

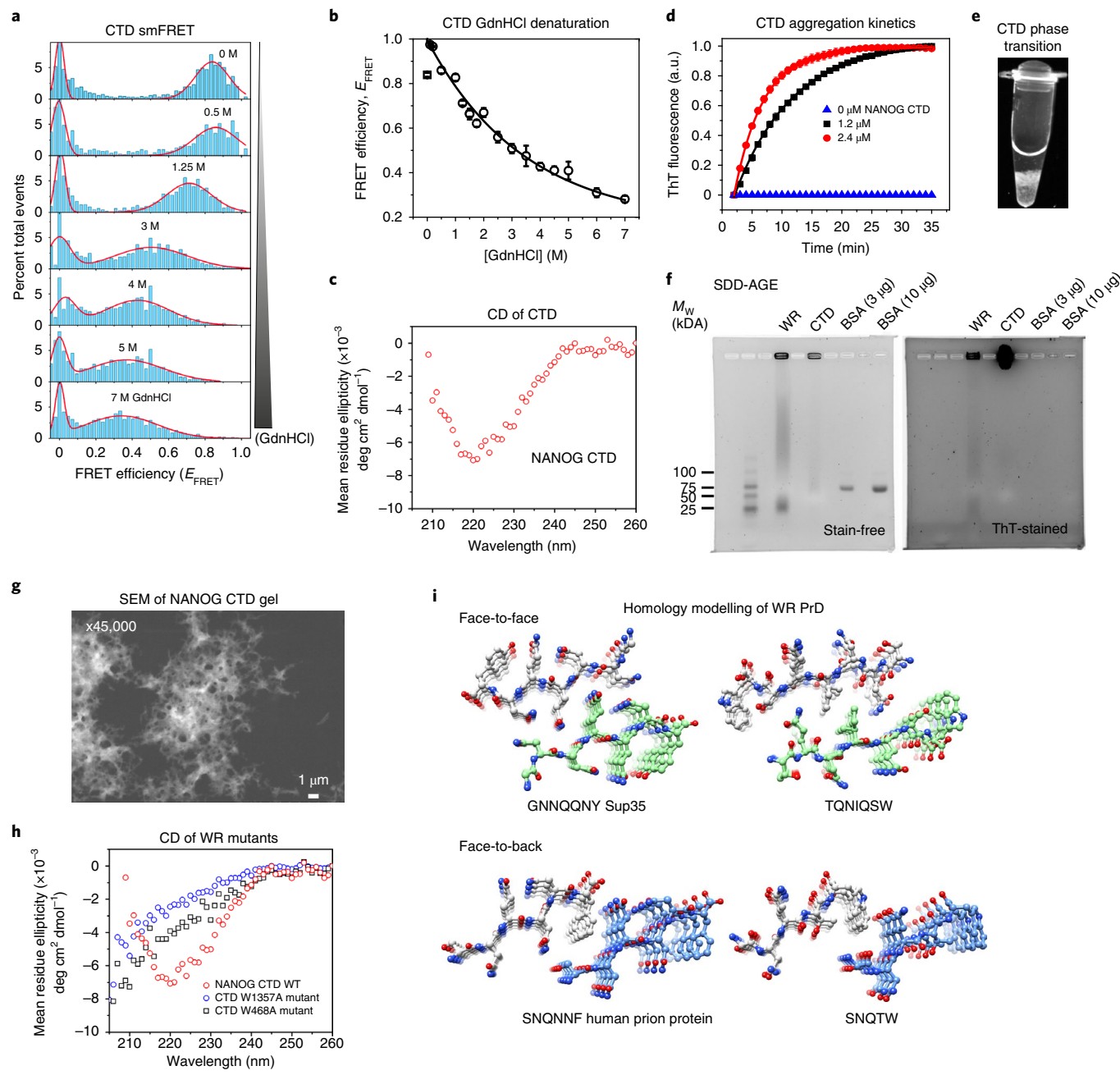

**Fig. 2 | The NANOG CTD displays prion-like behaviour. a**, smFRET histograms showing the relative number of events versus FRET efficiency ($E_{FRET}$) of the NANOG CTD (conjugated with the AF488-AF594 FRET pair) in increasing GdnHCl concentrations (top to bottom). **b**, $E_{FRET}$ as a function of GdnHCl concentration. Symbols and error bars represent nonlinear least-squares-fitted $E_{FRET}$ and fitting error values, respectively (Methods). **c**, CD spectra of 2 μM NANOG CTD. **d**, CTD aggregation kinetics monitored by ThT fluorescence. Graphs fitted to an exponential function $y = Ae^{Rx}$ with time constants ($1/R$) of 4.8 ± 0.2 min (mean ± s.d.; $n = 3$ independent replicates) for a concentration of 1.2 μM and 9.5 ± 0.1 min for 2.4 μM ($n = 3$ independent replicates). **e**, CTD can form gel-like condensates at a low concentration of 5 μM. Similar results were observed for two independent replicates. **f**, SDD-AGE gels (stain-free and ThT-stained) show WR and CTD as high-$M_w$ ThT-positive complexes using BSA as negative control. **g**, Scanning electron micrographs of NANOG CTD gel show porous, fibril-like networks. Similar results were observed for two independent experiments. **h**, Far-UV CD spectra of CTD alternating tryptophan mutants (W1357A and W468A), revealing a reduction in β-sheet propensity relative to WT. The data shown represent two independent experiments. **i**, Homology modelling of NANOG WR repeats, similar to known prion proteins.

To check the role of the WR domain in oligomerization, we generated WR mutants (ΔWR deletion, and the W8A mutant where all eight Trp residues were mutated to Ala) and transfected them into HEK 293T cells. Consistently, we observed that wild-type (WT) NANOG easily crosslinked to form higher-order species, whereas the WR mutants could not.

Because NANOG is a known hub protein that interacts with many cellular proteins[29], oligomerization in cells may not solely be due to NANOG self-assembly. We thus characterized the oligomerization behaviour of purified full-length NANOG WT and W8A mutant for side-by-side evaluation. We prepared NANOG constructs with GB1- and MBP-fused tags to increase their solubility

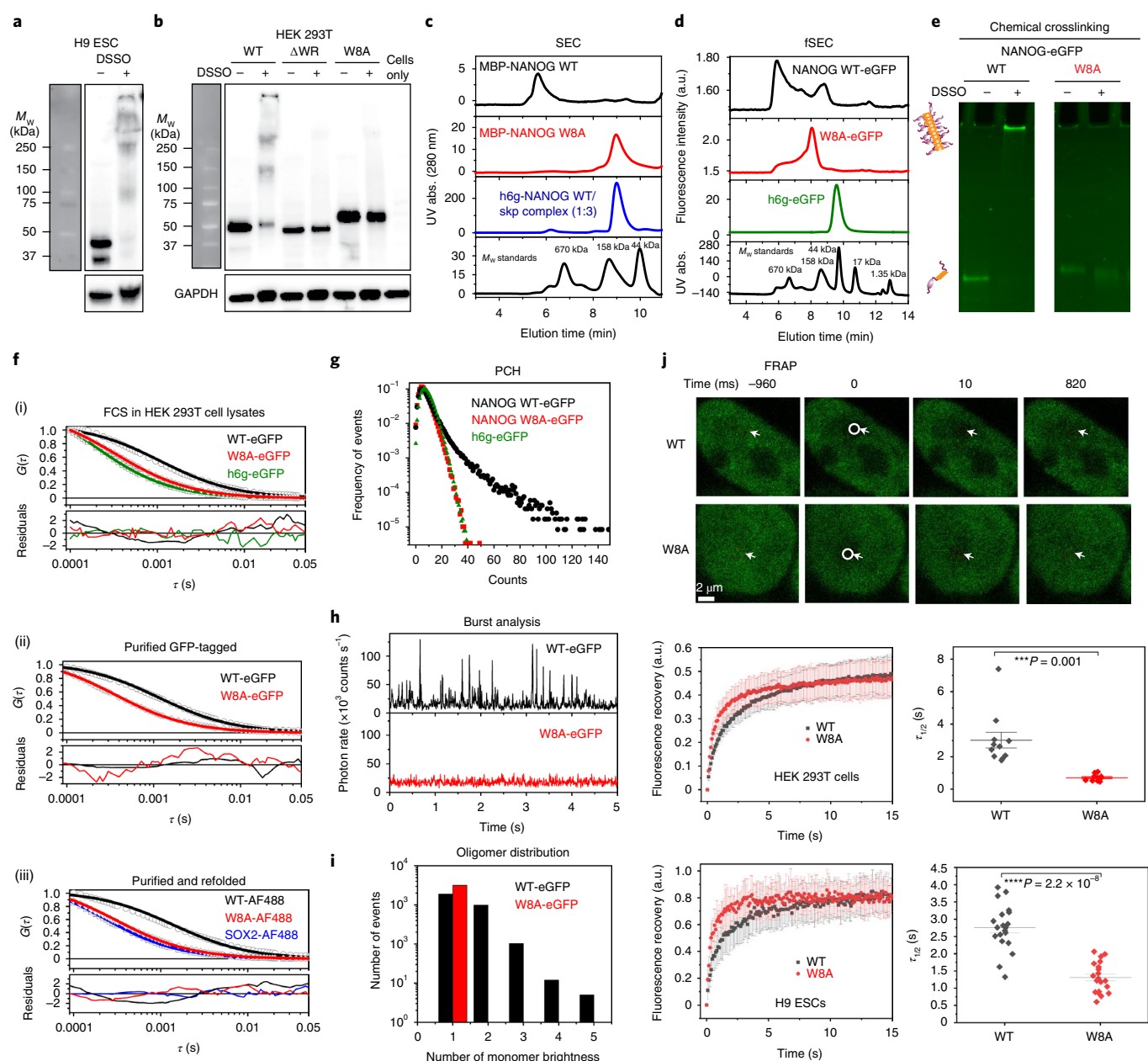

**Fig. 3 | NANOG readily oligomerizes at low nanomolar concentrations. a,b,** Chemical crosslinking of endogenous NANOG in H9 ESCs (**a**) and NANOG variants (WT, ΔWR and W8A) expressed in HEK 293T cells (**b**) (two biological replicates each). **c,** UV-detected SEC of MBP-fused NANOG WT (~300 nM), W8A (~1.5 μM), and h6g-NANOG:Skp complex with $M_w$ calibration standards. Similar results were observed for two independent experiments. **d,** Fluorescence-detected SEC of GFP-tagged NANOG WT/W8A (~10 nM) and h6g-eGFP. NANOG WT elutes in the void volume corresponding to high-$M_w$ aggregates (Extended Data Fig. 3a). Similar results were observed in two independent experiments. **e,** DSSO chemical crosslinking of purified GFP-tagged NANOG WT/W8A (2.5 nM). Similar results were observed in four independent replicates. **f,** Autocorrelation FCS curves of (i) h6g-eGFP and GFP-tagged NANOG WT and W8A constructs (~10 nM) in HEK 293T mammalian cell lysates; (ii) purified GFP-tagged NANOG WT and W8A constructs (~5 nM); (iii) refolded AF488-conjugated NANOG WT/W8A and SOX2 proteins (~10 nM). Similar results were observed in two to six independent experiments. Additional information is provided in Supplementary Table 1. HEK 293T cell lysates were treated with benzonase to prevent DNA–protein interactions that could affect the diffusion properties of NANOG. **g,** Photon-counting histograms (PCH) of NANOG WT (black), W8A (red) and h6g-eGFP control (green). Similar results were observed in two independent replicates. **h,** Distribution of photon bursts for NANOG WT and the W8A mutant. Similar results were observed in two independent replicates. **i,** Binned burst histograms (multiples of the average photon rates or counts per second) of WT and the W8A mutant. **j,** Top: representative FRAP images of GFP-fused NANOG WT and W8A mutant in HEK 293T cells. Middle and bottom: FRAP curves (left; mean ± s.d.; $n = 20$ cells across two biological replicates) and corresponding fluorescence recovery half-times (right; $\tau_{1/2}$) of overexpressed GFP-tagged NANOG WT (black) and W8A mutant (red) in HEK 293T cells (middle row) and H9 ESCs (bottom row). $P$ values were based on two-sided paired Student's $t$-tests. ***$P \leq 0.001$, ****$P \leq 0.0001$.

(Extended Data Fig. 2). GB1 fusion resulted in the accumulation of WT protein in inclusion bodies upon *Escherichia coli* expression. However, consistent with previous observations[22], GB1-fusion

NANOG WT became soluble when co-expressed with the Skp chaperone. Using size exclusion chromatography (SEC), both Skp- and GB1-fused NANOG WT proteins co-eluted as a 3:1 complex

(Fig. 3c and Extended Data Fig. 2a). It is most likely that the trimeric Skp chaperone provides a hydrophobic cage[30] that wraps around NANOG WT and interacts with exposed Trp residues. This is consistent with the co-elution of WT but not the W8A mutant (Extended Data Fig. 2a). With MBP fusions, the proteins were soluble. However, strong detergents were necessary for the purification process. To verify that purified proteins were still functional for DNA binding, fluorescence-based electrophoretic mobility shift assays (fEMSAs) were performed (Extended Data Fig. 2d,e). Despite binding DNA effectively, MBP-NANOG WT eluted as a high-$M_w$ aggregate by SEC (Fig. 3c). By contrast, the W8A mutant migrated as a monomeric peak based on its estimated $M_w$ (Fig. 3c and Extended Data Fig. 3a). We also purified green fluorescent protein (GFP)-tagged NANOG from HEK 293T cells, speculating that low expression makes it less prone to aggregation (Extended Data Fig. 2c). Interestingly, using fluorescence-detected SEC and 10 nM injection concentrations, GFP-tagged WT eluted in the void volume (high-$M_w$ complex, >2 MDa), whereas W8A migrated mostly as a monomeric peak (Fig. 3d and Extended Data Fig. 3). DSSO crosslinking of purified proteins confirmed that the WT assembled readily and immobilized in SDS–PAGE wells, whereas W8A failed to crosslink inter-molecularly (Fig. 3e).

We then characterized the NANOG oligomerization states at low nanomolar concentrations using fluorescence fluctuation spectroscopy (FFS). By quantifying the fluorescence intensities of a few molecules at a time, FFS data can be analysed using a number of different strategies: fluorescence correlation spectroscopy (FCS)[31] to obtain the molecule's diffusion coefficient, photon-counting histograms (PCHs)[32,33] to probe sample heterogeneity, and burst analysis[34] to map a molecule's 'brightness' distribution, which gives insight into the oligomer sizes (for example, a dimer will have twice the brightness of a monomer).

By means of FCS, we investigated the diffusion behaviour of NANOG using different NANOG preparations and solution conditions: (1) GFP-tagged in mammalian cell lysates; (2) purified GFP-tagged in vitro, with detergent and high-salt buffer; (3) purified AF488-conjugated NANOG in vitro. To estimate the molecular sizes, we used empirical equations relating the hydrodynamic radii and the number of residues for folded and denatured proteins[35]. Both the positive control h6g-eGFP (eGFP, enhanced GFP) and GFP-tagged W8A mutant had diffusion coefficients consistent with their monomeric sizes, falling within the molecular size boundary calculations for folded and denatured proteins (Fig. 3f(i),(ii), Extended Data Fig. 3b and Supplementary Table 1). Meanwhile, GFP-tagged NANOG WT (in vitro and in mammalian cell lysates) diffused three times more slowly than the W8A mutant, and its hydrodynamic radius ($R_h$) fell significantly outside the monomeric size range, even with the assumption that NANOG is completely disordered (Extended Data Fig. 3b and Supplementary Table 1). Consistently, data with AF488-conjugated, refolded

proteins (Fig. 3f, Extended Data Fig. 3b and Supplementary Table 1) show that SOX2-AF488 (similar-size disordered protein) and the mutant W8A-AF488 were in good agreement with monomeric sizes, whereas the WT-AF488 behaved more like a large oligomer. Back-calculation from the diffusion coefficients to the number of residues using empirical equations estimated that the WT could be 8–100 times larger than the W8A mutant (Supplementary Table 1).

Using independent PCH analyses, h6g-eGFP and GFP-tagged W8A show photon count distributions that could be approximated by a Poisson distribution for a particle with uniform molecular brightness (Fig. 3g and Extended Data Fig. 3c). By contrast, WT revealed a 'long tail' distribution, indicative of large aggregates[33]. Consequently, the PCH curve could only be fitted segmentally, with each fit characterized by a different molecular brightness or oligomer size (Extended Data Fig. 3c). Alternatively, using burst analysis or direct observation of raw intensity fluctuations, we observed that the W8A mutant had uniform fluorescence intensity fluctuations, whereas the WT had 'bursts' or large intensity deviations (~10–12 times the average fluctuations; Fig. 3h). To estimate the sizes of the oligomers present, we assumed that the average intensity fluctuations represented the monomeric species, and the data were binned based on average counts per second. Consistent with other data, the W8A mutant showed only a monomeric distribution (Fig. 3i). The WT, however, showed several oligomeric size distributions and no distinct oligomeric size. We observed similar trends in FFS experiments with mCherry-tagged NANOG in-cell lysates. More oligomers were observed in the presence of the WR PrD (NANOG-mCherry) than in its absence (NANOG ΔWR-mCherry; Extended Data Fig. 3d). Thus, NANOG self-assembled readily at low nanomolar concentrations. To further validate that NANOG cellular concentration permits self-assembly, we estimated the endogenous NANOG cellular concentration in H9 ESCs using GFP and purified NANOG calibrations and western blot imaging (Extended Data Fig. 4). The concentrations of both endogenous NANOG in H9 ESCs and GFP-tagged NANOG expressed in HEK 293T cells were found to be ~80–750 nM, well above the ~5 nM concentration used in our FFS experiments. We thus expect NANOG to self-assemble into higher-order oligomers once expressed in cells. To confirm this, we performed fluorescence recovery after photobleaching (FRAP) experiments in HEK 293T cells and H9 ESCs with overexpressed WT or W8A (Fig. 3j). The WT NANOG demonstrates significantly slower fluorescence recovery (or longer recovery lifetimes) than the W8A mutant.

**NANOG assembly permits DNA bridging.** We speculated that the NANOG assembly properties have relevance to its function as a master TF. To investigate how NANOG oligomerization affects DNA binding, we performed fEMSA with untagged WT and W8A mutants (Fig. 4a), as well as GFP-tagged WT and W8A (Extended Data Fig. 5a–c). As a control, we checked that the

---

**Fig. 4 | NANOG assembly mediates intermolecular DNA bridging in vitro and pluripotency function in H9 ESCs. a**, Representative fEMSA results for NANOG WT and W8A mutant with 5 nM *Gata6*-AF647. Similar results were obtained with three independent replicates. **b**, NANOG WT/W8A oligomer population, that is, band intensities in fEMSA wells. Similar results were obtained in three independent replicates. **c**, Fractions of unbound DNA (shown as white rectangles in **a** with 125 and 250 nM NANOG WT/W8A (mean ± s.d.; $n = 3$ independent replicates). $P$ values were based on two-sided paired Student's $t$-tests. **$P \leq 0.05$, ***$P \leq 0.01$. **d**, smFRET of ~100 pM each of *Gata6*-AF488 and *Gata6*-AF647 intermolecular diffusion in (i) the absence or (ii,iii) the presence of 250 nM NANOG W8A or WT NANOG. The peak at $E_{FRET} \approx 0$ corresponds to AF488-conjugated unbound/bound DNA. Similar results were observed in two independent experiments. **e**, Representative cross-correlation FCCS curves of *Gata6*-AF488 and *Gata6*-AF647 with various concentrations of WT ((i) 0 nM; (iii) 63 nM; (iv) 875 nM) and W8A mutant ((ii) 875 nM). Similar results were observed in three independent experiments. Results for other measured concentrations are reported in Extended Data Fig. 7 and the Supplementary Information. **f**, Number of cross-correlated particles ($N_{ad}$) per volume (µm³; left $y$ axis) and diffusion coefficients of WT/W8A–DNA complexes (right $y$ axis) in relation to protein concentrations. $N_{ad}$ and diffusion coefficients were derived and calculated from FCCS fits (Methods; mean ± s.d.; $n = 3$ independent experiments). **g,h**, Fluorescence (**g**) and bright-field (**h**) (cells stained with crystal violet) microscopy images of H9 ESCs overexpressing GFP-fused NANOG WT (left) or W8A (right), showing the maintenance of characteristic round ESC colonies for the WT versus differentiation and dispersion of ESC colonies for the W8A mutant. The data represent two biologically independent replicates.

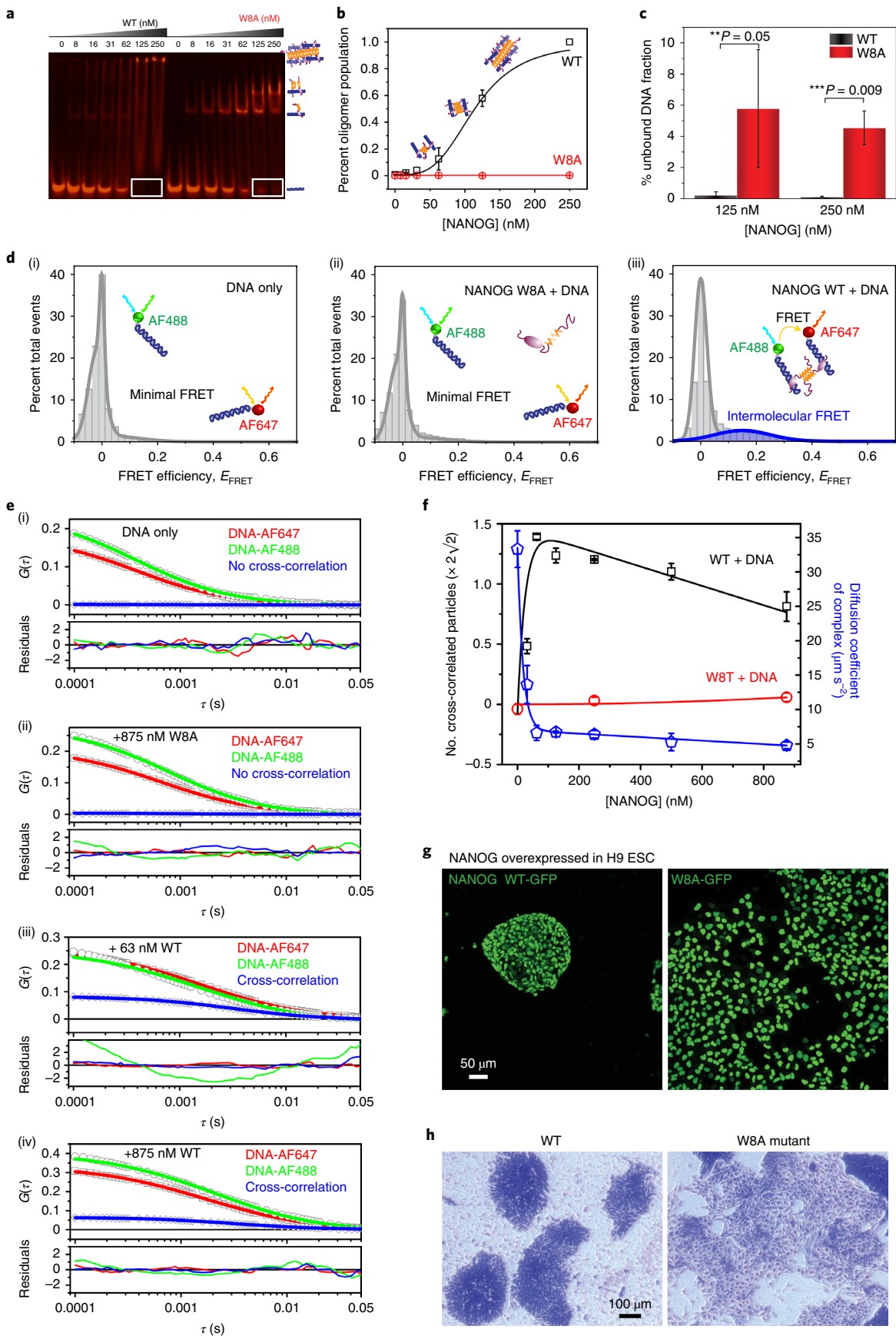

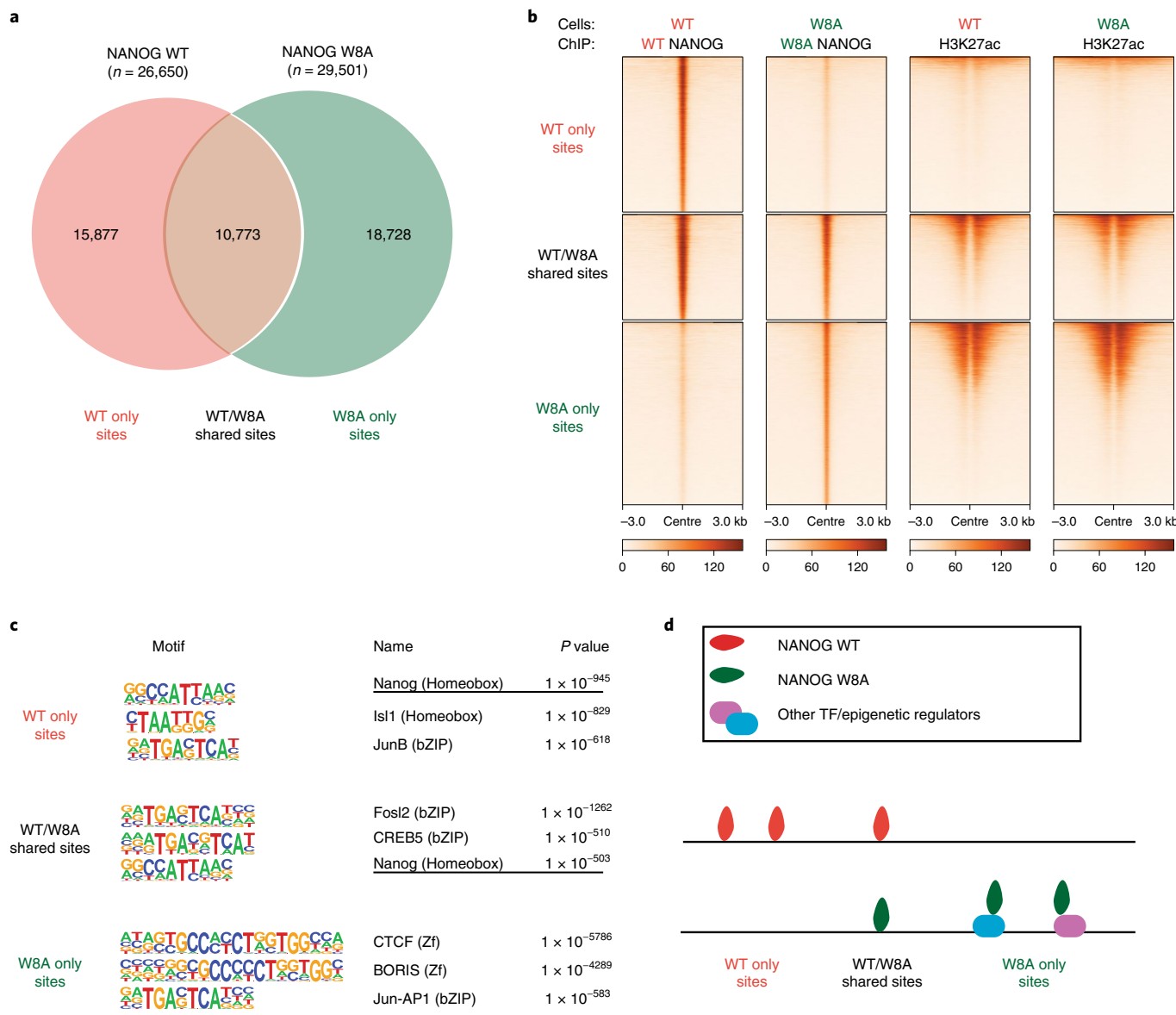

**Fig. 5 | NANOG PrD mutations alter DNA recognition in cells. a**, Venn diagram summarizing the numbers of NANOG ChIP-seq peaks observed for WT or W8A expressed in HEK 293T cells. The binding sites are classified into three groups: WT only sites, WT/W8A shared sites and W8A only sites. **b**, Heatmaps of the normalized ChIP-seq reads of NANOG and H3K27ac on three groups (top to bottom): NANOG WT only, WT/W8A shared and W8A only sites, respectively. **c**, Top three non-redundant DNA motifs identified by HOMER from NANOG ChIP-seq data for the three classified groups. *P* values were generated by hypergeometric tests in HOMER. **d**, Cartoon diagram describing the possible modes (direct or indirect) of DNA recognition for the three groups. The shown ChIP-seq data were derived using two biologically independent replicates.

NANOG CTD or oligomerization domain does not specifically interact with DNA (Extended Data Fig. 5d). As expected, both the NANOG WT and W8A bound DNA effectively due to the presence of DBDs. However, we observed dramatic differences in the band and migration patterns. There were two distinct bands with W8A, representing species singly and doubly bound to the *Gata6* DNA (40 bp with two cognate sites), whereas WT had one distinct band (singly bound) and a smeared distribution consistent with variably sized oligomers. At higher NANOG concentrations, proteins were immobilized in the fEMSA gel wells, consistent with the presence of high-$M_w$ DNA:NANOG complexes. Quantification of bands as a function of NANOG concentration revealed a sigmoidal curve consistent with a cooperative assembly process present in the WT but not in W8A (Fig. 4b). More importantly, we observed a higher amount of DNA bound to the WT than W8A when larger NANOG

oligomers were present (Fig. 4a, white rectangles; quantification in Fig. 4c). Consistently, fEMSA experiments with GFP-tagged NANOG with 40-bp *Gata6* and other known DNA targets (257-bp *pγSat* satellite DNA and 404-bp *Nanog* promoter) confirmed that the WT bound DNA more cooperatively and tightly than the W8A mutant (Extended Data Fig. 5a–c). The presence of DNA:NANOG oligomers suggests that NANOG may bridge two isolated DNAs through its CTD PrD.

To test our DNA bridging hypothesis in vitro, we performed smFRET diffusion experiments. Intermolecular diffusion smFRET is challenging to carry out because of low chance encounters at dilute picomolar concentrations. *Gata6* DNA was independently labelled with AF488 or AF647. With DNA alone or in the presence of NANOG W8A, we hardly observed FRET events (Fig. 4d and Extended Data Fig. 6). However, in the presence of 250 nM NANOG

WT, we observed a FRET efficiency of ~0.15, which proved that WT was able to bring separate DNAs to within ~70-Å proximity (Fig. 4d(iii)). To further validate NANOG-dependent DNA bridging, we used fluorescence cross-correlation spectroscopy (FCCS). Binding of individual DNAs (differentially labelled with AF488 or AF647) to form a complex would result in correlated fluorescence signals. With the DNAs alone or with W8A mutant (Fig. 4e(i),(ii)), no cross-correlation (blue line) was observed. However, in the presence of WT NANOG, we observed significant cross-correlation (Fig. 4e(iii),(iv)). FCCS experiments with various NANOG concentrations confirmed cross-correlation with as little as ~30 nM WT NANOG (Extended Data Fig. 7, Fig. 4f and Supplementary Information). The increase in cross-correlated particles coincided with a decrease in the molecular complex diffusion coefficient (Fig. 4f, blue line). The decrease in cross-correlated particles at high NANOG concentrations may reflect competition for DNA with excess NANOG. Similar results were observed with different DNA sequences (*GATA6-AF488* and *OCT4-AF647*; Extended Data Fig. 7c). Thus, NANOG assembly is essential for cooperative recognition and intermolecular DNA bridging. To test whether NANOG assembly is critical for the NANOG pluripotency function, we generated stable H9 cell lines transduced with lentiviral constructs carrying doxycycline (DOX)-inducible GFP-fused NANOG WT or W8A. After 3–5 days of DOX induction, we observed differentiation of ESC colonies with NANOG W8A overexpression, but not with WT (Fig. 4g,h and Extended Data Fig. 8). These results confirmed the essential role of the NANOG oligomerization domain in maintaining pluripotency.

**NANOG oligomerization facilitates specific DNA recognition.** To investigate the role of NANOG assembly in DNA recognition in cells, we performed chromatin immunoprecipitation sequencing (ChIP-seq) in HEK 293T cells (without endogenous NANOG expression) expressing exogenous GFP-tagged NANOG WT or W8A mutant. This showed that the NANOG WT and W8A bind distinct and shared groups of chromatin sites, classified as WT only sites ($n = 15,877$), W8A only sites ($n = 18,728$) and WT/W8A shared sites ($n = 10,773$) (Fig. 5a,b). Regardless of groups, there were no obvious changes of the active histone mark H3K27ac, supporting that exogenous expression of WT or W8A NANOG did not alter the overall cell epigenome (Fig. 5b). Further inspection indicated that W8A only sites displayed higher H3K27ac signals than WT only,

suggesting that mutant NANOG tends to associate with highly active chromatin sites (Fig. 5b). Analysis of the TF DNA motifs enriched in the three groups identified the NANOG cognate motif and various motifs of the bZIP family (three displayed in Fig. 5c; a full list is provided in Extended Data Fig. 9). The NANOG motif can be identified for WT only and WT/W8A shared sites, consistent with a direct DNA interaction of WT NANOG (Fig. 5c,d). By contrast, W8A only sites did not contain obvious NANOG motif enrichment, but instead other motifs (for example, CTCF and BORIS, Fig. 5c,d). These data suggest that the W8A mutant may be recruited to these sites via other TFs or epigenetic regulators, but not through its own direct DNA binding ability (Fig. 5d). The results were unexpected, because the W8A mutations are in the WR oligomerization domain, not the DNA-binding domain. Interestingly, the in-cell data are consistent with in vitro biophysical data (Fig. 4a–c and Extended Data Fig. 5), demonstrating that the oligomerization domain could facilitate cooperative DNA recognition.

**NANOG PrD assembly enables distant DNA bridging in cells.** To directly investigate whether NANOG bridges DNA contacts in a cellular setting, we used Hi-C 3.0[36,37]. This method is similar to in situ Hi-C[38], but employs two crosslinkers (formaldehyde and disuccinimidyl glutarate (DSG)) and two restriction enzymes to improve the data resolution[36,37,39]. We generated two Hi-C 3.0 biological replicates from HEK 293T cells expressing either WT or W8A NANOG and sequenced them to ~500 million reads (Supplementary Table 2). Our Hi-C 3.0 data showed high concordance between replicates, as shown by stratum adjusted correlation coefficient (SCC) analysis[40] (Extended Data Fig. 10a). We thus pooled the two replicates for the following analysis. By calculating the contact probability ($P$) in relation to the genomic distance ($s$) ($P(s)$ curve in Fig. 6a), we found no global difference in the DNA contact probability across genomic distances, indicating that the cells' chromatin was not massively reorganized by the expression of WT or W8A (Fig. 6a). As an example, the Hi-C 3.0 contact heatmap for a randomly selected ~15-Mb genomic region showed similar chromatin organization patterns in WT and W8A cells (Fig. 6b). The magnified view of a ~1.5-Mb region confirmed largely unaltered contact maps and structures of topologically associating domains (TADs; Fig. 6c).

Mammalian chromatin is organized into A/B compartments, TADs and chromatin loops[41–43]. A/B compartments were unaltered by WT or W8A NANOG, as shown by the principal component

**Fig. 6 | NANOG PrD assembly mediates distant DNA–DNA contacts by Hi-C 3.0 analyses. a**, $P(s)$ curves showing the probability of contact in relation to genomic distance in HEK 293T cells expressing either NANOG WT (red) or W8A (green). **b**, Contact heatmaps showing normalized interaction frequencies (20-kb bin) in one example region (chr7: 15–30 Mb). **c**, Magnified view of a region (chr7: 26–27.5 Mb) that hosts TAD structures. **d**, Diagram explaining the strategies in calculating pairwise DNA contacts between NANOG binding sites (Methods; adopted from a previous method, paired-end spatial chromatin analysis (PE-SCAn)). Black ovals indicate NANOG, and *a* and *b* indicate two distant genomic bins (25 kb) harbouring NANOG binding sites. Sliding windows of 25 kb were used to scan each site of *a* or *b* for 250 kb, and the interactions between each sliding window bin next to *a* or *b* were calculated as the background interactions surrounding specific *a*–*b* interactions. For any NANOG binding sites ($a_1$–$a_n$ versus $b_1$–$b_n$), the aggregated interactions between each pair of sites and nearby background are shown below. **e**, DNA contact strength under two conditions (WT or W8A) between WT only sites ($n = 1,979$), shared sites ($n = 1,152$) or W8A only sites ($n = 2,577$) (high-density NANOG binding cluster with ≥2 peaks in 25 kb). (i) Data plots based on the PE-SCAn method. (ii) Box plots showing quantitative counts of the central peaks shown in (i). The boxplot centre lines represent medians, the box limits indicate the 25th and 75th percentiles, and the whiskers extend 1.5 times the interquartile range (IQR) from the 25th and 75th percentiles. *P* values are based on two-sided paired Student's *t*-tests. (iii) Cartoon diagrams based on DNA contacts. **f**, Example of Hi-C 3.0 DNA contacts between NANOG binding sites. (i)–(vi) Representative genomic loci showing DNA–DNA contacts formed between NANOG binding sites in Hi-C 3.0 (arc plots). From the top to the bottom, the data shown are (i,ii) NANOG ChIP-seq (either WT or W8A), (iii,iv) DNA interaction arc plots (either WT in purple or W8A in blue), (v) log$_2$ fold changes of DNA interactions (with green–red coloured scales) and (vi) genomic annotations and coordinates. The purple and blue arcs in panels (iii) and (iv) indicate chromatin interactions between two NANOG sites in cells expressing either WT (purple) or W8A (blue), respectively. The arc thickness represents the interaction strength in Hi-C 3.0. For the fold changes shown in (v), they were plotted on the colour scale shown to the right, which were calculated using Hi-C 3.0 normalized contacts between the sites; green and red indicate their reduction or increase, respectively. Most of these contacts were weaker in the W8A condition than in the WT condition. The shown Hi-C 3.0 data were derived using two biologically independent replicates. **g**, Model of how NANOG can reshape the pluripotent genome. Through prion-like assembly, NANOG can initiate or stabilize intragenomic (promoter–enhancer) contacts, as well as connect distant intergenomic loci to form superenhancer clusters with other TFs and coactivators (green/yellow).

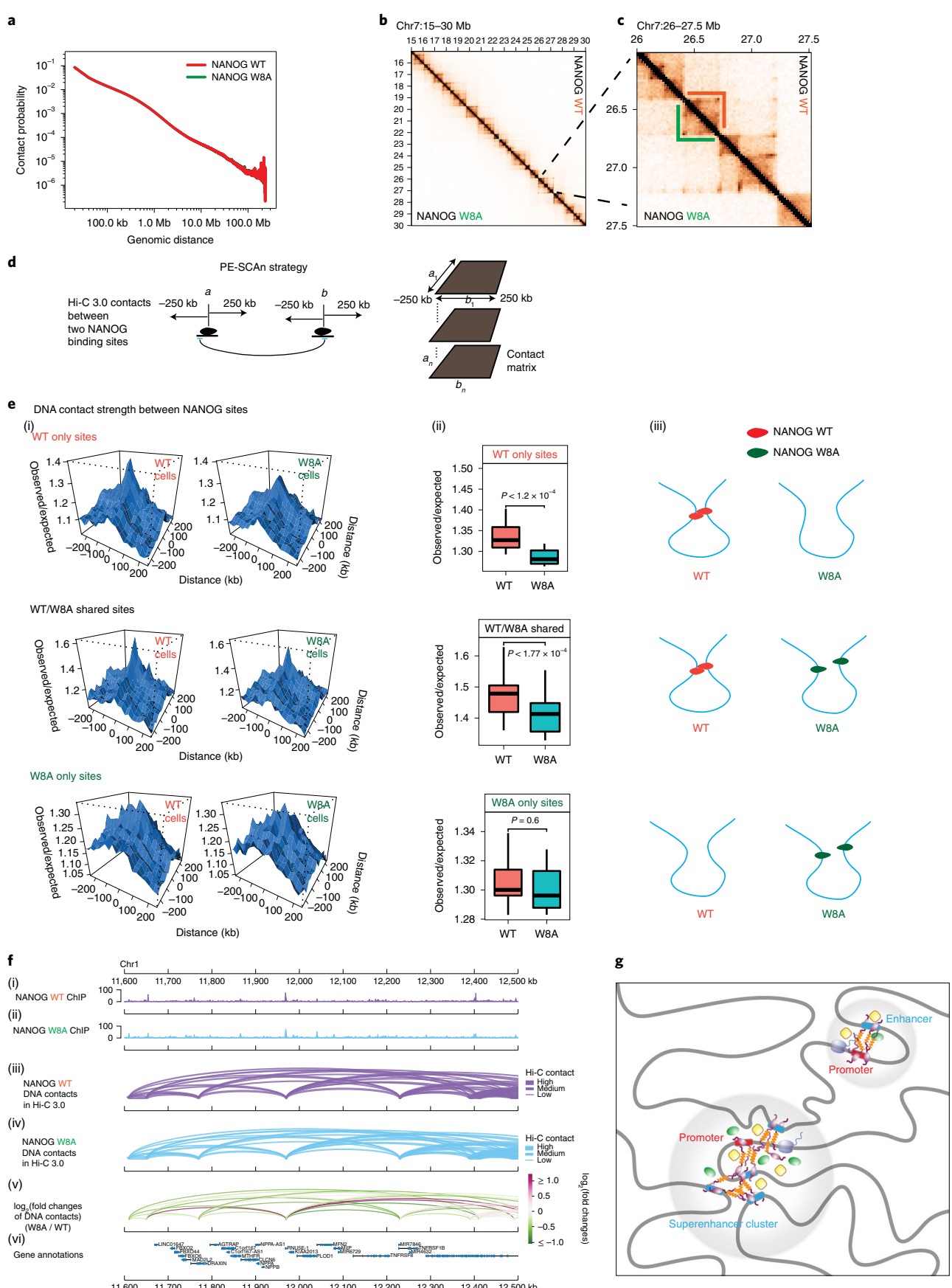

analysis (PCA) E1 values across the genome (Extended Data Fig. 10b,c). Saddle plots that describe pairwise intercompartmental interactions also showed no obvious changes (Extended Data Fig. 10d). TADs are self-interacting genomic architectures[44]. For the 3,413 TADs identified from our Hi-C 3.0 data[45], the aggregated TADs strength was largely identical in both WT and W8A conditions (Extended Data Fig. 10e). Thus, WT or W8A did not significantly affect the A/B compartments or TADs.

We next tested the roles of NANOG in bridging DNA–DNA contacts that occur between NANOG binding sites, and how WT and W8A may differ in such functions. We adopted the paired-end spatial chromatin analysis (PE-SCAn) method[8] that was designed to measure DNA–DNA contacts formed between TF binding sites (Methods and Fig. 6d). For the three groups of sites (WT only, shared or W8A only), we calculated chromatin contacts formed between two NANOG binding sites in cells expressing WT or W8A. The results showed that WT binding significantly increased DNA–DNA contact (Fig. 6e(i)); on chromatin sites that were bound by both WT and W8A, the DNA–DNA contact strength remained significantly higher in cells expressing WT (Fig. 6e(ii)). By contrast, W8A did not significantly change DNA–DNA contacts, as shown by the sites bound by W8A only (Fig. 6e(iii)). Quantification of the contacts for each of these groups is shown in Fig. 6e(ii). A schematic diagram is provided in Fig. 6e(iii). Representative regions with DNA–DNA contacts formed between NANOG binding sites and their strengths in WT or W8A conditions are shown in Fig. 6f. These results together strongly support that WT NANOG can significantly bridge DNA–DNA contacts, but W8A mutation abolishes this ability.

## Discussion

Our data irrevocably demonstrate that NANOG oligomerizes at low nanomolar concentrations, at least three orders of magnitude lower than those of most protein assemblies (for example, amyloids, signalosomes and multivalent complexes)[41,42]. This unique property may explain NANOG's dose-sensitive action and why NANOG level is critical to the activation of pluripotency[4,5,7]. As a pioneering TF[43], NANOG associates with high-density TF/coactivator super-enhancer clusters[1,44] and interacts with satellite DNAs to decompact or remodel heterochromatin for the acquisition of pluripotency[45]. Our results suggest how NANOG can mechanistically help shape the pluripotent genome (Fig. 6g). Our data corroborate the 4C-seq results of mouse NANOG in ref. [8] that demonstrated NANOG's direct role in bringing distant loci together. The introduction of synthetic NANOG that could target a specific locus led to newfound specific *Nanog* contacts[8]. Bridging of multiple intergenomic loci in a concerted manner can be readily accomplished with a prion-like assembly mechanism (Fig. 6g). NANOG prion-like assembly may also serve to recruit other coactivators/TFs through interaction with the PrD domain or other NANOG domains. Further studies with coactivators/TFs are necessary to assess homo-/hetero-oligomeric states and stoichiometric ratios. Nevertheless, NANOG's observed dose sensitivity[4–6] is consistent with our experimental observations, as the aggregation process is a highly concentration-dependent mechanism. We propose that the rise in NANOG levels triggers a timely switch, resulting in cooperative NANOG assembly in synchrony with the chromatin reorganization required for the activation of stem cell pluripotency.

NANOG's phase transition behaviour and mechanism are distinct from KLF4's LLPS[9]. Liquids, solids and gels can develop from LLPS, depending on solution conditions, protein sequence and material properties[46]. KLF4's LLPS results from recognition of the partial cognate sequence by the multivalent zinc fingers DNA-binding domain[9]. NANOG aggregation/condensation is mainly due to the oligomerization domain and behaves more like a functional amyloid (liquid-to-solid phase transition). However, KLF4 and NANOG condensation may have distinct and overlapping cellular functions. It is possible that KLF4 condensation is more critical in the early stages of induced reprogramming, where it facilitates chromatin opening and OCT4/SOX2 cooperative recruitment, consistent with literature studies on KLF4's role in cooperating with OCT4 and SOX2[47,48] and activating NANOG expression[49,50], whereas NANOG expression and assembly is more critical in the late reprogramming stages, where it enhances and stabilizes the pluripotency promoter/enhancer contacts required for final activation of pluripotency.

## Online content

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

## Methods

**Construction of mammalian and bacterial expression plasmids.** DNA/primer sequences and constructs used in the study are listed in Supplementary Tables 3 and 4, respectively. All generated constructs and mutations were confirmed by DNA sequencing (Eurofins Genomics). Details of construct generation are provided in the Supplementary Information.

**Cells.** The *E. coli* strain DH5α (Thermo Fisher Scientific) was used for plasmid cloning and large-scale preparations of plasmid DNAs. *E. coli* strain BL21 Star (DE3; Thermo Fisher Scientific) was used for large-scale protein production. HEK 293T cells were cultivated in Dulbecco's modified Eagle medium (DMEM, Corning) with 10% (vol/vol) fetal bovine serum (FBS, Corning) and 1X antibiotic-antimycotic solution (Corning). H9 human ESCs were obtained from the Human Stem Cell Core at the Baylor College of Medicine. All cells used in this study tested negative for mycoplasma contamination. The generation of h6f-NANOG WT/W8A-eGFP H9 stable ESC lines is detailed in the Supplementary Information.

**Protein and DNA sample preparations.** Detailed methods for protein expression and purification from *E. coli* and mammalian cells, as well as protein/DNA fluorescent labelling and concentration determination, are provided in the Supplementary Information.

**Determination of NANOG solubility.** Lyophilized NANOG constructs (FL NANOG WT, NTD, DBD, CTD and WR) were dissolved in 6 M GdnHCl and incubated at room temperature (r.t.) overnight. All proteins were diluted into phosphate buffered saline (PBS; 1.8 mM $KH_2PO_4$, 10 mM $Na_2HPO_4$, 2.7 mM KCl, 137 mM NaCl, pH 7.4) with 100 μM final GdnHCl concentration, to a final concentration of 6 μM protein concentration (except for NTD, which was at 100 μM). Percent solubility is the ratio between the absorbance (UV at 280 nm) of the supernatant versus the pellet after centrifugation for 5 min at 17,000*g*.

**NANOG CTD gel formation.** NANOG CTD (200 μM) in 7.2 M GdnHCl and 10 mM dithiothreitol (DTT) was diluted into PBS buffer to a final concentration of 5 μM and incubated overnight on ice to allow settling of the NANOG CTD in the microcentrifuge tube.

**Chemical crosslinking.** *Purified GFP-tagged NANOG in vitro.* Purified proteins (h6f-NANOG WT/W8A-eGFP, 2.5 nM) were mixed with 5 mM DSSO crosslinker (Thermo Scientific) and incubated at r.t. for 15 min. Final buffer conditions consisted of 80% PBS buffer, 10% RIPA-IMI buffer and 10% dimethyl sulfoxide (DMSO). Crosslinking reactions were quenched with the gel loading buffer and run in 4–20% TGX gels. The experiments were repeated with 5 and 10 nM NANOG, with similar results.

*NANOG expressed in HEK 293T cells and H9 human ESCs. Nanog* pcDNA-based constructs (*pcDNA-H6f-Nanog, pcDNA-H6f-Nanog_W8A* and *pcDNA-H6f-Nanog_ΔWR*) were transiently transfected into HEK 293T cells using a 4D-Nucleofector X Unit following the manufacturer's protocol. Cells were grown in DMEM 10% FBS for 24 h and collected. H9 human ESCs were obtained from the Human Stem Cell Core (Baylor College of Medicine). All cells were resuspended in WC lysis buffer (20 mM HEPES pH 7.3, 150 mM NaCl, 1.5 mM $MgCl_2$ and 0.5 mM DTT) containing a protease inhibitor cocktail (GenDEPOT) and lysed by sonication. Soluble fractions from the cell lysate (total protein concentration 1 mg ml⁻¹) were crosslinked with 1 mM DSSO for 1 h at r.t. under gentle rotation. The reactions were quenched with 20 mM Tris-HCl pH 8.0. Crosslinked proteins were denatured with 6.8 M urea/1 mM DTT and concentrated by a centrifugal concentrator (10-kDa cutoff). Oligomerization was analysed by SDS–PAGE and western blot using anti-NANOG antibodies.

**Quantitation of NANOG in mammalian cells.** *H6f-NANOG-eGFP concentration in HEK 293T cells.* We quantified NANOG concentration in two ways (Extended Data Fig. 4): in-gel fluorescence and live-cell imaging. Both methods utilized calibrations of known h6g-eGFP concentrations. Purified GFP proteins and cell lysate were separated by 4–20% Tris-glycine gel (TG; Bio-Rad) SDS–PAGE gels. GFP fluorescent bands on the gel were quantified using a ChemiDoc MP image system with Image Lab Software. Nuclear concentrations of fluorescent NANOG were estimated using a nuclear volume of 220 fl (ref. [51]) and ~80% transfection efficiency.

For direct fluorescence quantitation in live cells, HEK 293T cells expressing h6f-NANOG-eGFP WT or W8A were plated on a poly-D-lysine-coated 35-mm Ibidi μ-dish and imaged using an EVOS fluorescence imaging system with ×60 objective and GFP filter (Thermo Fisher Scientific). Owing to differences in the NANOG expression levels, different exposure times and h6g-eGFP calibration plots were used: 50% power and 60-ms exposure time for the W8A mutant or 120 ms for the WT. Fluorescence images of purified h6g-eGFP proteins placed on the Ibidi μ-dish were collected and quantified using ImageJ. Blue fluorescent microsphere beads (Molecular Probe) were used to ensure the same focal plane for the images. The nuclei boundaries for the NANOG HEK 293T-expressing

cells were manually drawn in ImageJ and the mean fluorescence intensities were quantified. The calibration data were fitted to a line using Origin.

*Endogenous NANOG in H9 ESCs.* The endogenous NANOG concentration was estimated from western blots by calibration with known NANOG concentrations as standards. The cells were detached from the culture dishes using trypsin/EDTA or Accutase and counted using a TC20 automated cell counter (Bio-Rad). The detached cells were collected by centrifugation at 300*g* for 10 min at r.t. and resuspended in M-PER lysis buffer (Thermo Fisher Scientific) containing 1X protease inhibitor cocktail followed by 5-min incubation at r.t. The cell debris was removed by centrifugation at 17,000*g* for 5 min at 4 °C. Total cellular protein concentrations were estimated using UV absorbance at 280 nm (1 absorbance unit = 1 mg ml⁻¹), measured with NanoDrop (Thermo Scientific). Various protein amounts were separated on 4–20% SDS–PAGE gels (Bio-Rad). Proteins were transferred to polyvinylidene difluoride (PVDF) membranes and incubated with primary antibodies (Anti-Nanog, 1:200, Santa Cruz sc-374103 and Anti-GAPDH, 1:6,000, Millipore CB1001) overnight at 4 °C, followed by secondary antibody (anti-mouse immunoglobulin-G (IgG) horseradish peroxidase (HRP), 1:1,000, Cell Signaling #7076) at r.t. for 1 h. Bound antibodies were detected using electrochemiluminescence (ECL) substrate (Bio-Rad) and quantitated with a ChemiDoc MP imaging system (Bio-Rad). The calibration plot of NANOG standards was fit to a line using Origin. The number of moles in the samples was divided by the number of cells followed by measurement of the nuclear NANOG concentration. Nuclear NANOG concentration was estimated using a nuclear volume of 220 fl (ref. [51]).

**Stem cell pluripotency assay.** To assess the effects of NANOG WT and mutant overexpression on stem cell differentiation, H9 ESCs with lentiviral transfected NANOG WT or W8A were plated on Cultrex-coated 12-well plates. The cells were initially singularized with Accutase (Sigma-Aldrich, A6964–500ML), then counted and centrifuged. The cell pellets were recovered in StemFlex medium with 10 μM Y-27632 (Tocris, 1254) and 2 μg ml⁻¹ puromycin. Finally, each well was seeded with 40,000 cells. On the next day, the medium (without Y-27632) was changed and NANOG expression was induced with 2 μg ml⁻¹ of DOX for 3–7 days. To check for ESC colonies and undifferentiated cells, alkaline phosphatase activity was assessed using a Vector Blue alkaline phosphatase substrate kit (Vector Laboratories, SK-5300) following the manufacturer's protocol. The plates were washed with 0.5% crystal violet solution (protein/nucleic acid stain) in Dulbecco's phosphate buffered saline (DPBS) to detect all cells. The plates were imaged using a Nikon Ti2E microscope system with a Yokogawa W1 spinning disk module. The large six-well images were derived from stitching of smaller images using the NIS software (Nikon).

**ThT-detected SDD-AGE.** To confirm large aggregate/fibril formation, SDD-AGE was used. Standard agarose (1.5%) was melted in 1X Tris acetate EDTA (TAE) buffer (Fisher Scientific) combined with 0.1% SDS, 0.5 μM 2,2,2-trichloroethanol (Alfa Aesar) for stain-free detection and 1 μM ThT for fibril detection (Sigma). The hot solution was poured into an Owl EasyCast B1A mini gel system (Thermo Fisher Scientific). NANOG WR domain and CTD gel samples were first formed by diluting stock samples in 7 M GdnHCl to PBS buffer. The gel/precipitates were resuspended in 6X sample buffer (5:1) and pipetted several times before 5 min of r.t. incubation. Electrophoresis was performed in 1X TAE containing 0.1% SDS for 90 min at 80 V. After electrophoresis, the gel was imaged using stain-free and Alexa 488 filters on the ChemiDoc MP imaging system (Bio-Rad). Bovine serum albumin (BSA; 3 and 10 μg) was used as negative control.

**ThT aggregation assay.** Aggregation of the NANOG CTD was monitored by following changes in ThT fluorescence using a Spark microplate reader (Tecan) and 440-nm excitation and 480-nm emission wavelengths. Lyophilized NANOG CTD was first dissolved in 7.2 M GdnHCl with 100 mM DTT and incubated for 24 h at r.t. Protein aggregation reactions (0, 1.2 and 2.4 μM NANOG CTD) were conducted at r.t. and monitored for several hours. The final buffer conditions include 3 μM ThT, 100 mM GdnHCl in PBS buffer (1.8 mM $KH_2PO_4$, 10 mM $Na_2HPO_4$, 2.7 mM KCl, 137 mM NaCl), pH 7.4. The data were fitted to a single exponential function ($y = Ae^{Rx}$) using Origin, with *y* as the normalized fluorescence reading, *x* as time in minutes, *R* as the rate constant and $-1/R$ as the time constant of aggregation. To test the effects of Gata6-DNA, the same experiments were performed but using 3.4 μM NANOG CTD in the absence and presence of 1 μM Gata6-DNA.

**Scanning electron microscopy.** NANOG CTD gel aggregates were incubated in 300-mesh copper grids (FCF300-Cu, Electron Microscopy Sciences). After incubation, the grids were washed three times by soaking in distilled water for 30 s, then were negatively stained with 1% uranyl acetate for 30 s, followed by washing the grids three times with distilled water. The NANOG CTD grids were imaged using a high-resolution Hitachi SU-8230 scanning electron microscope using the Bright-Field Scanning Transmission Electron Microscopy (BFSTEM) mode with an accelerating voltage of 5 kV.

**Fluorescence SEC and UV-SEC.** To estimate the molecular size of full-length NANOG WT and W8A mutants at nanomolar concentrations, the fluorescence SEC (fSEC) technique was employed using a Bio-SEC3 column and HPLC Infiniti 1260 instrument (Agilent Technologies) with fluorescence detection at 488-nm excitation and 510-nm emission. The fSEC experiments were performed in RIPA-IMI elution buffer conditions with 10 nM NANOG WT/W8A or 500 nM h6g-eGFP. UV-SEC experiments were performed in high-salt buffer (25 mM HEPES, 1 M NaCl, 5 mM imidazole, pH 7.3) with 100-μl injections of 300 nM MBP-NANOG WT, 1.8 μM MBP-NANOG W8A and 9 μM h6g-NANOG/Skp (1:3) complex (direct elutions from cobalt resins). The molecular weight calibration standards (thyroglobulin (670 kDa), γ-globulin (158 kDa), ovalbumin (44 kDa), myoglobin (17 kDa) and vitamin B12 (1.35 kDa)) were detected using UV absorbance at 214 nm. SEC $M_w$ calibration plots (log $M_w$ versus retention time) were generated in Origin, and the observed $M_w$ values for the samples were calculated using the calibration plots.

**fEMSA.** The binding reactions for the EMSA consisted of EMSA buffer (0.01 mg ml$^{-1}$ BSA, 0.1 mM DTT and 0.05 mM tris(2-carboxyethyl)phosphine (TCEP), 5% glycerol, 50 mM NaCl, 20 mM Tris pH 8), protein and labelled DNA. Various protein concentrations were prepared by twofold serial dilution (either directly in RIPA-IMI buffer for h6f-NANOG WT/W8A-eGFP or in 3.6 M GdnHCl, αβγ buffer for untagged, recombinant NANOG WT/W8A). For untagged recombinant FL NANOG WT and W8A, protein stock solutions were first prepared in 7.2 M GdnHCl, αβγ buffer with 100 mM DTT, and incubated for 24 h to ensure complete unfolding. Binding reactions were made by combining each diluted protein sample with 1X EMSA buffer (with the fluorescent AF647-Gata6 DNA probe). The samples were loaded onto 5% Tris-borate–EDTA (TBE) pre-cast Mini-PROTEAN Tris-glycine gel (TG; Bio-Rad) and run for 25–45 min at 120 mV 4 °C in TG buffer. The gels were then imaged using ChemiDoc with the appropriate filters and analysed using Image Lab software (Bio-Rad).

**EMSA.** To check for the Gata6-DNA interaction with the NANOG CTD, EMSA was performed as described for the fEMSA method, except that the assay was performed with a 12% TG gel and staining with EtBr.

**CD spectroscopy.** NANOG NTD and CTD (WT and W1357A and W468A mutants) secondary structures were observed by far-UV CD spectroscopy using an AVIV model 425 CD spectrometer (AVIV) equipped with a Peltier automated temperature controller. Experiments were performed in a 0.1-cm quartz cuvette, with a wavelength bandwidth of 1 nm, a minimum averaging time of 3 s and a temperature dead band of 0.1 °C. Ellipticity was recorded from 200 to 260 nm at 21 °C (NANOG CTD) and 24 °C (NANOG NTD). All CD and absorption measurements were corrected for solvent signal. CTD WT (2 μM) and tryptophan CTD mutant (W1357A and W468A CTD; 2 μM) measurements were performed in αβγ buffer (0.2 M NaCl, 10 mM sodium acetate, 10 mM NaH$_2$PO$_4$, 10 mM glycine, pH 7.5), whereas NANOG NTD (100 μM) measurements were performed in 10 mM Tris, pH 8, 10 mM TCEP.

**NMR spectroscopy.** *NANOG NTD.* The 2D $^1$H-$^{15}$N HSQC was performed with an 800-MHz Bruker Avance HD III instrument using the Bruker hsqcetf3gpsi pulse program. The experiment was carried out at 25 °C using 500 μM NANOG NTD in 20 mM Tris, 50 mM NaCl, 10% D$_2$O pH 6.8. NMR spectra were analysed using NMRPipe[52] and NMRFAM-Sparky[53].

*NANOG CTD.* NANOG CTD (10 μM) was dissolved in 3 mM NaoAc, 10% D$_2$O, 1 mM DTT, pH 5.2. The $^1$H 1D NMR data were collected with the 800-MHz Bruker Avance HD III using the Bruker 1h ZGESGP pulse program with 256 scans for ~10 min. The data were processed using Topspin 4.03 (Bruker).

**FRAP imaging in cells.** FRAP imaging of GFP-fused NANOG WT and W8A mutant in HEK 293T cells and H9 ESCs (~3 days after DOX induction) was carried out using an LSM880 laser-scanning confocal microscope system at 37 °C and 5% CO$_2$. Identical imaging parameters were used on both WT and W8A mutant. Different nuclear region of interest (ROI) spots (1.2-μm diameter) were selected, and reference ROIs were drawn in adjacent regions (neighbouring cells). Following two or three baseline images, the ROIs were bleached for 100 iterations at 100% laser power (488 nm) and imaged for up to 30 s post-bleaching for fluorescence recovery. FRAP recovery curves were corrected for background photobleaching and normalized against pre-bleach intensity values. The FRAP data were fitted with an exponential function ($y = Ae^{Rx}$) using Origin, with $y$ as the normalized fluorescence reading, $x$ as time in seconds and $R$ as the rate constant. The recovery half-time ($\tau_{1/2}$) was calculated as $-1/R$. FRAP recovery curves (mean and s.d.) were plotted in Origin.

**smFRET.** *Gata6 intermolecular FRET mediated by NANOG.* Gata6 AF488 and Gata6 AF647 DNA (~100 pM) binding interactions in the presence and absence of NANOG WT/W8A (250 nM) were monitored by single-molecule spectroscopy using a custom-built Alba confocal laser microscopy system (ISS). smFRET measurements were conducted in EMSA buffer (0.01 mg ml$^{-1}$ BSA, 0.1 mM

DTT and 0.05 mM TCEP, 5% glycerol, 50 mM NaCl, 20 mM Tris pH 8), at r.t. (21.5 ± 1 °C) by mixing 100 pM Gata6 N-terminally labelled with Alexa Fluor 488 (FRET donor; Thermo Fisher Scientific) and 500 pM Gata6 N-terminally labelled with Alexa Fluor 647 (FRET acceptor; Thermo Fisher Scientific), with or without 250 nM NANOG WT/W8A. Measurements were performed with five replicates for the NANOG WT and two replicates for the W8A mutant. Freely diffusing FRET samples were excited with a 488-nm laser (ISS; 115 ± 5 μW) and a 594-nm laser (ISS; 115 ± 5 μW), pulse interleaved (25 ns apart)[54]. Fluorescence emission was split into donor–acceptor fluorescence by a 605-nm long-pass beamsplitter dichroic, and donor and acceptor signals were further filtered using 535/50-nm and 641/75-nm bandpass emission filters, respectively. Emission was detected using SPCM-ARQH-16 avalanche photodiode detectors (Excelitas Technologies Corp.). The smFRET data were collected using pulsed interleaved excitation (PIE)[54,55], where two pulsed lasers of different wavelengths were synchronized in a nanosecond scale to correct any fluorescence bleed-through, in time-tagged time-resolved (TTTR) mode at a sampling frequency of 2 kHz. Data acquisition and FRET efficiency analysis were performed using VistaVision (64) 4.2.220.0 (ISS), applying a binning time of 500 μs and correcting for acceptor emission due to direct excitation (1%) and fluorescence bleed-through (3%). Photocounts were time-gated during analysis to select intervals with optimal donor and acceptor signals. There were 9,109, 11,762 and 25,171 events collected for DNA samples without NANOG, and with NANOG WT and W8A mutant, respectively (Fig. 4). smFRET histograms were phenomenologically fitted to Gaussian functions using OriginPro 2020 (OriginLab). FRET efficiencies ($E_{FRET}$) were calculated (using a value of unity for γ, a correction factor dependent on donor/acceptor fluorescence quantum yields and donor/acceptor channel detection efficiencies) from the corrected donor ($I_D$) and acceptor ($I_A$) fluorescence intensities as given by

$$E_{FRET} = \frac{I_A}{I_A + \gamma I_D} \tag{1}$$

*GdnHCl denaturation of NANOG CTD.* Chemical denaturant titration was conducted using (0–7 M) and ~100-pM N-terminal (AF488/AF594-conjugated) NANOG CTD. smFRET data measurements were collected as described above.

**FFS.** The time-dependent spontaneous intensity fluctuations of the fluorescence signals were collected on a custom-built Alba confocal laser microscopy system (ISS). For FCS, freely diffusing GFP-tagged or AF488-conjugated samples were excited with a 488-nm laser (ISS; 115 ± 5 μW). For FCCS, the samples were excited with a 488-nm laser (ISS; 115 ± 5 μW) and a 594-nm laser (ISS; 115 ± 5 μW), pulse-interleaved at 25 ns apart. Fluorescence emission was split into donor–acceptor fluorescence by a 605-nm long-pass beamsplitter dichroic, and donor and acceptor signals were further filtered using 535/50-nm and 641/75-nm bandpass emission filters, respectively. Emission was detected using SPCM-ARQH-16 avalanche photodiode detectors (Excelitas Technologies Corp.) and recorded for various durations in TTTR mode at a sampling frequency of 200 kHz. Data acquisition, FCS, PCH and burst analysis were performed using VistaVision (64) 4.2.220.0 (ISS).

FFS measurements were performed in different buffer conditions: RIPA-IMI buffer (40 mM Tris, pH 7.5, 120 mM NaCl, 0.4% sodium deoxycholate, 0.8% Triton X-100, 200 mM imidazole); high-salt buffer (25 mM HEPES, 1 M NaCl, 5 mM imidazole, pH 7.6); TBST buffer (25 mM Tris, 140 mM NaCl, 0.1% Tween-20); PBS buffer (1.8 mM KH$_2$PO$_4$, 10 mM Na$_2$HPO$_4$, 2.7 mM KCl, 137 mM NaCl, pH 7.4). For FFS HEK 293T mammalian cell lysates experiments, ~1 × 10$^7$ cells from h6f-NANOG WT/W8A-eGFP, NANOG WT/ΔWR-mCherry HEK 293T stable cell lines were collected and lysed in TBST buffer (25 mM Tris, 140 mM NaCl, 0.1% Tween-20), supplemented with 150 U of benzonase (EMD Millipore), 1X protease inhibitor cocktail (GenDepot), 0.5 mM CaCl$_2$, 2.5 mM MgCl$_2$, 1 mM EDTA and 10 mM DTT. The samples were sonicated with 20 pulses at 10% power using a handheld sonicator (Microson), then incubated on ice for 30 min. The samples were clarified with centrifugation. The supernatant containing the fluorescent proteins was collected and concentrations were estimated by in-gel-based calibrations with h6g-eGFP and h6g-mCherry. The samples were diluted to ~20 nM concentrations in TBST buffer before data collection. For in vitro experiments, GFP-tagged proteins and untagged proteins were purified as described in the Protein and DNA sample preparation section (Methods). GFP-tagged samples were dialysed to the necessary buffer conditions (RIPA-IMI or high-salt buffer) overnight and quantified using in-gel eGFP calibrations (for example, Extended Data Fig. 9e). To prepare the refolded proteins (AF488-conjugated SOX2, NANOG WT and W8A) for FFS data collection, lyophilized samples were initially dissolved in 7.2 M GdnHCl αβγ buffer and then incubated at r.t. overnight. The samples were refolded (~10 nM) in PBS buffer with 10 mM DTT, pH 7.4. The samples were incubated for ~5 min or ~4 h before FFS data collection.

*FCS.* The FCS autocorrelation function was built using the ISS VistaVision (64) 4.2.220.0 software. The observation timescale was divided into intervals and the number of photons collected in each time interval was measured. The

autocorrelation function (equation ($2$)) was fitted with a physical model using a 3D Gaussian profile for the laser excitation for one photon excitation and one species:

$$G(\tau) = \left( \frac{1}{\pi\sqrt{\pi}w_0^2 z_0 C} \right) \frac{1}{\left(1 + \frac{4D\tau}{w_0^2}\right)\sqrt{1 + \frac{4D\tau}{z_0^2}}} \exp\left[ \frac{-(V\tau)^3}{\omega_0^2 z_0 \left(1 + \frac{4D\tau}{\omega_0^2}\right)\sqrt{1 + \frac{4D\tau}{z_0^2}}} \right] \quad (2)$$

where $\tau$ is the time delay of the correlation curve and $V$ is the velocity of the flow (which is zero in our case). The beam waist, $w_0$, and the beam height $z_0$ were obtained from rhodamine 110 calibration ($430\,\mu m^2\,s^{-1}$). From the autocorrelation fits, we obtained the diffusion coefficient ($D$) and concentration ($C$) of the fluorescent species. The hydrodynamic radii ($R_h$) were estimated from the diffusion coefficients using the Stokes–Einstein equation. The predicted number of residues for the corresponding $R_h$ values were estimated using empirical equations derived for denatured or folded species[35]. Statistical tests (Student's paired $t$-test) for WT and W8A experimental diffusion data were calculated using Origin.

*PCH.* PCH data were analysed at 2 kHz and fitted assuming a 3D homogeneous profile for the laser excitation, as described in the ISS manual. PCH for a homogeneously distributed brightness is expressed as

$$\prod(k;\bar{N},\varepsilon) = \sum_{N=0}^{\infty} p(N)p(k\,|N) = \sum_{N=0}^{\infty} p(N)p_{homogeneous}^{(N)}(k;V_0,\varepsilon)$$
$$= \sum_{N=0}^{\infty} \frac{(\bar{N})^N}{N!}e^{-\bar{N}}\frac{(N\varepsilon)^k}{k!}e^{-N\varepsilon} \quad (3)$$

where $p(N)$ is the Poissonian distribution of the number of molecules, with mean value $\bar{N}$, $p(k|N)$ is the conditional distribution of the number of photon counts, provided there are $N$ molecules inside the confocal volume, which is also a Poissonian distribution, with mean value $N\varepsilon$. The data were fitted to a model 'PCH with one uniform species'. PCH data simulations were performed with ISS software using the same equation ($3$) and user input values for $N$ and $\varepsilon$.

*Burst analysis.* Photon burst data from FCS data collection were analysed at 50 Hz, and binning (based on the average counts per second) histograms were generated using the ISS VistaVision software.

**FCCS.** FCCS data were collected using a set-up similar to that described for FCS. Lyophilized NANOG WT/W8A samples were initially dissolved in 7.2 M GdnHCl $\alpha\beta\gamma$ buffer then diluted to different concentrations (50–0.125 μM) with final 6 M GdnHCl and 170 mM DTT. The samples were subsequently incubated for 30 min at 37 °C then r.t. overnight for complete unfolding. For each NANOG concentration, the protein sample was directly refolded into final EMSA buffer (0.01 mg ml⁻¹ BSA, 0.1 mM DTT and 0.05 mM TCEP, 5% glycerol, 50 mM NaCl, 20 mM Tris pH 8), with 5 nM each of *Gata*6-AF488 and *Gata*6-AF647 DNA.

To avoid fluorescence crosstalk or false cross-correlation, FCCS data were collected using pulsed interleaved excitation (PIE)[54,55] as described above. Photon counts were time-gated to select for time intervals with optimal donor and acceptor emissions before analysis. FCCS data (from acceptor (a), donor (d) and donor–acceptor (da) channels) were simultaneously fitted with the 'one photon cross-correlation 3D Gaussian model' as described in the ISS manual with the following equations:

$$G_a(\tau) = \frac{1}{\pi\sqrt{\pi}w_0^2 z_0 (C_a+C_{ad})^2}$$
$$\left( \frac{C_a}{\left(1 + \frac{4D_a\tau}{w_0^2}\right)\sqrt{1 + \frac{4D_a\tau}{z_0^2}}} + \frac{C_{ad}}{\left(1 + \frac{4D_{ad}\tau}{w_0^2}\right)\sqrt{1 + \frac{4D_{ad}\tau}{z_0^2}}} \right) \quad (4)$$

$$G_d(\tau) = \frac{1}{\pi\sqrt{\pi}w_0^2 z_0 (C_d+C_{ad})^2}$$
$$\left( \frac{C_d}{\left(1 + \frac{4D_d\tau}{w_0^2}\right)\sqrt{1 + \frac{4D_d\tau}{z_0^2}}} + \frac{C_{ad}}{\left(1 + \frac{4D_{ad}\tau}{w_0^2}\right)\sqrt{1 + \frac{4D_{ad}\tau}{z_0^2}}} \right) \quad (5)$$

where the excitation volume is $V_{eff} = \frac{\pi\sqrt{\pi}w_0^2 z_0}{2\sqrt{2}}$ and the concentration of cross-correlated particles is $C_{ad} = \frac{G_x(0)}{G_a(0)G_d(0)V_{eff}}$

because the average number of cross-correlated particles per μm³ volume ($N_{ad}$) is correlated with concentration by the equation $N_{ad} = C_{ad}V_{eff}$.

Then $N_{ad} = \frac{G_x(0)}{(2\sqrt{2})G_a(0)G_d(0)}$.

**Homology modelling of NANOG WR fragments.** Homology modelling of the NANOG WR sequences 'SNQTW' and 'TQNIQSW' was performed in MODELLER software[56] using the human prion peptide (residues 170–175; PDB 2OL9) and yeast prion peptide Sup35 (residues 7–13; PDB 2OMM) as templates, respectively. We modelled the rotamers of the side chains to minimize the steric

clashing in Coot[57]. All models were extended to four layers with eight chains based on the symmetry of crystal structures using Chimera[58] and optimized by Rosetta energy minimization using Relax with all-heavy-atom constraints[59].

**ChIP-seq.** ChIP-seq was performed as previously described[36], with minor modifications. Cells were crosslinked by 1% formaldehyde (Sigma, F8775) for 10 min at r.t. After quenching with 0.125 M glycine, cell pellets were collected by scraping, and nuclei were extracted initially using buffer LB1 (50 mM HEPES-KOH (pH 7.5), 140 mM NaCl, 1 mM EDTA (pH 8.0), 10% (vol/vol) glycerol, 0.5% NP-40, 0.25% Triton X-100 and 1X protease inhibitor cocktail (Roche, 11836145001) and SUPERase inhibitor (Invitrogen, AM2694)) and subsequently with LB2 (10 mM Tris-HCl (pH 8.0), 200 mM NaCl, 1 mM EDTA (pH 8.0), 0.5 mM EGTA (pH 8.0) and 1X protease inhibitor cocktail). After centrifugation, the cell nuclei were resuspended in buffer LB3 (10 mM Tris-HCl (pH 8.0), 100 mM NaCl, 1 mM EDTA (pH 8.0), 0.5 mM EGTA (pH 8.0), 1% SDS, 0.5% *N*-lauroyl sarcosine and 1X protease inhibitor cocktail), and the chromatin was fragmented using a Q800R sonicator (QSONICA). Sheared chromatin was collected by centrifugation and incubated with ~2–3 μg of the appropriate antibodies to NANOG or H3K27ac at 4 °C overnight. The next morning, the antibody–protein–chromatin complex was retrieved by adding 30 μl of Protein G Dynabeads (Thermo Fisher Scientific, 10004D). Immunoprecipitated protein–DNA was treated with proteinase K, de-crosslinked by 65 °C heating overnight, and collected by phenol chloroform or with a Qiagen Quick DNA extraction kit. The DNA samples were subjected to sequencing library construction using the NEBNext Ultra II DNA Library Prep Kit for Illumina (NEB, E7645L).

*ChIP-seq analyses.* Sequencing reads (after removing low-quality reads; base quality score, <20) were aligned to hg19 human genome assembly using Bowtie2[60] with default parameters. Alignments were processed by SAMtools[61] to remove low mapping quality reads ('-q 30'), PCR duplicates and mitochondrial reads. Reads that passed this filter were used to call peaks with MACS2[62] with default settings. For quantification and visualization, each individual library was normalized by the total reads numbers to reads per kilobase of transcript per million reads mapped (RPKM) and then the visualization track for each library was generated with bamCoverage. Heatmaps of ChIP-seq signals for the ±3-kb regions centred around peaks were plotted using deepTools[63]. Overlapping peaks were determined by bedTools (two peaks with 1 bp overlap were considered to have overlaps).

*Motif analysis.* HOMER (hypergeometric optimization of motif enrichment)[64] was used to search for motifs enriched in each classified group of ChIP-seq sites (NANOG WT only sites, WT/W8A shared sites and NANOG W8A only sites) using default parameters with the '-mask' parameter used only with repeat-masked sequences. Random genomic background regions matching the GC content distribution of the tested sequences of each group were used as controls. The motif search was performed using the position weight matrix file of known motifs from the motif database (included in the HOMER tool set).

**Hi-C 3.0.** Hi-C 3.0 was performed based on a recent protocol[36,37,39] that is largely modified from in situ Hi-C[38]. Briefly, ~5 million cells were washed once with ice-cold PBS to remove debris and dead cells, trypsinized the culture dishes and then crosslinked with 1% formaldehyde for 10 min at r.t. Crosslinking was quenched with 0.75 M Tris-HCl at pH 7.5 for 5 min. Cells were further crosslinked with 3 mM DSG for 50 min and quenched again with 0.75 M Tris-HCl pH 7.5 for 5 min. Cell pellets were kept at −80 °C until further use. For Hi-C 3.0 experiments, crosslinked cell pellets were washed with cold PBS and then resuspended in 0.5 ml of ice-cold Hi-C lysis buffer (10 mM Tris-HCl, pH 8.0; 10 mM NaCl, 0.2% NP-40 and protease inhibitor cocktail) and rotated at 4 °C for 30 min. After one-time washing of the nuclei with 0.5 ml of ice-cold Hi-C lysis buffer, 100 μl of 0.5% SDS was used to resuspend and permeabilize the nuclei at 62 °C for 10 min. Afterwards, 260 μl H₂O and 50 μl 10% Triton-X100 were added to quench the SDS at 37 °C for 15 min. Subsequently, enzyme digestion of chromatin was performed at 37 °C overnight by adding 50 μl of 10X NEB buffer 2, 250 U of MboI (NEB, R0147M) and 250 U of DdeI (NEB, R0175L). After overnight incubation, the restriction enzymes were inactivated at 62 °C for 20 min. To fill in the DNA overhangs and add biotin tags, 35 U of DNA polymerase I (Klenow, NEB, M0210L), 10 μl of 1 mM biotin-14-dATP (Jena Bioscience, NU-835-BIO14-S) and 1 μl of 10 mM dCTP/dGTP/dTTP were added and incubated at 37 °C for 1 h with rotation. Blunt end DNA ligation was performed using 4,000 U of NEB T4 DNA ligase (NEB, M0202M) in 10X NEB T4 ligase buffer with 10 mM ATP, 90 μl of 10% Triton X-100 and 2.2 μl of 50 mg ml⁻¹ BSA at r.t. for 4 h with rotation. After ligation, the nuclei were pelleted down and resuspended with 440 μl of Hi-C nuclear lysis buffer (50 mM Tris-HCl pH 7.5, 10 mM EDTA, 1% SDS and protease inhibitor cocktail), and sheared using a QSonica 800R sonicator. Around 10% of the sonicated chromatin was subjected to overnight de-crosslinking at 65 °C and protein K treatment, followed by DNA extraction with Qiagen PCR purification columns. After DNA extraction, biotin-labelled Hi-C 3.0 DNAs were purified by 20 μl Dynabeads MyOne streptavidin C1 beads (Thermo Fisher, 65002). The biotinylated DNA on C1 beads was used to perform on-beads library making with an NEBNext Ultra II DNA library prep kit for Illumina (NEB, E7645L) following the manufacturer's

instructions. The sequencing for both ChIP-seq and Hi-C 3.0 was performed on a NextSeq 550 platform with the paired-end (PE40) mode.

*Hi-C 3.0 data processing, mapping and ICE normalization.* For Hi-C 3.0 data analysis, the method followed the standard processing pipeline of HiC-Pro[65]. Briefly, raw data were initially trimmed to remove adaptor sequences and low-quality reads by Trimmomatic[66]. The paired-end Hi-C 3.0 reads were then mapped to the human genome (hg19) using HiC-Pro[65]. After mapping, we discarded reads that mapped to the same enzyme-digestion DNA fragment, re-ligation reads and PCR duplicates. The raw contact matrices were generated at binning resolutions of 5, 10, 20, 25, 50, 100 and 250 kb. The ICE[67] normalization was applied to remove bias in the raw matrix. Concordance analysis (SCC) between Hi-C 3.0 replicates was conducted using HiCRep[40].

Mammalian chromatin is organized into several layers, including A/B compartments, TADs and chromatin loops, which are formed via potentially distinct biochemical processes, and appear differently on Hi-C contact maps[68–70]. We analysed the Hi-C 3.0 data in this Article by looking at these several layers of chromatin architectures.

*Identification of chromatin A/B compartments.* A and B compartments were identified as described previously[71,72], with some modifications. The expected interaction matrices were calculated after removing the bins that had no interactions with any other bins, most of which were in unmappable regions of the genome. For normalized 20-kb interaction matrices, observed/expected matrices were generated using a sliding-window approach with a bin size of 100 kb and a step size of 20 kb. PCA was performed on the correlation matrices generated from the observed/ expected matrices. The first principal component (E1) of the correlation matrix coupled with GC content and gene density were used to identify A/B compartments. Mitochondrial and chromosome Y were excluded from the downstream analysis (because 293T is from a female fetus). The eigenvector E1 scores from the PCA analysis are often shown in the figures; regions with E1 score >0 indicate the A compartment, and those with E1 score <0 are defined to be in the B compartment.

*Analysis of TADs.* We followed a NIH '4D nucleome' consortium standard pipeline (4dn-insulation-scores-and-boundaries-caller, https://data.4dnucleome. org/resources/data-analysis) to identify TADs. The insulation scores for each bin were computed using the 20-kb resolution of our Hi-C 3.0 data. The insulation score was calculated by sliding a $1\,Mb \times 1\,Mb$ square along the diagonal of the interaction matrix for every chromosome. A 200-kb window was used for calculation of the transition points of insulation score changes (which are defined as TAD boundaries).

*Analyses of DNA contacts between NANOG binding sites.* To perform DNA contacts analyses between NANOG binding sites, the ICE-normalized Hi-C 3.0 contacts of both bins overlapping with NANOG binding sites were extracted (to this end we only used intrachromosomal contacts for this analysis). We then used a strategy similar to the published paired-end spatial chromatin analysis (PE-SCAn[8]) to calculate DNA contacts specifically formed between NANOG binding sites. The diagram in Fig. 6d explains the strategies. Briefly, genomic bins (25 kb) that overlap NANOG binding sites were searched for their interaction with another bin on the same chromosome that also overlaps with NANOG binding sites (for each of the groups of WT only, WT/W8A shared or W8A only). A pair of DNA sites that both have NANOG binding are defined as *a* or *b* in Fig. 6d, and there can be $a_1–a_n$ and $b_1–b_n$ such bin–bin contacts for each of the three groups (WT only, shared and W8A only). The pairwise contacts in Hi-C 3.0 were calculated for the NANOG binding bins, and we also calculated the background DNA contacts in the surrounding regions. For this, we used sliding windows of 25 kb to scan both the left and right sides of the NANOG binding bin (for 250-kb distances to each side), and for each sliding bin near *a*, its Hi-C contact with the similarly slid bin near *b* was calculated; this (*a* - 25 kb) and (*b* - 25 kb) bin–bin contact serves as one of the nearby background interactions. For each *a*–*b* contact taking place between two NANOG binding sites, in total $20 \times 20$ (so 400) background bin–bin contacts were used to calculate the nearby background interactions (because −/+250 kb on two sides equals 20 bins of the scanning window). To calculate the observed/expected score (*z* axis in Fig. 6e, left), the centre bin–bin Hi-C contact between *a* and *b* was divided by the average of all the 400 background bins in the surrounding regions of *a*–*b*. The aggregated plots made from thousands of pairs of such NANOG binding sites and their nearby background regions for each group of WT only, shared and W8A only sites were used in Fig. 6e(i). The sliding-window approach of PE-SCAn[8] is a robust strategy, because the resulting background contact matrix in the surrounding regions serves as an internal normalization for the observed Hi-C 3.0 data between specific NANOG binding sites.

**Statistics and reproducibility.** No statistical method was used to predetermine sample size. No data were excluded from the analyses and the experiments were not randomized. The investigators were not blinded to allocation during the experiments and outcome assessment. Hi-C 3.0 and ChIP-seq were conducted with biological replicates, and their statistical tests are described in each figure legend or methods.

**Reporting Summary.** Further information on research design is available in the Nature Research Reporting Summary linked to this Article.

## Data availability

Source data for plots, raw data for counts and intensity measurements, and the uncropped gel images generated in this study are provided in the source data and Supplementary Information. All raw and processed high-throughput sequencing data generated in this study have been deposited to GEO under accession no. GSE190567. Additional information on sequencing data reported in this paper is available from the corresponding authors upon request. Source data are provided with this paper.

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

## Acknowledgements

We thank the Baylor College of Medicine (BCM) Optical Imaging and Vital Microscopy Core (OIVM) for use of confocal microscopes. We acknowledge J. Law and S. Corr (BCM) for SEM data collection. This work was supported by an NIGMS NIH grant (R01 GM122763) to J.C.F. Fluorescence-based methodologies for amyloid studies were supported by NINDS NIH grant support to A.C.M.F. (R01 NS105874, R21 NS107792

and R21 NS109678). W.L. is a Cancer Prevention and Research Institute of Texas (CPRIT) Scholar. Part of this work is supported by the NIH '4D Nucleome' programme (U01HL156059), NIGMS (R21GM132778, R01GM136922), CPRIT (RR160083) and the Welch Foundation (AU-2000-20190330) to W.L. and (Q1279) to B.V.V.P. J.H.L. is supported in part by a UTHealth Innovation for Cancer Prevention Research Training Program Post-doctoral Fellowship (CPRIT RP210042). Our sequencing work was conducted with the UTHealth Cancer Genomics Core (CPRIT RP180734). Research reported in this publication was also supported by the Eunice Kennedy Shriver National Institute of Child Health & Human Development of the National Institutes of Health under award no. P50HD103555 for use of the Human Neuronal Differentiation Core facility. The content is solely the responsibility of the authors and does not necessarily represent the official views of the National Institutes of Health.

## Author contributions

Conceptualization was provided by A.C.M.F. and J.C.F. Methodology was provided by A.B., W.L., A.C.M.F. and J.C.F. Software application and formal analysis were carried out by K.-J.C., M.D.Q., C.Q., A.B., L.H., B.V.V.P., S.-C.J.L., A.C.M.F. and J.C.F. Investigations were performed by K.-J.C., M.D.Q., C.Q., J.-H.L., P.S.T., M.Z., A.B., A.C.M.F. and J.C.F. Visualization and writing of the manuscript (original draft, review and editing) were carried out by W.L., A.C.M.F. and J.C.F. Supervision was provided by A.B., W.L., A.C.M.F. and J.C.F. Funding acquisition was conducted by W.L., A.C.M.F. and J.C.F.

## Competing interests

The authors declare no competing interests.

## Additional information

**Extended data** is available for this paper at https://doi.org/10.1038/s41556-022-00896-x.

**Correspondence and requests for materials** should be addressed to Wenbo Li, Allan Chris M. Ferreon or Josephine C. Ferreon.

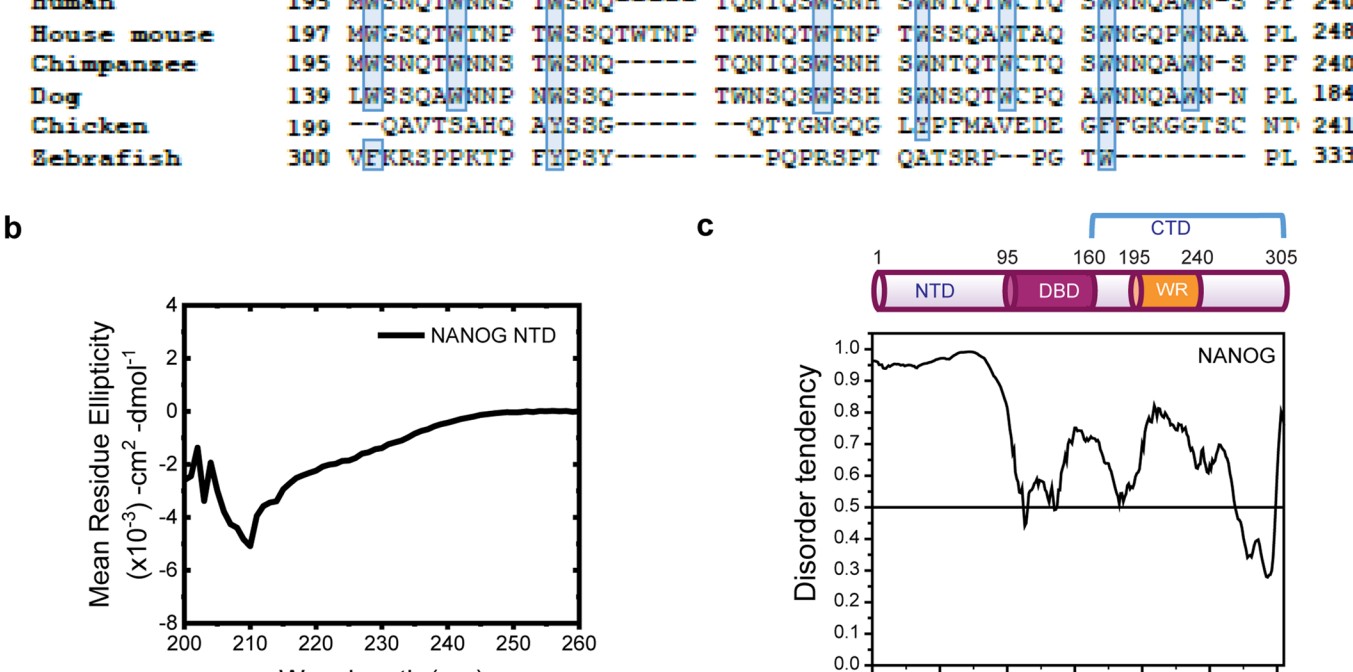

**Extended Data Fig. 1 | NANOG NTD and CTD characterization. a**, NANOG WR sequence conservation. Protein sequence alignment of NANOG WR domains using BioEdit ClustalW[73]. The sequence origins are: human (NP_079141.2), house mouse (NP_082292.1), chimpanzee (NP_001065295.1), dog (XP_025327655.2), chicken (NP_001139614.1), and zebrafish (NP_001091862.1). The relatively conserved hydrophobic residues tryptophan, tyrosine, and phenylalanine are highlighted in light blue. **b**, CD spectra of NANOG NTD (100 μM) show random coil signature. Similar results were observed for 2 independent experiments. **c**, NANOG predicted disorder. Domain organization of NANOG (top) and the disordered region prediction (bottom) by PONDR VS-L21 (www.pondr.com).

**a** h6g-NANOG IMAC Purification

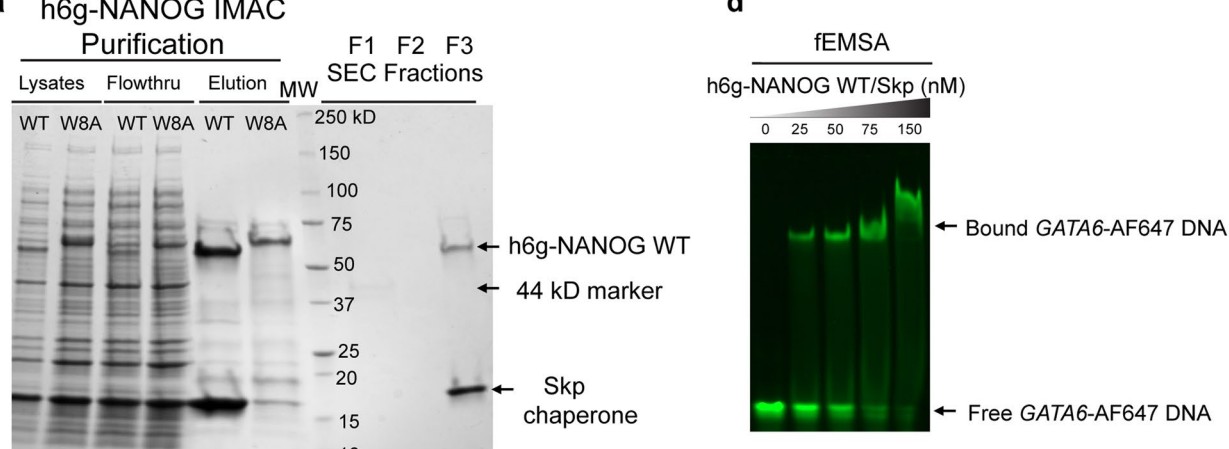

**b** MBP-NANOG IMAC Purification

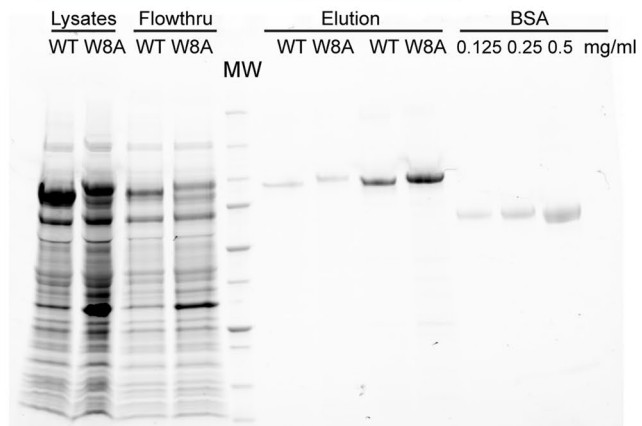

**d** fEMSA

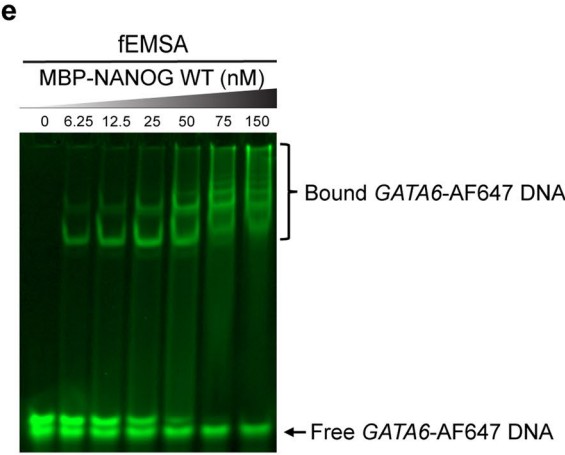

**c** h6f-NANOG-eGFP IMAC Purification

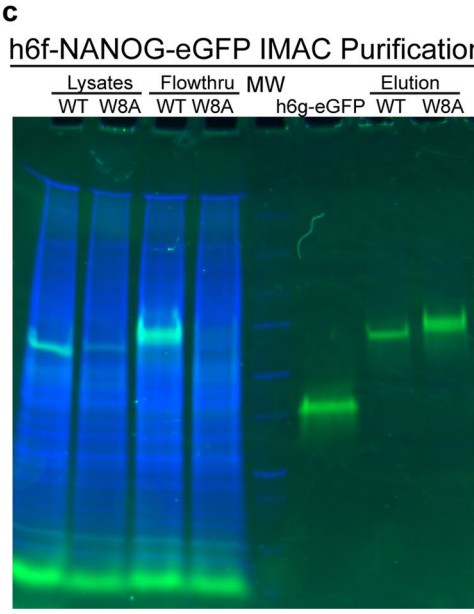

**f** Fluorescently Labeled Purified NANOG and SOX2

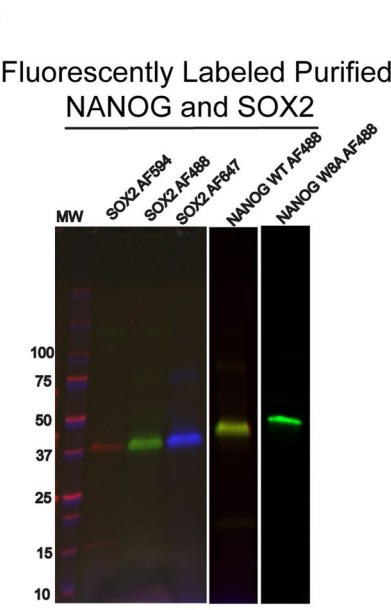

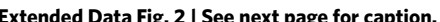

**Extended Data Fig. 2 | See next page for caption.**

**Extended Data Fig. 2 | NANOG purification and DNA binding activity. a**, Co-elution of Skp chaperone with h6g-NANOG WT by IMAC and SEC purification (Stain-free gel detection). NANOG W8A mutant did not co-elute with Skp, suggesting that the tryptophan residues mediate interactions between NANOG and Skp. **b**, IMAC purification of MBP-NANOG WT and W8A (Stain-free gel detection). Due to the presence of detergent, NANOG concentration was determined using BSA calibration. **c**, IMAC purification of h6f-NANOG WT/W8A-eGFP. NANOG concentration was determined from h6g-eGFP calibration. Green (GFP filter), Blue (Stain-free filter for all protein detection). **d**, **e**, Fluorescent EMSAs (fEMSA) with 5 nM *GATA6*-AF647 (green) and various concentrations of h6g-NANOG WT/Skp complex and MBP-NANOG WT. **f**, Fluorescently labeled purified SOX2, NANOG WT and W8A constructs. MW markers loaded in gels **a-c,f** are identical and comprised of 10,15, 20, 25, 37, 50, 75, 100,150, and 250 kD standards (as labeled in **a**). Data shown represent 2 independent experiments.

**a**

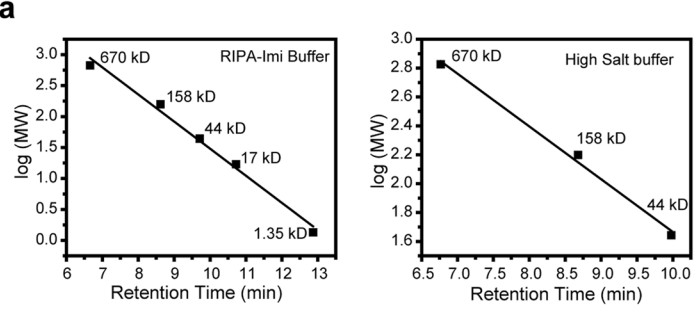

| Protein | Retention time (min) | MW (kD) | Calculated MW (kD) |
|---|---|---|---|
| *RIPA-Imi buffer* | | | |
| h6g-eGFP | 9.58 | 35.5 | 46.5 |
| h6f-NANOG WT-eGFP | 5.90 | 64.2 | 1887.5 |
| h6f-NANOG W8A-eGFP | 8.06 | 63.3 | 214.2 |
| | | | |
| *High Salt buffer* | | | |
| MBP-NANOG WT | 5.66 | 81.5 | 2406.2 |
| MBP-NANOG W8A | 8.97 | 80.6 | 85.2 |
| h6g-NANOG/SKP | 9.00 | 90.3 (1:3) | 107.8 |

**b**

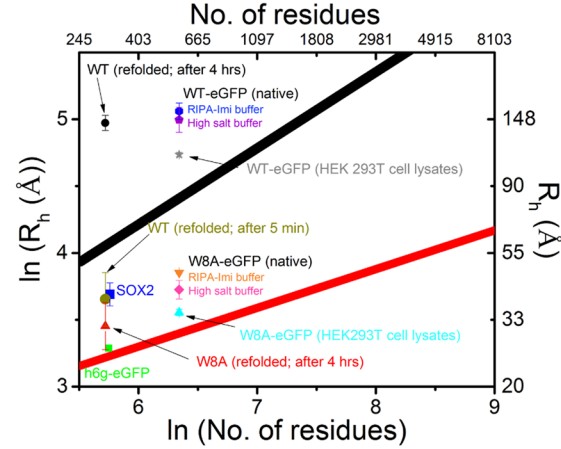

**c**

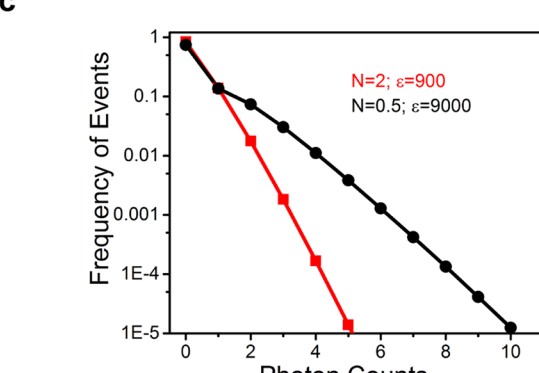

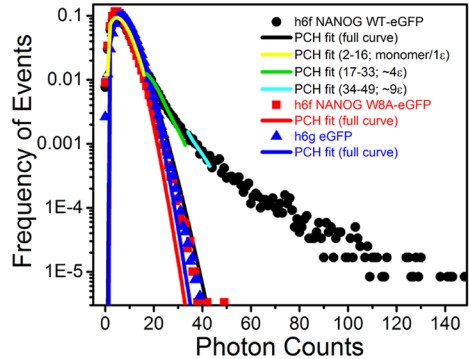

**d**

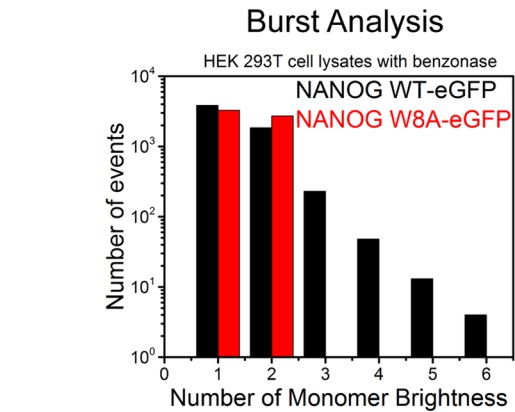

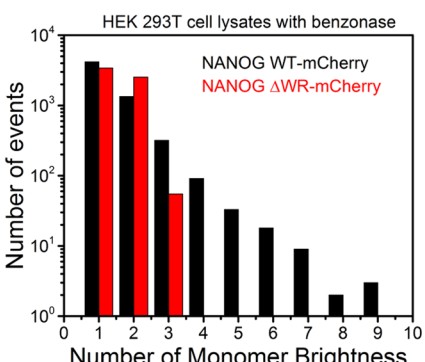

**Extended Data Fig. 3 | See next page for caption.**

**Extended Data Fig. 3 | NANOG diffusion properties. a**, fSEC and UV-SEC MW estimation of NANOG. (Top panels), SEC MW calibration plots for fSEC (h6f-NANOG WT/W8A-eGFP) and UV-SEC (MBP-NANOG WT/W8A and h6g-NANOG/Skp complex) experiments. (Bottom panel), Summary table of back-calculated MW for various proteins based on the MW calibration standards above. **b**, Plot of ln ($R_h$, hydrodynamic radii) vs ln (number of residues). The simulated lines were derived using empirical equations[35] (see Supplementary Table 1; black, denatured proteins; red, folded proteins). For convenience, the right and top axis labels correspond to actual $R_h$ values and number of residues. Also plotted are actual FCS-derived $R_h$ data (from diffusion coefficients using Stokes-Einstein equation; see Supplementary Table 1) for various NANOG constructs, SOX2 and h6g-eGFP under different experimental conditions (see Supplementary Table 1). SOX2-AF488, h6g-eGFP and W8A mutants (AF488- and GFP-tagged) fall within the boundaries for folded and denatured protein sizes. However, NANOG WT $R_h$ data (except for data taken immediately after refolding) are significantly larger than predicted from denatured proteins. **c**, Photon counting histogram simulation and fitting. (Left panel), PCH simulation of two particles with different molecular brightness. Increase in molecular brightness ($\varepsilon$) of particles results in wider Poisson distribution. (Right panel), PCH data of h6g-eGFP (▲) and W8A mutant (■) follow a Poisson distribution and the PCH curve fits (one uniform species; blue, red, respectively) approximate the actual data. In contrast, h6f-NANOG WT-eGFP PCH full data (●) deviate significantly from a fit for one uniform species (■, black). Segmental fitting of PCH data points results in different degrees of molecular brightness (yellow, 1$\varepsilon$; green ~4$\varepsilon$; cyan,~9$\varepsilon$). **e**, Burst analysis in mammalian cell lysates. (Left panel), Histogram depicts the number of events vs the 'monomer' brightness/ multiples of average counts per sec of the WT vs W8A mutant (h6f-NANOG WT/W8A-eGFP, ~20 nM). (Right panel), Histogram show the number of events vs monomer brightness of WT (NANOG-mCherry, ~20 nM) vs ΔWR mutant (NANOG ΔWR-mCherry, ~20 nM). The cells were treated with benzonase to remove protein-DNA interactions. Data shown represent 2 independent experiments.

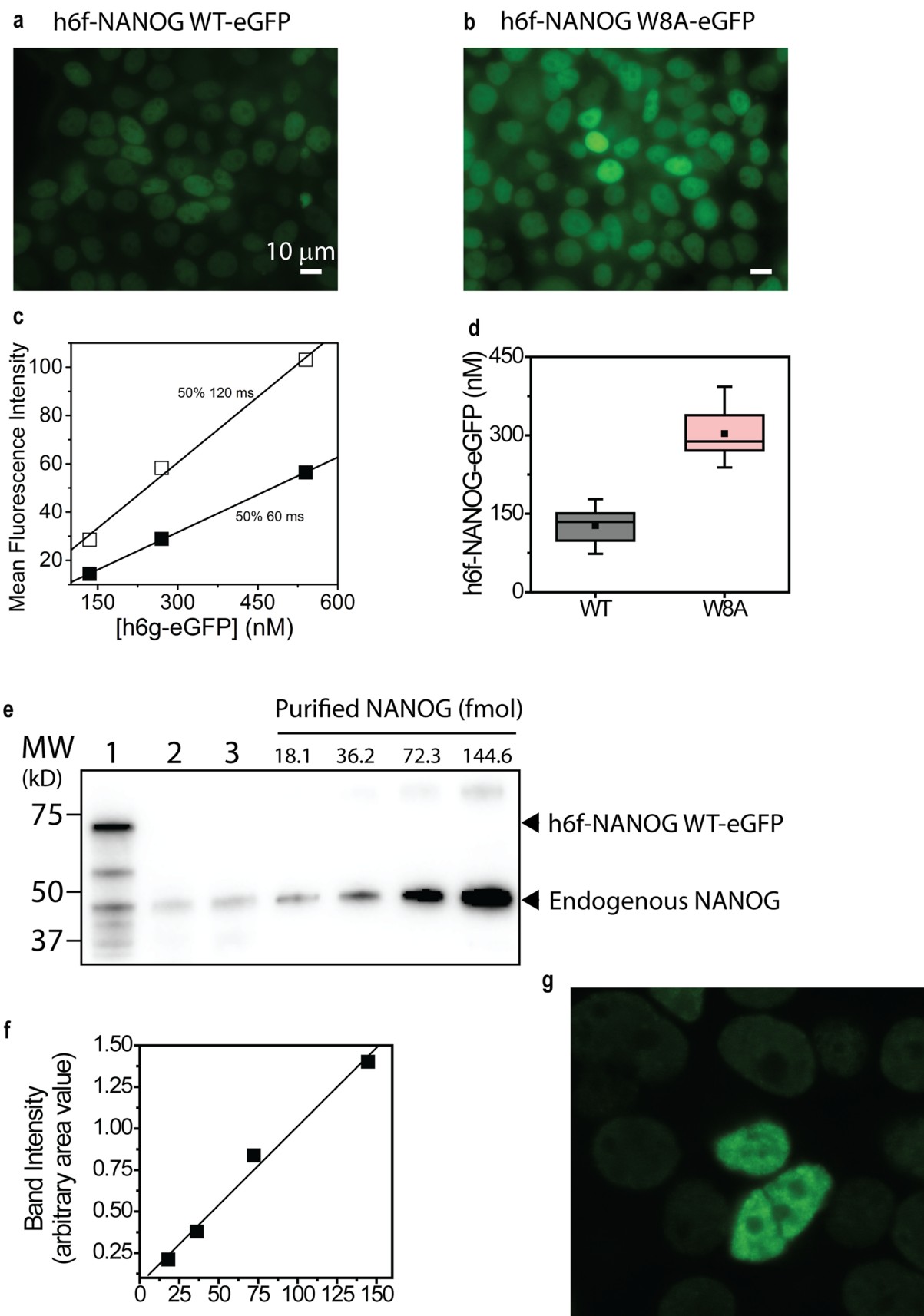

**a** h6f-NANOG WT-eGFP

**b** h6f-NANOG W8A-eGFP

10 μm

**c** Mean Fluorescence Intensity vs [h6g-eGFP] (nM)

50% 120 ms

50% 60 ms

**d** h6f-NANOG-eGFP (nM) — WT, W8A

**e**

Purified NANOG (fmol)

MW (kD) | 1 | 2 | 3 | 18.1 | 36.2 | 72.3 | 144.6

75 —

◄ h6f-NANOG WT-eGFP

50 —

◄ Endogenous NANOG

37 —

**f** Band Intensity (arbitrary area value) vs Purified NANOG (fmol)

**g**

5 μm

**Extended Data Fig. 4 | See next page for caption.**

**Extended Data Fig. 4 | Quantitation of NANOG concentration in HEK 293T stable cell line and H9 ES cells.** Quantitation of NANOG in HEK 293T cells was performed in two ways: direct live cell quantification (**a-d**), and comparison of the expressed fluorescent NANOG and endogenous NANOG against known purified NANOG concentrations detected by SDS-PAGE gels (**e-f**). **a,b**, Fluorescence microscopy images of h6f-NANOG WT-eGFP (**a**) and W8A mutant (**b**) in HEK 293T cells. WT expresses less than the W8A mutant. **c**, Due to differences in expression levels, calibration plots of h6g-eGFP were performed at the same 50% power but two different exposure times (60 and 120 ms for W8A and WT, respectively). **d**, Distribution of NANOG concentrations in HEK 293T stable cell lines (WT, n=28 and W8A, n=14 cells examined over 2 independent replicates). Mean (■), Median (----). Box limits indicate the 25th and 75th percentiles, and whiskers extend 1.5 times the interquartile range (IQR) from the 25th and 75th percentiles. **e**, Western blot quantitation of exogenous h6f-NANOG WT-eGFP in 293T cells (Lane 1, 40 μg/lane, 2.3x10$^5$ cells/lane) and endogenous NANOG in H9 ES cells (lane 2, 120 μg/lane, 3.3x10$^5$ cells/lane; lane 3, 120 μg/lane, 3.1x10$^5$ cells/lane). Estimated nuclear NANOG concentration of exogenous h6f-NANOG WT-eGFP is 750 ± 260 nM (lane 1) and endogenous NANOGs are 87.0 ± 1.0 nM (lane 2) and 160 ± 40 nM (lane 3), respectively. Data shown represent 2 independent experiments. **f**, Calibration plot of purified NANOG based on imaging band intensities (2 biological replicates for the western blots). **g**, Fluorescence microscopy image of rare (~1 in 1000 cells) HEK 293T cells with GFP-NANOG at higher expression levels and puncta compared to surrounding cells with average low NANOG expression (~150 nM). Similar results were observed in 2 biological replicates.

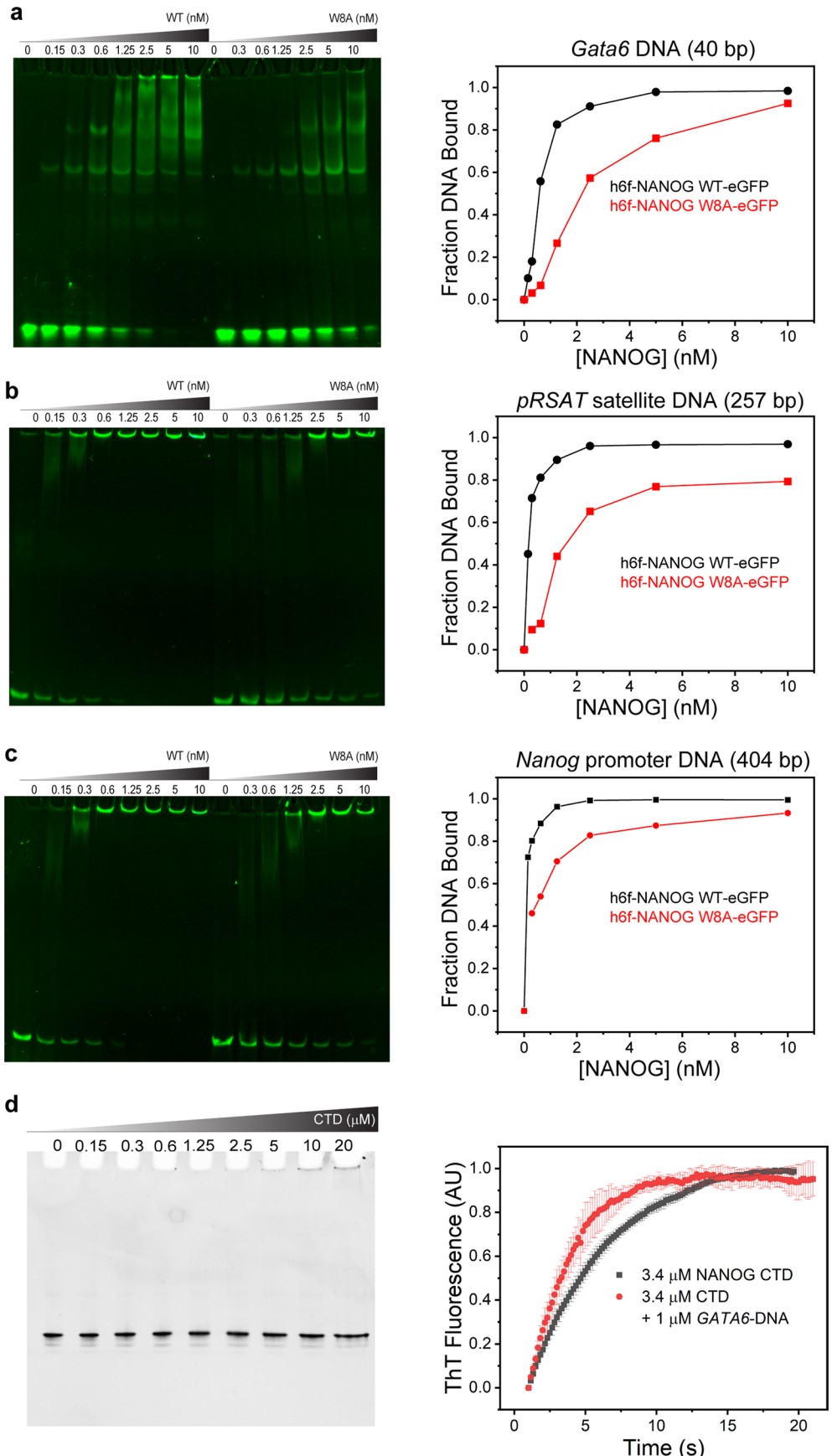

**Extended Data Fig. 5 | See next page for caption.**

**Extended Data Fig. 5 | fEMSA of NANOG WT and W8A mutant against various NANOG DNA targets.** H6f-NANOG WT-eGFP displays tighter and more cooperative binding to fluorescently labeled (AF647) DNA targets (**a**, 40 bp *GATA6* (1 nM); **b**, 257 bp *pRSAT* satellite DNA (1 nM); **c**, 404 bp *Nanog* promoter (1 nM)) than W8A mutant. Two independent measurements were performed for each DNA target. (Right panels), corresponding plots quantifying the fraction bound vs NANOG concentration from the fEMSA data shown in **a-c**, respectively. Fraction bound DNA was determined by subtracting band intensities relative to 100% free DNA (without NANOG, lanes 1 and 9). Data represent 2 independent experiments. **d**, (Left panel), NANOG CTD does not specifically interact with DNA. EMSA of NANOG CTD (0–20 μM) with 1 μM *GATA6-DNA*. (Right panel), CTD (3.4 μM) aggregation kinetics monitored by ThT fluorescence with 1 μM *GATA6-DNA*. Data are presented as mean values +/- SD; n=3 independent replicates.

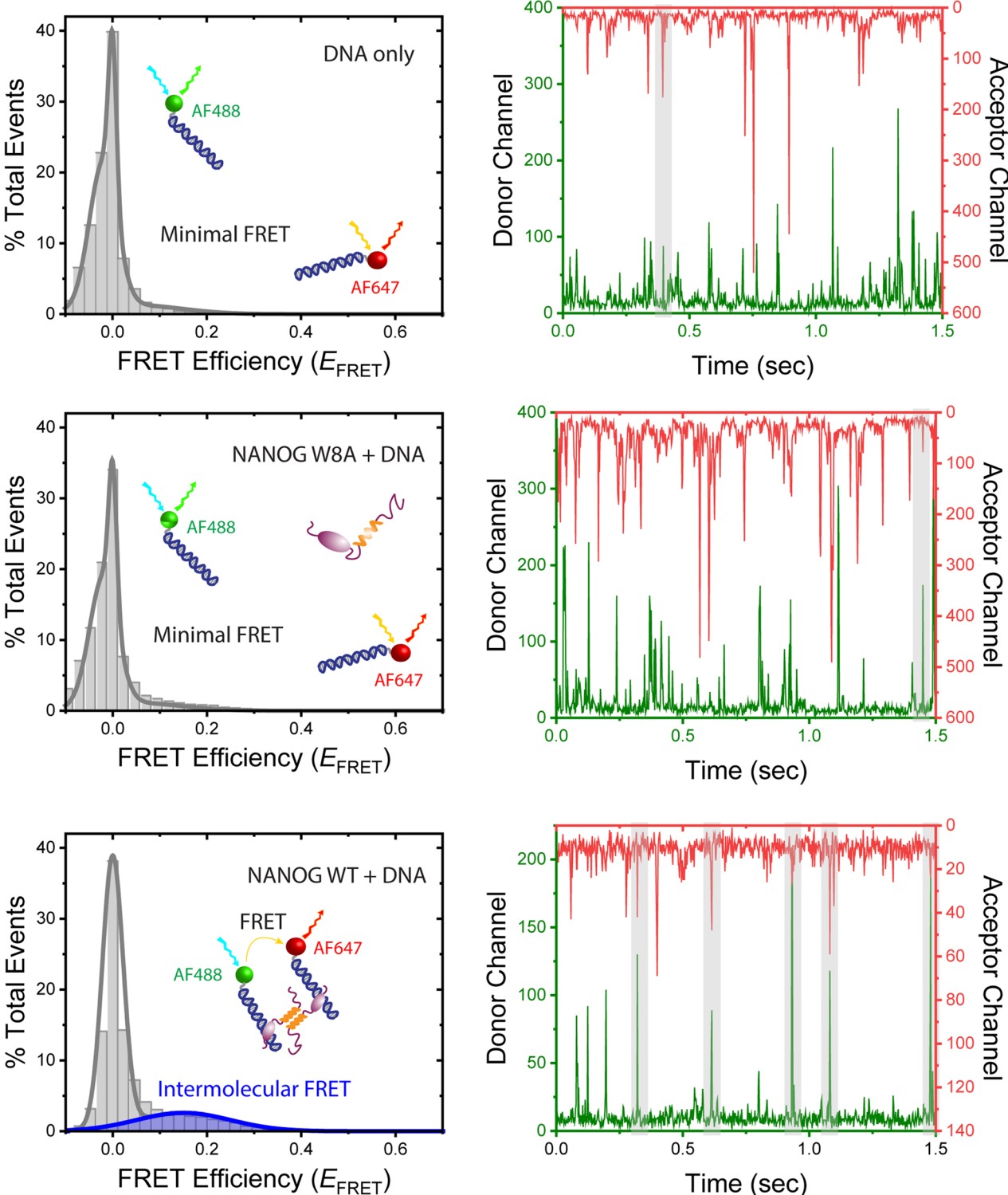

**Extended Data Fig. 6 | Representative raw smFRET data.** (Left panels) smFRET histograms (relative number of events vs. FRET efficiency ($E_{FRET}$)) of DNA alone (top), with NANOG W8A (middle) and NANOG WT (bottom). The peak at $E_{FRET}$ ~0 corresponds to AF488-conjugated bound/unbound DNA. (Right panels) Corresponding single-molecule photon bursts using PIE excitation (donor channel, green; acceptor channel, red). Highlighted gray areas indicate coincidence between donor and acceptor signals but not necessarily FRET between the bursts. From time-gating FRET analysis, intermolecular FRET efficiencies were calculated. Similar results were obtained from 2 independent experiments.

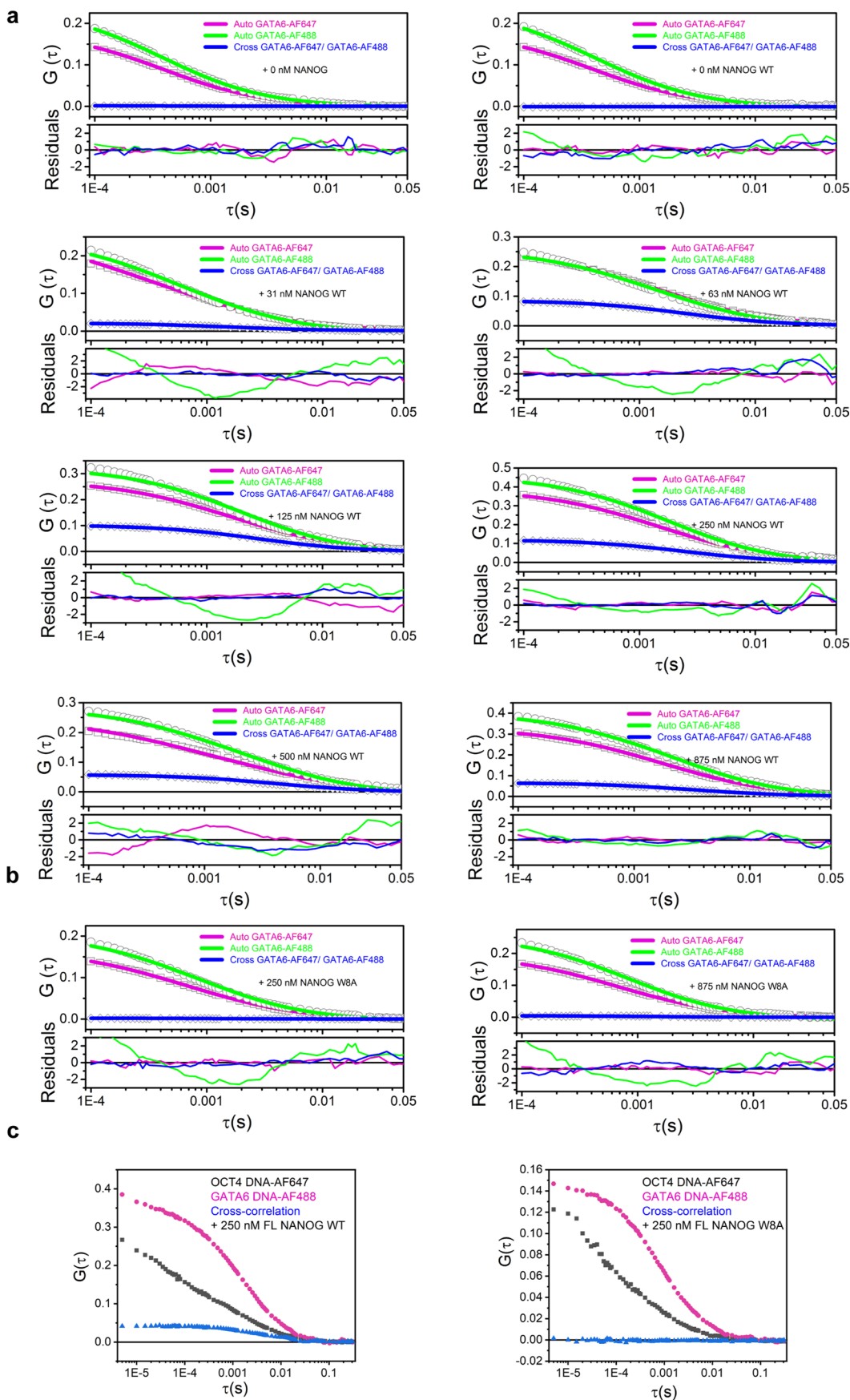

**Extended Data Fig. 7 | See next page for caption.**

**Extended Data Fig. 7 | Representative Auto and Cross-Correlation FCCS measurements. a-b**, Auto and Cross-Correlation FCCS measurements of *Gata6-DNA* (AF488/AF647) with 0–875 nM NANOG WT (**a**) and W8A (**b**). Individual normalized auto and cross-correlation functions (measurements, symbols; FCCS fits and residuals, lines). Additional data are shown in Supplementary Information. **c**, Auto and Cross-Correlation FCCS measurements of *Gata6-DNA* and *Oct4-DNA* (AF488/AF647) with 250 nM NANOG WT (left panel) and W8A (right panel). (Left panel), Auto FCCS curves of *Oct4-DNA* AF647 (black squares), *Gata6-DNA* AF488 (magenta circles) and cross-correlation curve (blue triangles) in the presence of WT NANOG (250 nM). (Right panel), Auto FCCS curves of *Oct4-DNA* AF647 (black squares), *Gata6-DNA* AF488 (magenta circles) and cross-correlation curve (blue triangles) in the presence of mutant W8A NANOG (250 nM). Cross-correlation (blue line) is observed only with NANOG WT.

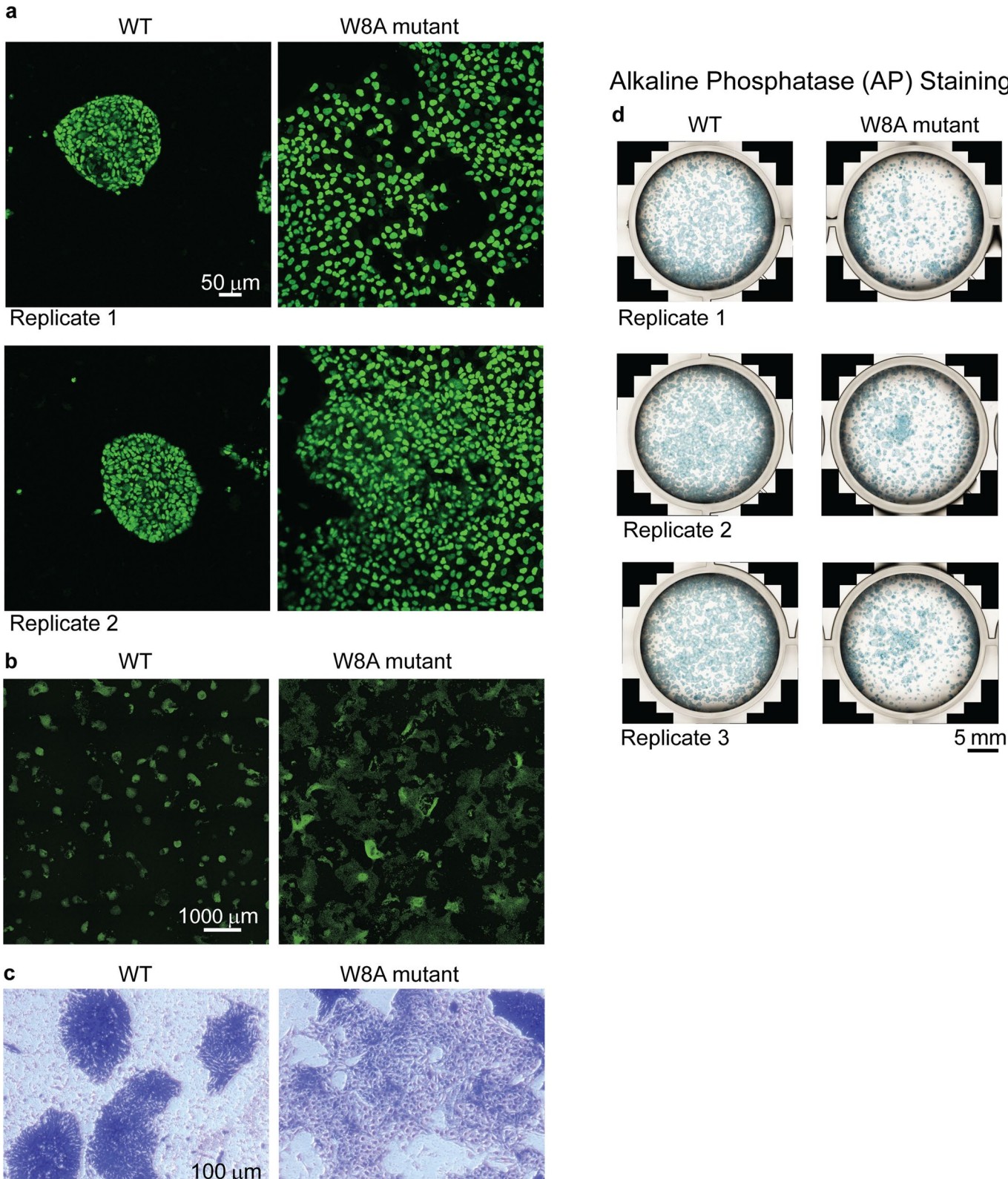

**Extended Data Fig. 8 | Stem Cell Pluripotency Assays. a,** Fluorescence microscopy images of overexpressed GFP-tagged NANOG WT (left) and W8A mutant in ESCs. Similar results were observed with 2 biological replicates. The characteristic stem cell colonies are maintained in WT but not in W8A mutant. **b,** Lower magnification fluorescence microscopy images of overexpressed GFP-tagged NANOG WT (left) and W8A mutant (right) in ESCs showing widespread differentiation in the mutant. Similar results were observed with 2 biological replicates. **c,** Crystal violet staining (stain all cells) of ESCs. Similar results were observed with 2 biological replicates. **d,** Alkaline Phosphatase (AP) staining of ESC colonies with overexpressed GFP-tagged NANOG WT (left) and W8A mutant (right). More AP+ colonies were observed with WT than mutant. Similar results were observed with 3 biological replicates.

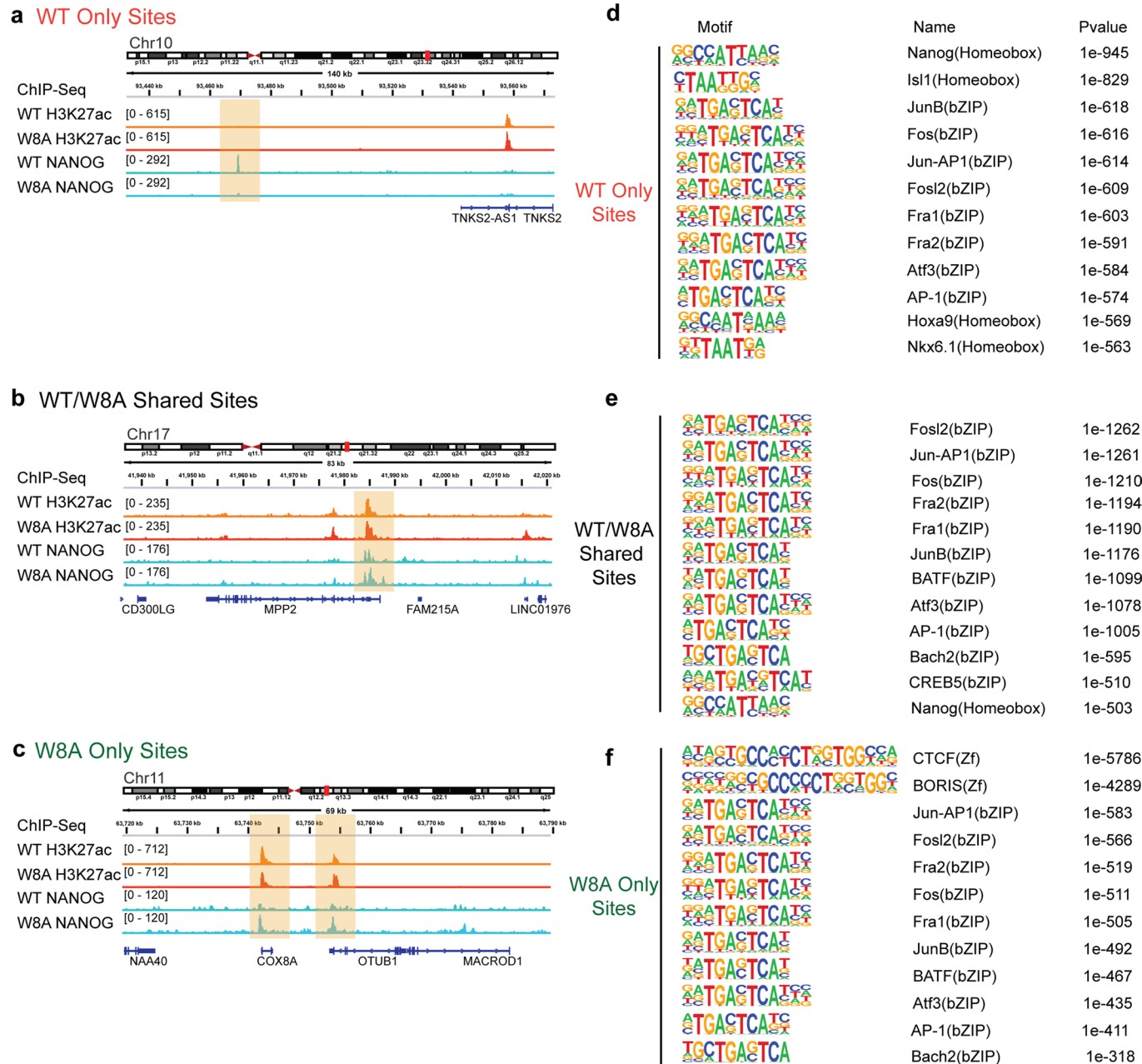

**Extended Data Fig. 9 | NANOG ChIP-seq Analysis. a-c,** Example regions of ChIP-seq binding sites observed for the three classified groups (NANOG WT Only Sites, WT/W8A Shared Sites and W8A Only Sites, respectively). **d-f,** Top 12 sequence motifs identified by HOMER for each classified group. P values were calculated by HOMER using hypergeometric tests. Select, non-redundant motifs are shown in Fig. 5c. ChIP-seq data shown were derived using 2 biologically independent replicates.

**a**

SCC correlation coefficient between replicates of Hi-C 3.0

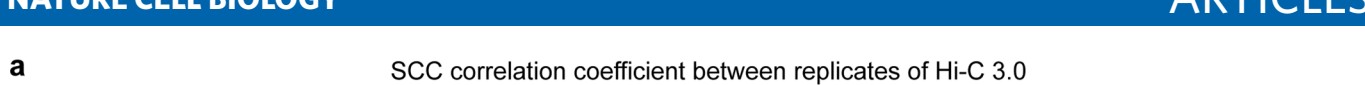

NANOG WT                                          NANOG W8A

**b**                                    **c**

Chr1:19,094,379-116,391,138

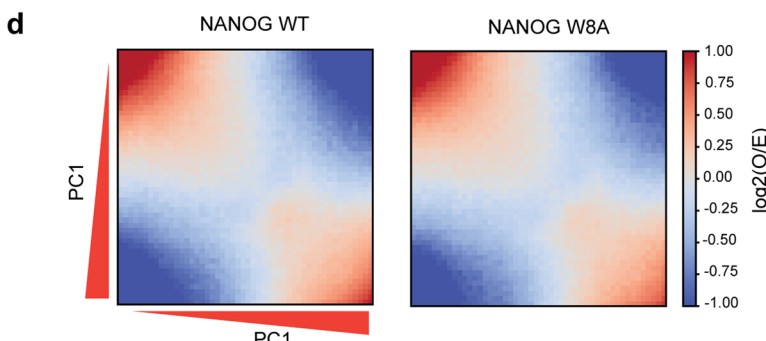

**d**

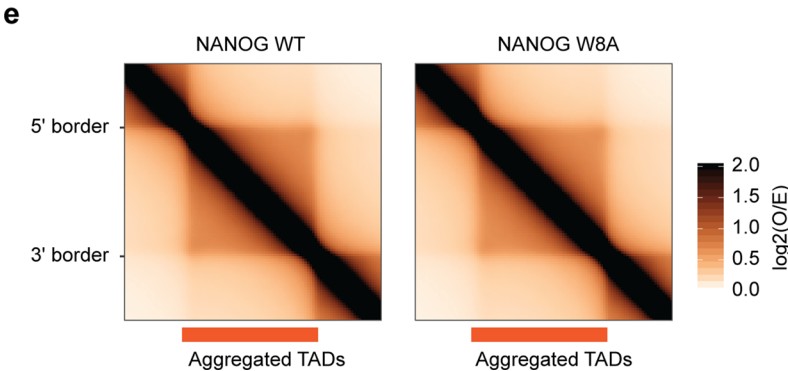

**e**

**Extended Data Fig. 10 | See next page for caption.**

**Extended Data Fig. 10 | NANOG Hi-C 3.0 Analysis. a**, Concordance of Hi-C 3.0 replicates data by the analysis of stratum adjusted correlation coefficient (SCC) between two replicates in WT and W8A conditions across different chromosomes. Left and right graphs show the SCC in HEK 293T cells expressing GFP-tagged WT or W8A NANOG, respectively. High scores of correlation indicate strong concordance of replicates. **b**, Scatter plot showing the correlation between Principal Component Analysis (PCA) E1 values of Hi-C 3.0 in WT NANOG cells (x-axis) versus E1 values in W8A NANOG cells (y-axis). E1 values were calculated for each 20 kb bin in the Hi-C 3.0 data (see Methods). **c**, An example region of ~100 Mb size from chromosome 1 showing largely identical PCA E1 values, and thus little change of A/B compartments in 293T cells expressing NANOG WT or W8A mutant. **d**, Saddle plots showing inter-compartment interactions, generated by identifying intra-chromosomal interaction frequencies between any 20-kb genomic bins that are ranked by their PCA E1 scores in the WT condition. The observed interaction frequency between any two bins was then normalized by their expected interaction frequency solely on the basis of genomic distance, which is the basis for the Observed/Expected (O/E) values to make the heatmap. The color was based on a log2 scale of the O/E values. In these plots, B–B compartmental interactions are in the upper left corner, and A–A interactions are in the lower right corner. See methods and Abramo et al[74]. **e**, Aggregate TAD analysis depicting the average contact frequency across all TADs. Hi-C 3.0 data shown were derived using 2 biologically independent replicates.

# Reporting Summary

Nature Research wishes to improve the reproducibility of the work that we publish. This form provides structure for consistency and transparency in reporting. For further information on Nature Research policies, see our Editorial Policies and the Editorial Policy Checklist.

## Statistics

For all statistical analyses, confirm that the following items are present in the figure legend, table legend, main text, or Methods section.

| n/a | Confirmed | |
|---|---|---|
| ☐ | ☒ | The exact sample size (*n*) for each experimental group/condition, given as a discrete number and unit of measurement |
| ☐ | ☒ | A statement on whether measurements were taken from distinct samples or whether the same sample was measured repeatedly |
| ☐ | ☒ | The statistical test(s) used AND whether they are one- or two-sided<br>*Only common tests should be described solely by name; describe more complex techniques in the Methods section.* |
| ☒ | ☐ | A description of all covariates tested |
| ☒ | ☐ | A description of any assumptions or corrections, such as tests of normality and adjustment for multiple comparisons |
| ☐ | ☒ | A full description of the statistical parameters including central tendency (e.g. means) or other basic estimates (e.g. regression coefficient) AND variation (e.g. standard deviation) or associated estimates of uncertainty (e.g. confidence intervals) |
| ☐ | ☒ | For null hypothesis testing, the test statistic (e.g. *F*, *t*, *r*) with confidence intervals, effect sizes, degrees of freedom and *P* value noted<br>*Give P values as exact values whenever suitable.* |
| ☒ | ☐ | For Bayesian analysis, information on the choice of priors and Markov chain Monte Carlo settings |
| ☒ | ☐ | For hierarchical and complex designs, identification of the appropriate level for tests and full reporting of outcomes |
| ☒ | ☐ | Estimates of effect sizes (e.g. Cohen's *d*, Pearson's *r*), indicating how they were calculated |

*Our web collection on statistics for biologists contains articles on many of the points above.*

## Software and code

Policy information about availability of computer code

| Data collection | For microscopic imaging, Zen 2.3, Fiji (ImageJ 1.53f51), LSM780/880 laser-scanning confocal microscope software and Nikon NIS software. For smFRET, FCS,FCCS data, we use VistaVision (64) 4.2.220.0. Details of data collection are described in methods.<br><br>We generated ChIP-Seq and Hi-C 3.0 data in HEK293T cell line using Illumina NextSeq 550 platform. The data was demultiplexed using bcl2fastq(v2.2.0). |
|---|---|
| Data analysis | For data analysis, we used OriginPro 2020 and VistaVision (64) 4.2.220.0. For microscopic imaging, Fiji (ImageJ 1.53f51). Details of data analysis were described in the manuscript.<br><br>NGS data software we used: Trimmomatic(v0.36), Bowtie2(v2.2.9), samtools(v1.9), MACS2(v2.2.7), Bedtools(v2.1.0), HiC-Pro(v2.8.10,https://github.com/nservant/HiC-Pro), cooler(https://github.com/open2c/cooler), cooltools (https://github.com/open2c/cooltools), coolpup.py(https://github.com/open2c/coolpuppy), deeptools(v3.5.0), IGV(v2.8.13). |

For manuscripts utilizing custom algorithms or software that are central to the research but not yet described in published literature, software must be made available to editors and reviewers. We strongly encourage code deposition in a community repository (e.g. GitHub). See the Nature Research guidelines for submitting code & software for further information.

## Data

Policy information about availability of data

All manuscripts must include a data availability statement. This statement should provide the following information, where applicable:

- Accession codes, unique identifiers, or web links for publicly available datasets
- A list of figures that have associated raw data
- A description of any restrictions on data availability

Source data for plots, raw data for counts and intensity measurements, and uncropped gel images generated in this study are provided in a Source Data file and Supplementary Information. All raw and processed high-throughput sequencing data generated in this study have been deposited to GEO with accession number: GSE190567. Additional information on sequencing data reported in this paper is available from the corresponding authors upon request.

# Field-specific reporting

Please select the one below that is the best fit for your research. If you are not sure, read the appropriate sections before making your selection.

☒ Life sciences  ☐ Behavioural & social sciences  ☐ Ecological, evolutionary & environmental sciences

For a reference copy of the document with all sections, see nature.com/documents/nr-reporting-summary-flat.pdf

# Life sciences study design

All studies must disclose on these points even when the disclosure is negative.

| | |
|---|---|
| Sample size | No statistical method was used to predetermine sample size. The sample sizes were ~50-100 cells (each biological replicate) for the cell imaging experiments, consistent with those reported in the literature. |
| Data exclusions | There were no data exclusions. |
| Replication | All attempts at replication were successful. There were 2-3 independent replicate experiments for in vitro and in cell studies. |
| Randomization | No randomization was explicitly performed because analysis is required, consistent with typical practice in biophysical experiments. |
| Blinding | Blinding was not performed for most experiments because analysis is required, consistent with typical practice in biophysical experiments. However, for some FCCS data experiments, blinding was performed because the data collector have no information on the samples. |

# Reporting for specific materials, systems and methods

We require information from authors about some types of materials, experimental systems and methods used in many studies. Here, indicate whether each material, system or method listed is relevant to your study. If you are not sure if a list item applies to your research, read the appropriate section before selecting a response.

### Materials & experimental systems

| n/a | Involved in the study |
|---|---|
| ☐ | ☒ Antibodies |
| ☐ | ☒ Eukaryotic cell lines |
| ☒ | ☐ Palaeontology and archaeology |
| ☒ | ☐ Animals and other organisms |
| ☒ | ☐ Human research participants |
| ☒ | ☐ Clinical data |
| ☒ | ☐ Dual use research of concern |

### Methods

| n/a | Involved in the study |
|---|---|
| ☐ | ☒ ChIP-seq |
| ☒ | ☐ Flow cytometry |
| ☒ | ☐ MRI-based neuroimaging |

## Antibodies

| | |
|---|---|
| Antibodies used | Anti-Nanog, 1:200 Santa Cruz sc-374103; Anti-GAPDH, 1:6000 Millipore CB1001; Anti-Mouse IgG, 1:1000, Cell Signaling #7076)<br><br>For ChIP-seq experiments, the antibodies (3 ug) we used are described in the Methods section, including commercially available antibodies for NANOG (R&D Systems, AF1997-SP, Polyclonal Goat IgG) and H3K27ac (Abcam, AB4927, Rabbit polyclonal antibody). We use 3ug antibody for each ChIP-seq. |
| Validation | Anti-Nanog was validated by WB with purified Nanog protein, and validated that it does not react with Oct4 and Sox2. |

| Validation | All antibodies for ChIP-seq we used here have been validated by manufacturers. Furthermore, these antibodies have been widely used in the human genomic studies; such as PMID:30122536, PMID:33828098 and PMID:33915080. |
|---|---|

## Eukaryotic cell lines

Policy information about cell lines

| Cell line source(s) | The HEK 293T cells were obtained from ATCC; H9 ESC from WiCell and Lenti-X 293T from TaKaRa Bio. |
|---|---|
| Authentication | The cell lines were not authenticated. Authentication came from the source. |
| Mycoplasma contamination | All cells used in this study tested negative for mycoplasm contamination. |
| Commonly misidentified lines (See ICLAC register) | No commonly misidentified lines were used in the study |

## ChIP-seq

### Data deposition

☒ Confirm that both raw and final processed data have been deposited in a public database such as GEO.

☒ Confirm that you have deposited or provided access to graph files (e.g. BED files) for the called peaks.

| Data access links May remain private before publication. | GSE190567 (https://www.ncbi.nlm.nih.gov/geo/query/acc.cgi?acc=GSE190567) |
|---|---|
| Files in database submission | all RAW FASTQ, BIGWIG and mcool were uploaded to NCBI GEO. |
| Genome browser session (e.g. UCSC) | https://genome.ucsc.edu/s/qicy/NANOG |

### Methodology

| Replicates | We did two replicates for NANOG ChIP-Seq experiments; two replicates for H3K27ac in 293T with overexpressing NANOG WT or W8A. |
|---|---|
| Sequencing depth | HEK293T-NANOG_WT_H3K27ac_ChIP-Seq_rep1,ChIP-seq, overexpressing NANOG WT, H3K27ac,biological replicate1, 16286301 HEK293T-NANOG_WT_H3K27ac_ChIP-Seq_rep2,ChIP-seq, overexpressing NANOG WT, H3K27ac,biological replicate2, 10863588 HEK293T-NANOG_W8A_H3K27ac_ChIP-Seq_rep1,ChIP-seq, overexpressing NANOG W8A,H3K27ac,biological replicate1, 18761363 HEK293T-NANOG_W8A_H3K27ac_ChIP-Seq_rep2,ChIP-seq, overexpressing NANOG W8A,H3K27ac,biological replicate2, 9604136 HEK293T-NANOG_WT_NANOG_ChIP-Seq_rep1,ChIP-seq, overexpressing NANOG WT,NANOG,biological replicate1, 18455755 HEK293T-NANOG_WT_NANOG_ChIP-Seq_rep2,ChIP-seq, overexpressing NANOG WT,NANOG,biological replicate2, 24,826,386 HEK293T-NANOG_W8A_NANOG_ChIP-Seq_rep1,ChIP-seq, overexpressing NANOG W8A ,NANOG,biological replicate1, 28767501 HEK293T-NANOG_W8A_NANOG_ChIP-Seq_rep2,ChIP-seq, overexpressing NANOG W8A ,NANOG,biological replicate2, 25916080 |
| Antibodies | ChIP-seq experiments, The antibodies we used are described in this study, including NANOG (R&D Systems AF1997) and H3K27ac (Abcam, AB4927); 3ug antibodies were used for each ChIP-seq. |
| Peak calling parameters | MACS2 (v2.2.7) was used to call ChIP-Seq peaks with --nomodel and --qvalue 0.01 parameter. |
| Data quality | fastqc was used to check raw fastq quality. |
| Software | Trimmomatic(v0.36), Bowtie2(v2.2.9) and MACS2(v2.2.7) |

