## [Peer Review File · Nature Cell Biology]

Peer Review Information

Journal: Nature Cell Biology

Manuscript Title: NANOG prion-like assembly mediates DNA bridging to facilitate chromatin reorganization and activation of pluripotency

Corresponding author name(s): Wenbo Li, Allan Chris M. Ferreon, Josephine C. Ferreon

Reviewer Comments & Decisions:

Decision Letter, initial version:

Subject: Decision on Nature Cell Biology submission NCB-F46094-T

Message:

*Please delete the link to your author homepage if you wish to forward this email to co-authors.

Dear Dr Ferreon,

Your manuscript, "NANOG prion-like assembly mediates DNA bridging", has now been seen by 3 referees, who are experts in phase separation (referees 1 and 2) and pluripotency (referee 3). As you will see from their comments (attached below) they find this work of potential interest, but have raised substantial concerns, which in our view would need to be addressed with considerable revisions before we can consider publication in Nature Cell Biology.

Nature Cell Biology editors discuss the referee reports in detail within the editorial team, including the chief editor, to identify key referee points that should be addressed with priority, and requests that are overruled as being beyond the scope of the current study. To guide the scope of the revisions, I have listed these points below. I should stress that the referees' concerns point to a premature dataset and these points would need to be addressed with experiments and data, and reconsideration of the study for this journal and re-engagement of referees would depend on strength of these revisions.

In particular, it would be essential to:

a) validate the key discoveries in cells, including the formation of gel-like NANOG condensates and the connection between NANOG's ability to form gel-like condensates and its function on DNA contact formation by performing genomic analysis such as Hi-C or 3C, as noted by:

Referee 1:

However, the study could benefit from additional experiments to show that the DNA bridging phenomenon is occurring in vivo and that this is tied to how NANOG works. That's where I find the study to be lacking, particularly for Nature Cell Biology readership. The authors even have performed work in examining endogenous NANOG in human embryonic stem cells, but there's no in vivo data linking oligomerization or expression level control to NANOG function. I think it would markedly enhance the story if they could correlate expression level with DNA bridging potency, for example.

2) Extensive work was done in characterizing NANOG from cell lysates, including crosslinking and fluorescence fluctuation spectroscopy. I'm interested in knowing about NANOG's propensity to form puncta in cells. Curiously, Extended Data Fig 9 shows a diffuse distribution of h6f-NANOG WT-eGFP in cell nuclei. Is this not surprising given the propensity for NANOG WT to oligomerize particularly at the estimated concentrations? On a similar note, how does endogenous NANOG stain?

Referee 2:

4) In lines 180-181, the authors posit that NANOG forms oligomers in cells. The authors seem to be in prime position to test this. To test this, mEos2-NANOG WT or W8A could be expressed in cells, a portion of the expressed protein could be photoconverted, and molecules tracked using single molecule fluorescence microscopy. While not necessary for their conclusions, this experiment would offer experimental insight into cellular oligomerization of WT-NANOG.

Referee 3:

3. It remains unclear what impact the ability of NANOG to oligomerize, phase-separate and establish DNA-bridges has on chromatin structure in PSCs. The authors should consider performing Hi-C or minimally 3C assays for select genes in PSCs expressing WT and mutant NANOG to assess their effects on 3D chromatin architecture.

b) add stem cell functional analyses, as noted by referee 3:

For example, I would find it important to show at least some functional pluripotency assays using the mutants the authors generated, specifically the W8A version of NANOG (either via overexpression or knock-in in PSCs).

2. To be a contender for NCB, the authors should at least provide some basic pluripotency assays of the mutants they've generated, e.g. overexpression of WT vs W8A NANOG in PSCs under self-renewal vs differentiation conditions.

c) Show specificity of NANOG's ability to oligomerize and phase-separate, as noted by referee 3:

1. The authors claim that NANOG's ability to oligomerize and phase-separate may explain its unique dose-sensitivity in PSCs. However, the authors also state that they have unpublished data on SOX2 and KLF4 undergoing phase separation, raising questions about specificity. I'd find it important to repeat at least some of the assays with a well-known pluripotency factor that does not form condensates or phase-separates, otherwise the specificity of this observation and its functional consequences remain unclear.

d) All other referee concerns pertaining to strengthening existing data, providing controls, methodological details, clarifications and textual changes, should also be addressed.

e) Finally please pay close attention to our guidelines on statistical and methodological reporting (listed below) as failure to do so may delay the reconsideration of the revised manuscript. In particular please provide:

We would be happy to consider a revised manuscript that would satisfactorily address these points, unless a similar paper is published elsewhere, or is accepted for publication in Nature Cell Biology in the meantime.

- ensure that it conforms to our format instructions and publication policies (see below and <https://www.nature.com/nature/for-authors>).

- provide a point-by-point rebuttal to the full referee reports verbatim, as provided at the end of this letter.

- provide the completed Reporting Summary (found here <https://www.nature.com/documents/nr-reporting-summary.pdf>). This is essential for reconsideration of the manuscript will be available to

editors and referees in the event of peer review. For more information see <http://www.nature.com/authors/policies/availability.html> or contact me.

When submitting the revised version of your manuscript, please pay close attention to our [href="https://www.nature.com/nature-research/editorial-policies/image-integrity">Digital Image Integrity Guidelines](https://www.nature.com/nature-research/editorial-policies/image-integrity). and to the following points below:

Nature Cell Biology is committed to improving transparency in authorship. As part of our efforts in this direction, we are now requesting that all authors identified as 'corresponding author' on published papers create and link their Open Researcher and Contributor Identifier (ORCID) with their account on the Manuscript Tracking System (MTS), prior to acceptance. ORCID helps the scientific community achieve unambiguous attribution of all scholarly contributions. You can create and link your ORCID from the home page of the MTS by clicking on 'Modify my Springer Nature account'. For more information please visit www.springernature.com/orcid.

This journal strongly supports public availability of data. Please place the data used in your paper into a public data repository, or alternatively, present the data as Supplementary Information. If data can only be shared on request, please explain why in your Data Availability Statement, and also in the correspondence with your editor. Please note that for some data types, deposition in a public repository is mandatory - more information on our data deposition policies and available repositories appears below.

[REDACTED]

We would like to receive a revised submission within six months.

We hope that you will find our referees' comments, and editorial guidance helpful. Please do not hesitate to contact me if there is anything you would like to discuss.

Best wishes,

Jie Wang

Jie Wang, PhD
Senior Editor
Nature Cell Biology

Tel: +44 (0) 207 843 4924
email: jie.wang@nature.com

Reviewers' Comments:

Reviewer #1:

Remarks to the Author:

This is a very interesting, technical, biophysical and cell biology study that examines the biophysical properties of the master transcription factor NANOG. Choi et al. identify a possible role of the Trp-containing, prion-like CTD of NANOG in driving NANOG oligomerization and DNA bridging. NANOG expression levels are highly regulated in cells and this dose sensitivity potentially confers NANOG functionality in promoting ground state pluripotency. First, the authors provide compelling results regarding NANOG's ability to oligomerize. The authors use a battery of biophysical methods including smFRET, SEC, and crosslinking studies to probe the extent of oligomerization, in addition to mutants that are largely monomeric, and perform a side-by-side comparison of WT and mutant properties. They also find that NANOG and these variants oligomerize to similar extents in vivo by investigating several NANOG constructs (with different solubility tags and fusion proteins) in several cell lines. Second, the authors find that NANOG can 'bridge' DNA, and they investigated this using purified NANOG assemblies. This is an interesting study that proposes a link between NANOG oligomerization and DNA bridging that potentially confers functionality that is tunable based on NANOG expression level. However, the study

could benefit from additional experiments to show that the DNA bridging phenomenon is occurring in vivo and that this is tied to how NANOG works. That's where I find the study to be lacking, particularly for Nature Cell Biology readership. The authors even have performed work in examining endogenous NANOG in human embryonic stem cells, but there's no in vivo data linking oligomerization or expression level control to NANOG function. I think it would markedly enhance the story if they could correlate expression level with DNA bridging potency, for example. I do want to reiterate that the work is high-quality and they have examined a difficult system that oligomerizes at very low concentrations (low nM) that makes it challenging to examine the molecular mechanism involved in self-assembly. Additionally, I have suggestions below on probing the role of DNA in NANOG oligomerization (along the lines of how nucleic acids contribute to formation of protein/nucleic acid granules in cells).

In addition to the above comments, I have the following concerns:

- 1) The authors mention that NANOG CTD forms gel-like condensates – the study doesn't currently probe their liquidity and that would be recommended if the 'condensates' terminology is to be used to describe their morphology.
- 2) Extensive work was done in characterizing NANOG from cell lysates, including crosslinking and fluorescence fluctuation spectroscopy. I'm interested in knowing about NANOG's propensity to form puncta in cells. Curiously, Extended Data Fig 9 shows a diffuse distribution of h6f-NANOG WT-eGFP in cell nuclei. Is this not surprising given the propensity for NANOG WT to oligomerize particularly at the estimated concentrations? On a similar note, how does endogenous NANOG stain?
- 3) Related to the oligomer sizes reported, have the authors performed DLS (dynamic light scattering) studies to look at particle size of NANOG oligomers?
- 4) Have the authors tried to mix the NANOG CTD with DNA? Part of this experiment would address a question as to whether the prion-like CTD also interacts with DNA directly. The authors should try incubating the CTD with DNA and attempt a refolding experiment to see if CTD aggregation propensity is altered by DNA; this would provide additional evidence that DNA could be integral to how NANOG oligomerizes.
- 5) The authors use fluorescently-labeled DNA on the 5' end for their critical smFRET diffusion experiments to demonstrate DNA bridging in Figure 4. To corroborate their data, could the authors also label one of the DNA molecules with a 3' fluorescent probe to provide another set of experimental data that could provide additional distance constraints on the proximity of the two labeled DNA molecules? Another suggestion could be to use two entirely different DNA sequences (each with different dyes) that could mimic what is happening in the cell.

6) A suggestion – as the authors show that W8A is monomeric, could the authors make a W8A CTD, label with ^{15}N , and collect a NMR spectrum to compare against the WT CTD in Extended Data? This would further demonstrate that the severe peak broadening in WT CTD is a result of oligomerization.

7) Trp to Ala mutations are substantial. Are there other mutations that could be made that create a NANOG mutant that is intermediate in oligomerization behavior between NANOG W8A and NANOG WT?

I found the manuscript to be data-rich and very concise.

However, there are minor errors:

Line 73 – should be extended data figure 2?

Line 85 – should be Figure 2?

Line 89 – Figure reference correct?

Figure 4c – y-axis needs a label

Extended Data Figure 4 – need MW markers on gels in panel a and b at least

Reviewer #2:

Remarks to the Author:

In the manuscript entitled “NANOG prion-like assembly mediates DNA bridging”, Choi and colleagues investigate the ability of NANOG to form higher ordered structures using a combination of in vitro biochemistry and cell biology. The authors show that full-length NANOG forms higher order oligomers at extremely low concentrations and posit that this oligomerization enables NANOG to bridge DNA during the formation of DNA condensates.

The data for NANOG oligomerization presented in this manuscript is clear and convincing. In a couple of instances that this reviewer describes below, the authors seem to be in prime position to extend the study a bit further to continue to unravel the biophysical mechanisms that regulate NANOGs cellular functions. Additionally, the authors promote a dose-sensitive mechanism of NANOG function in cells in both their abstract, intro, and conclusion, but don't explicitly tie their results to this mechanism. A deeper discussion of how their data relates to dose sensitivity is necessary. If this can be addressed in the text, this paper is a strong candidate for publication in Nature Cell Biology.

Comments:

1) The authors perform well-controlled biochemical experiments to investigate the mechanism underlying cellular crosslinking experiments shown in Figure 3. While the results of these experiments are convincing and suggest that NANOG oligomerization can indeed account for the observed band

shifts in the gel, the cellular environment is far more complex than in in vitro experiments. It would be interesting to know if the cellular complexes include specific binding partners or if they are mostly NANOG. If experimentally possible, NANOG pulldown followed by mass spectrometry analysis may be able to parse the composition of the complexes and provide additional insight into NANOG oligomer interactions in cells.

2) In the fSEC chromatograms, GFP-NANOG appears to be eluted over multiple peaks, not just in the void. The right-most peak is slightly shifted when compared with GFP-NANOG W8A or GFP alone, suggesting that this may be some degradation product or that NANOG interacts with the fused GFP. Were the contents of this peak analyzed? If so, is the protein in this peak identifiable? If the protein in this peak is GFP-NANOG, does this suggest that the fusion of GFP to NANOG destabilizes the higher order complexes that are observed with other versions of NANOG? A comment from the authors would be helpful to properly understand the data.

3) The FFS and FCS data provides convincing evidence that WT NANOG forms higher-order oligomers. Is it possible to also run DLS on WT- and W8A-NANOG in vitro to determine whether these oligomers are mono- or poly-dispersed. This measurement would indicate whether WT NANOG forms a single oligomeric species or oligomers of random size. This type of data would also provide insight into potential cellular mechanisms that are described in the authors' model in Figure 4G.

4) In lines 180-181, the authors posit that NANOG forms oligomers in cells. The authors seem to be in prime position to test this. To test this, mEos2-NANOG WT or W8A could be expressed in cells, a portion of the expressed protein could be photoconverted, and molecules tracked using single molecule fluorescence microscopy. While not necessary for their conclusions, this experiment would offer experimental insight into cellular oligomerization of WT-NANOG.

5) In the text in the top paragraph on page 3, the authors refer to Figures 1C, 1D, and 1E. These should be Figures 2C, 2D, and 2E.

6) Low and high levels are mentioned to describe this dose-dependency. It would be helpful for the authors to discuss what these dosages or concentrations mean? If NANOG is oligomerizing at 5 nM and regular cell expression is 70-80 nM, is the low dose below or near 5 nM while the high dose is 70-80 nM? It would be helpful to quantitatively characterize the dose dependency considering the authors observations.

Reviewer #3:

Remarks to the Author:

Choi et al characterize different domains of the human pluripotency factor NANOG using a combination of biochemical and structural assays. They conclude that NANOG is a disordered protein with an unstructured NTD and a prion-like CTD. Only the prion-like domain can form phase-separated condensates. Moreover, they show that full-length NANOG oligomerizes in cells and extracts, and it has the potential to bridge DNA elements using fEMSA and FRET assays.

While this study makes potentially interesting observations, it is somewhat difficult to ascertain their relevance and fit for a cell biology audience. For example, I would find it important to show at least some functional pluripotency assays using the mutants the authors generated, specifically the W8A version of NANOG (either via overexpression or knock-in in PSCs). Similarly, the impact of this study for a cell biology audience would be elevated if the authors validated some of their predictions using genomic assays such as Hi-C in cells expressing WT vs mutant NANOG. In the absence of such additional experiments, this manuscript may be a better candidate for a more specialized journal.

Specific comments:

1. The authors claim that NANOG's ability to oligomerize and phase-separate may explain its unique dose-sensitivity in PSCs. However, the authors also state that they have unpublished data on SOX2 and KLF4 undergoing phase separation, raising questions about specificity. I'd find it important to repeat at least some of the assays with a well-known pluripotency factor that does not form condensates or phase-separates, otherwise the specificity of this observation and its functional consequences remain unclear.
2. To be a contender for NCB, the authors should at least provide some basic pluripotency assays of the mutants they've generated, e.g. overexpression of WT vs W8A NANOG in PSCs under self-renewal vs differentiation conditions.
3. It remains unclear what impact the ability of NANOG to oligomerize, phase-separate and establish DNA-bridges has on chromatin structure in PSCs. The authors should consider performing Hi-C or minimally 3C assays for select genes in PSCs expressing WT and mutant NANOG to assess their effects on 3D chromatin architecture.

FINANCIAL AND NON-FINANCIAL COMPETING INTERESTS – the authors must include one of three declarations: (1) that they have no financial and non-financial competing interests; (2) that they have financial and non-financial competing interests; or (3) that they decline to respond, after the Author Contributions section. This statement will be published with the article, and in cases where financial and

non-financial competing interests are declared, these will be itemized in a web supplement to the article. For further details please see <https://www.nature.com/licenceforms/nrg/competing-interests.pdf>.

Methods should be written concisely, but should contain all elements necessary to allow interpretation and replication of the results. As a guideline, Methods sections typically do not exceed 3,000 words. The Methods should be divided into subsections listing reagents and techniques. When citing previous methods, accurate references should be provided and any alterations should be noted. Information must be provided about: antibody dilutions, company names, catalogue numbers and clone numbers for monoclonal antibodies; sequences of RNAi and cDNA probes/primers or company names and catalogue numbers if reagents are commercial; cell line names, sources and information on cell line identity and authentication. Animal studies and experiments involving human subjects must be reported in detail, identifying the committees approving the protocols. For studies involving human subjects/samples, a statement must be included confirming that informed consent was obtained. Statistical analyses and information on the reproducibility of experimental results should be provided in a section titled “Statistics and Reproducibility”.

All Nature Cell Biology manuscripts submitted on or after March 21 2016 must include a Data availability statement as a separate section after Methods but before references, under the heading “Data Availability”. For Springer Nature policies on data availability see <http://www.nature.com/authors/policies/availability.html>; for more information on this particular policy see <http://www.nature.com/authors/policies/data/data-availability-statements-data-citations.pdf>. The Data availability statement should include:

- Accession codes for primary datasets (generated during the study under consideration and designated as "primary accessions") and secondary datasets (published datasets reanalysed during the study under consideration, designated as "referenced accessions"). For primary accessions data should be made public to coincide with publication of the manuscript. A list of data types for which submission to community-endorsed public repositories is mandated (including sequence, structure, microarray, deep sequencing data) can be found here <http://www.nature.com/authors/policies/availability.html#data>.
- Unique identifiers (accession codes, DOIs or other unique persistent identifier) and hyperlinks for datasets deposited in an approved repository, but for which data deposition is not mandated (see here for details <http://www.nature.com/sdata/data-policies/repositories>).
- At a minimum, please include a statement confirming that all relevant data are available from the authors, and/or are included with the manuscript (e.g. as source data or supplementary information), listing which data are included (e.g. by figure panels and data types) and mentioning any restrictions on availability.
- If a dataset has a Digital Object Identifier (DOI) as its unique identifier, we strongly encourage including this in the Reference list and citing the dataset in the Methods.

We recommend that you upload the step-by-step protocols used in this manuscript to the Protocol Exchange. More details can found at www.nature.com/protocolexchange/about.

All imaging data should be accompanied by scale bars, which should be defined in the legend. Cropped images of gels/blots are acceptable, but need to be accompanied by size markers, and to retain visible background signal within the linear range (i.e. should not be saturated). The boundaries of panels with low background have to be demarked with black lines. Splicing of panels should only be considered if unavoidable, and must be clearly marked on the figure, and noted in the legend with a statement on whether the samples were obtained and processed simultaneously. Quantitative comparisons between samples on different gels/blots are discouraged; if this is unavoidable, it should only be performed for

samples derived from the same experiment with gels/blots were processed in parallel, which needs to be stated in the legend.

- For line art, graphs, charts and schematics we prefer Adobe Illustrator (.AI), Encapsulated PostScript (.EPS) or Portable Document Format (.PDF). Files should be saved or exported as such directly from the application in which they were made, to allow us to restyle them according to our journal house style.
- We accept PowerPoint (.PPT) files if they are fully editable. However, please refrain from adding PowerPoint graphical effects to objects, as this results in them outputting poor quality raster art. Text used for PowerPoint figures should be Helvetica (preferred) or Arial.
- We do not recommend using Adobe Photoshop for designing figures, but we can accept Photoshop generated (.PSD or .TIFF) files only if each element included in the figure (text, labels, pictures, graphs, arrows and scale bars) are on separate layers. All text should be editable in 'type layers' and line-art such as graphs and other simple schematics should be preserved and embedded within 'vector smart objects' - not flattened raster/bitmap graphics.
- Some programs can generate Postscript by 'printing to file' (found in the Print dialogue). If using an application not listed above, save the file in PostScript format or email our Art Editor, Allen Beattie for advice (a.beattie@nature.com).

The total number of Supplementary Figures (not including the “unprocessed scans” Supplementary Figure) should not exceed the number of main display items (figures and/or tables (see our Guide to Authors and March 2012 editorial <http://www.nature.com/ncb/authors/submit/index.html#suppinfo>;

<http://www.nature.com/ncb/journal/v14/n3/index.html#ed>). No restrictions apply to Supplementary Tables or Videos, but we advise authors to be selective in including supplemental data.

GUIDELINES FOR EXPERIMENTAL AND STATISTICAL REPORTING

REPORTING REQUIREMENTS – We are trying to improve the quality of methods and statistics reporting in our papers. To that end, we are now asking authors to complete a reporting summary that collects information on experimental design and reagents. The Reporting Summary can be found here <https://www.nature.com/documents/nr-reporting-summary.pdf> If you would like to reference the guidance text as you complete the template, please access these flattened versions at <http://www.nature.com/authors/policies/availability.html>.

We strongly recommend the presentation of source data for graphical and statistical analyses as a separate Supplementary Table, and request that source data for all independent repeats are provided when representative experiments of multiple independent repeats, or averages of two independent experiments are presented. This supplementary table should be in Excel format, with data for different

figures provided as different sheets within a single Excel file. It should be labelled and numbered as one of the supplementary tables, titled "Statistics Source Data", and mentioned in all relevant figure legends.

Author Rebuttal to Initial comments

Reviewers' Comments

Reviewer #1:

Remarks to the Author:

This is a very interesting, technical, biophysical and cell biology study that examines the biophysical properties of the master transcription factor NANOG. Choi et al. identify a possible role of the Trp-containing, prion-like CTD of NANOG in driving NANOG oligomerization and DNA bridging. NANOG expression levels are highly regulated in cells and this dose sensitivity potentially confers NANOG functionality in promoting ground state pluripotency. First, the authors provide compelling results regarding NANOG's ability to oligomerize. The authors use a battery of biophysical methods including smFRET, SEC, and crosslinking studies to probe the extent of oligomerization, in addition to mutants that are largely monomeric, and perform a side-by-side comparison of WT and mutant properties. They also find that NANOG and these variants oligomerize to similar extents in vivo by investigating several NANOG constructs (with different solubility tags and fusion proteins) in several cell lines. Second, the authors find that NANOG can 'bridge' DNA, and they investigated this using purified NANOG assemblies. This is an interesting study that proposes a link between NANOG oligomerization and DNA bridging that potentially confers functionality that is tunable based on NANOG expression level. However, the study could benefit from additional experiments to show that the DNA bridging phenomenon is occurring in vivo and that this is tied to how NANOG works. That's where I find the study to be lacking, particularly for Nature Cell Biology readership. The authors even have performed work in examining endogenous NANOG in human embryonic stem cells, but there's no in vivo data linking oligomerization or expression level control to NANOG function. I think it would markedly enhance the story if they could correlate expression level with DNA bridging potency, for example. I do want to reiterate that the work is high-quality and they have examined a difficult system that oligomerizes at very low concentrations (low nM) that makes it challenging to examine the molecular mechanism involved in self-assembly. Additionally, I have suggestions below on probing the role of DNA in NANOG oligomerization (along the lines of how nucleic acids contribute to formation of protein/nucleic acid granules in cells).

We appreciate the positive comments and are thankful for the suggestions. We have now performed many in vivo/in-cell studies (in collaboration with independent groups Wenbo Li, Chuangye Qi, Joo-Hyung Lee for ChIP-seq and Hi-C 3.0 experiments; Aleksander Bajic and Mahala Zahabiyon for the pluripotency assays) that further validate the roles of NANOG oligomerization in cells (Fig. 1 below; Fig.3j in revised paper), for pluripotency (Fig.2 below; Fig.4g-h and Supplementary Fig. 17 in revised paper), for specific DNA recognition (Fig.3 below; Fig. 5 and Supplementary Fig. 18 in revised paper) and for DNA bridging (Fig.4 below; Fig.6 and Supplementary Fig.19-20 in revised paper). Additionally, please also consider our responses to reviewers 2 and 3.

Figure 1. WT NANOG diffuses slower than the mutant W8A NANOG. a, Representative FRAP images at different timepoints for WT (top) and W8A mutant (bottom). b, FRAP curves for GFP-tagged WT (black) and W8A (red) overexpressed in HEK293T (top) and H9 ES cells (bottom). c, Corresponding calculated $\tau_{1/2}$ or recovery lifetimes.

Figure 2. W8A mutant NANOG induces ES cell differentiation. **a**, Fluorescence microscopy images of overexpressed GFP-tagged NANOG WT (left) and W8A mutant in H9 ESCs. The characteristic stem cell colonies are maintained in WT but not in the mutant. **b**, Fluorescence microscopy images (large image formed by stitching) of overexpressed GFP-tagged NANOG WT (left) and W8A mutant in ESCs to show widespread differentiation in the mutant. **c**, Crystal violet staining (stain all cells) of ESCs. **d**, Alkaline Phosphatase (AP) staining of ESC colonies with overexpressed GFP-tagged NANOG WT (left) and W8A mutant (right). There are more AP+ colonies with WT than mutant.

Figure 3. NANOG PrD mutations alter DNA recognition in cells. **a**, Venn diagram summarizing the numbers of NANOG ChIP-seq peaks observed for WT or W8A expressed in HEK 293T cells. The binding sites are classified into 3 groups: WT Only Sites, WT/W8A Shared Sites and W8A Only Sites. **b**, Heatmaps of the normalized ChIP-Seq reads of NANOG and H3K27ac on three groups (top to bottom): NANOG WT Only, WT/W8A Shared, W8A Only Sites, respectively. **c**, Top 3 non-redundant DNA motifs identified by HOMER from NANOG ChIP-Seq data for the 3 classified groups. P values generated by hypergeometric tests in HOMER. **d**, Cartoon diagram describing the possible modes (direct or indirect) of DNA recognition for the 3 groups.

Figure 4. NANOG PrD assembly mediates distant DNA-DNA contacts by Hi-C 3.0 analyses.

a, The P(s) curves showing probability of contact in relation to genomic distance in HEK 293T cells expressing either NANOG WT (red) or W8A (green). **b**, Contact heatmaps showing normalized interaction frequencies (20-kb bin) in one example region (chr7:15-30 Mb). **c**, Zoom-in view of a region (chr7:26-27.5Mb) that hosts TAD structures. **d**, Diagram explaining strategies in calculating pair-wise DNA contacts between NANOG binding sites (see Methods); adopted from a previous method, paired-end spatial chromatin analysis (PE-SCAN). Black oval objects indicate NANOG, and a and b indicate two distant genomic bins (25kb) harboring NANOG binding sites. Sliding windows of 25 kb were used to scan each site of a or b for 250 kb, and the interactions between each sliding window bin next to a or b were calculated as the background interactions surrounding specific a-b interactions. For any NANOG binding sites (a1 - an vs. b1 - bn), the aggregated interactions between each pair of sites and nearby background were shown below. **e**, DNA contact strength in two cell conditions (expressing WT or W8A) that stem from NANOG WT Only sites (top row), WT/W8A Shared sites (middle row), and W8A Only Sites (bottom row). (i), Data plots based on PE-SCAN method. (ii), Box plots showing quantitative counts of the central peaks shown in (i). The boxplot center lines represent medians; box limits indicate the 25th and 75th percentiles; and whiskers extend 1.5 times the interquartile range (IQR) from the 25th and 75th percentiles; P-values were based on paired students' T-tests. (iii), Cartoon diagrams based on DNA contacts.

In addition to the above comments, I have the following concerns:

1) The authors mention that NANOG CTD forms gel-like condensates – the study doesn't currently probe their liquidity and that would be recommended if the 'condensates' terminology is to be used to describe their morphology.

Yes, the gel-like condensates by NANOG CTD are based on visual inspection. The literature is evolving, and our understanding is that condensates can also refer to varying degrees of phase transitions, either by liquid-liquid phase separation (LLPS), liquid-to-solid phase transitions, and gel-like behaviors^{1, 2}. In a recent primer and guidelines on condensates by Alberti et al.³, the authors stated that "Liquids, Solids, and Gels Can All Emerge from LLPS". NANOG CTD would be more of liquid-to-solid phase transition. We could manipulate shorter NANOG WR peptides into variable material states from more solid to liquid-like behavior (Ferreon, et al, manuscript in preparation) but this is beyond the scope of the current paper.

2) Extensive work was done in characterizing NANOG from cell lysates, including crosslinking and fluorescence fluctuation spectroscopy. I'm interested in knowing about NANOG's propensity to form puncta in cells. Curiously, Extended Data Fig 9 shows a diffuse distribution of h6f-NANOG WT-eGFP in cell nuclei. Is this not surprising given the propensity for NANOG WT to oligomerize particularly at the estimated concentrations? On a similar note, how does endogenous NANOG stain?

NANOG is toxic to HEK 293T cells when expressed at high concentration; cells that survive after selection usually have low NANOG expression levels (Supplementary Fig. 9 in revised paper). We do observe rare cells that have higher NANOG expression and show puncta (Figure 5 below; Supplementary Fig. 9g in revised paper). Puncta/droplet formation most likely comprise of hundreds to thousands of molecules, as quantitatively determined for KLF4 (Surface condensation of a pioneer transcription factor on DNA | bioRxiv). In another example, Zhang et al⁴ have shown that HoTAG oligomer formation (up to hexamer tested) has diffuse distribution in cells. For KLF4, we observe puncta formation at >500-700 nM and droplet formation at >1.5-2 μ M. Based on our western blot and GFP calibration, NANOG expression in individual cells falls well below this limit (~150 nM; Supplementary Fig. 9). H9 ESC cells (primed pluripotent cells) are also known to have lower amounts of NANOG (~80-160 nM, Supplementary Fig. 9) and not equivalent to naïve pluripotent stem cells as mouse ESCs. NANOG behave more like nanocondensates^{5,6}, rather than typical liquid condensates at mesoscale level^{7,8}.

Figure 5. NANOG forms puncta at higher expression levels.

Fluorescence microscopy of rare (~1 in 1000) HEK293T cells with GFP-NANOG at higher expression levels as compared to surrounding cells with lower NANOG expression (~150 nM).

3) Related to the oligomer sizes reported, have the authors performed DLS (dynamic light scattering) studies to look at particle size of NANOG oligomers?

Yes, we did attempt SEC-MALS (Size-Exclusion Chromatography-Multi-Angle Light Scattering), however with the relatively high concentration necessary for the MW determination (>0.1 mg/ml; μ M range), both full-length MBP-WT and MBP-W8A mutant showed large particles, aggregates in the void volume that obscures the monomeric peaks and hindered accurate MW determination (data not shown). Hence, we instead employed fluorescence-based techniques (at nM concentrations) to determine oligomeric sizes.

4) Have the authors tried to mix the NANOG CTD with DNA? Part of this experiment would address a question as to whether the prion-like CTD also interacts with DNA directly. The authors should try incubating the CTD with DNA and attempt a refolding experiment to see if CTD aggregation propensity is altered by DNA; this would provide additional evidence that DNA could be integral to how NANOG oligomerizes.

We tested NANOG CTD:DNA interaction by EMSA gel shift and ThT aggregation assay (Figure 6 below; Supplementary Fig. 11 in revised paper). Our data show that NANOG CTD does not significantly interact specifically with DNA of up to 20μ M, well above that necessary for binding of FL or DBD domain (nM affinity). We do observe a slight increase in CTD aggregation propensity, likely due to non-specific interactions with DNA.

Figure 6. NANOG CTD does not specifically interact with DNA. a. EMSA of NANOG CTD (0-20 μM) with 1 μM GATA6-DNA. b. CTD (3.4 μM) aggregation kinetics monitored by ThT fluorescence with 1 μM GATA6-DNA (n=3).

5) The authors use fluorescently-labeled DNA on the 5' end for their critical smFRET diffusion experiments to demonstrate DNA bridging in Figure 4. To corroborate their data, could the authors also label one of the DNA molecules with a 3' fluorescent probe to provide another set of experimental data that could provide additional distance constraints on the proximity of the two labeled DNA molecules? Another suggestion could be to use two entirely different DNA sequences (each with different dyes) that could mimic what is happening in the cell.

Accurate quantitation of distance restraints requires more intensive calibration of distances that is beyond the scope of this paper. However, we performed the experiment to test DNA bridging using two different DNAs (i.e., Oct4 DNA and Gata6-DNA) (Figure 7 below; Supplementary Fig. 17 in revised paper).

Figure 7. WT NANOG but not the W8A mutant bridges OCT4 and GATA6 dsDNA. a. Auto FCCS curves of OCT4-AF647 (black squares), GATA6-AF488 (red circles) and cross-correlation curve (blue triangles) in the presence of WT NANOG (250 nM) (n=8). b. Auto FCCS curves of OCT4-AF647 (black squares), GATA6-AF488 (red circles) and cross-correlation curve (blue triangles) in the presence of mutant W8A NANOG (250 nM) (n=8).

6) A suggestion – as the authors show that W8A is monomeric, could the authors make a W8A CTD, label with ^{15}N , and collect a NMR spectrum to compare against the WT CTD in Extended Data? This would further demonstrate that the severe peak broadening in WT CTD is a result of oligomerization.

The W8A does not aggregate at relatively low concentrations in our experiments ($<1 \mu\text{M}$). At higher concentrations, even the intermediate CTD mutants (W1357A, W468A) aggregate, making

it difficult to obtain good NMR spectra. This is because the WR domain sequence (even with Trp to Ala mutations), contain repeats of Asn, Gln and polar residues (Supplementary Fig. 1) that have strong propensities for amyloid formation⁹. We have tested the aggregation propensities of shorter WR peptides and observed that the Trp to Ala mutant still aggregates but on a longer time scale (manuscript in preparation).

7) Trp to Ala mutations are substantial. Are there other mutations that could be made that create a NANOG mutant that is intermediate in oligomerization behavior between NANOG W8A and NANOG WT?

We prepared intermediate mutants with only some of the possible Trp to Ala mutations (either only 3 or 4 of the Trp to Ala mutations, W1357A and W468A, respectively) and observed in-between oligomerization behavior by CD spectroscopy (Fig. 2). To probe the effect in cells, we have to work with the full W8A mutant for more straightforward data interpretations. As mentioned in comment #6 (above), even with full 8 Trp to Ala mutations, the W8A mutant can still aggregate significantly at higher concentrations ($>1 \mu\text{M}$), preventing accurate MW determination by SEC-MALS (comment #3).

I found the manuscript to be data-rich and very concise.

However, there are minor errors:

Line 73 – should be extended data figure 2?

Line 85 – should be Figure 2?

Line 89 – Figure reference correct?

Figure 4c – y-axis needs a label

Extended Data Figure 4 – need MW markers on gels in panel a and b at least

Thank you for the corrections. We have addressed all the minor errors mentioned.

Reviewer #2:

Remarks to the Author:

In the manuscript entitled “NANOG prion-like assembly mediates DNA bridging”, Choi and colleagues investigate the ability of NANOG to form higher ordered structures using a combination of in vitro biochemistry and cell biology. The authors show that full-length NANOG forms higher order oligomers at extremely low concentrations and posit that this oligomerization enables NANOG to bridge DNA during the formation of DNA condensates.

The data for NANOG oligomerization presented in this manuscript is clear and convincing. In a couple of instances that this reviewer describes below, the authors seem to be in prime position to extend the study a bit further to continue to unravel the biophysical mechanisms that regulate NANOGs cellular functions. Additionally, the authors promote a dose-sensitive mechanism of NANOG function in cells in both their abstract, intro, and conclusion, but don't explicitly tie their results to this mechanism. A deeper discussion of how their data relates to dose sensitivity is necessary. If this can be addressed in the text, this paper is a strong candidate for publication in Nature Cell Biology.

Comments:

1) The authors perform well-controlled biochemical experiments to investigate the mechanism underlying cellular crosslinking experiments shown in Figure 3. While the results of these experiments are convincing and suggest that NANOG oligomerization can indeed account for the observed band shifts in the gel, the cellular environment is far more complex than in in vitro experiments. It would be interesting to know if the cellular complexes include specific binding partners or if they are mostly NANOG. If experimentally possible, NANOG pulldown followed by mass spectrometry analysis may be able to parse the composition of the complexes and provide additional insight into NANOG oligomer interactions in cells.

We appreciate the positive comments. Please also read our response to Reviewers 1 and 3. We have been trying to answer such questions for the past few years. What is stronger, NANOG homo-oligomerization or hetero-oligomerization with partners? We have tried multiple times to obtain mass spectrometry crosslinking data of NANOG overexpressed in HEK293T cells but this approach has so far been challenging because human NANOG is not efficiently “pulled down” because of its tendency to aggregate. Mouse Nanog have been shown to have strong hetero-oligomerization interaction with SOX2, with the tryptophans being key to the interaction¹⁰. However, we failed to observe the same strong interaction of Sox2 with human NANOG (Figure 8). Currently, we are developing alternative methods where we have co-expressed different fluorescence tagged NANOG and binding partners in Sf9 insect cells (co-expression in mammalian cells have failed so far). Our preliminary FCCS data in cell lysates shows that NANOG homo-oligomerization is much stronger than hetero-oligomerization with suggested binding partners¹⁰⁻¹⁵. More detailed quantitative analysis, optimization, and experimentation of different NANOG and partner concentrations are necessary to accurately determine the strength and

specificity of interactions, but these studies are beyond the scope of this paper. We did mention in the text that the oligomerization domain might also have a significant contribution in other protein-protein interactions and in NANOG's multivalent hub assemblies.

Figure 8. NANOG homo-oligomerization is stronger than hetero-oligomerization. a, SDS-PAGE gel showing pull-down efficiency of h6g-SOX2 (bound to IgG Sepharose beads) with MBP-NANOG WT or W8A mutant. Other distinct bands (at ~25 and ~55 kDa) represent IgG proteins. b, Quantification of pull-down efficiency (based on gel band intensities between MBP-NANOG WT and W8A mutant with h6g-SOX2). c, Auto and Cross-Correlation FCCS measurements of NANOG and putative binding partners. d, Comparison between the cross-correlation curves of NANOG:NANOG vs NANOG and other putative binding partners.

2) In the fSEC chromatograms, GFP-NANOG appears to be eluted over multiple peaks, not just in the void. The right-most peak is slightly shifted when compared with GFP-NANOG W8A or GFP alone, suggesting that this may be some degradation product or that NANOG interacts with the fused GFP. Were the contents of this peak analyzed? If so, is the protein in this peak identifiable? If the protein in this peak is GFP-NANOG, does this suggest that the fusion of GFP to NANOG destabilizes the higher order complexes that are observed with other versions of NANOG? A comment from the authors would be helpful to properly understand the data.

It is difficult to determine whether the peak in question is some monomeric form of WT NANOG or truncated versions of GFP-tagged NANOG. We attempted to characterize the fractions and ran them in SDS-PAGE gels, but either the concentrations were too low even for fluorescence detection or the samples were non-specifically bound to the column/tube surfaces after peak elution. We also observed that the truncated versions and cleaved h6GeGFP passes through the SEC column more readily than the full-length versions.

3) The FFS and FCS data provides convincing evidence that WT NANOG forms higher-order oligomers. Is it possible to also run DLS on WT- and W8A-NANOG in vitro to determine whether these oligomers are mono- or poly-dispersed. This measurement would indicate whether WT NANOG forms a single oligomeric species or oligomers of random size. This type of data would also provide insight into potential cellular mechanisms that are described in the authors' model in Figure 4G.

Please see response to Reviewer 1 comment #3. Based on our fluorescence PCH data (Fig. 3), the oligomer are poly-disperse and exhibit variable oligomeric sizes, consistent with amyloid-like aggregation behavior.

4) In lines 180-181, the authors posit that NANOG forms oligomers in cells. The authors seem to be in prime position to test this. To test this, mEos2-NANOG WT or W8A could be expressed in cells, a portion of the expressed protein could be photoconverted, and molecules tracked using single molecule fluorescence microscopy. While not necessary for their conclusions, this experiment would offer experimental insight into cellular oligomerization of WT-NANOG.

We don't have the mEos2-based technique standardized or optimized in our laboratory. We instead carried out the alternative experiment of using standard FRAP technique to test for oligomerization in cells. Pls. see Figure 1 and response to Reviewer 1. Our data shows that overexpressed GFP-tagged WT NANOG diffuses much slower than the mutant and the fluorescence recovery lifetimes of WT is significantly longer than W8A mutant (tested both in HEK 293T and H9 ES cells).

5) In the text in the top paragraph on page 3, the authors refer to Figures 1C, 1D, and 1E. These should be Figures 2C, 2D, and 2E.

Thank you. We have edited these.

6) Low and high levels are mentioned to describe this dose-dependency. It would be helpful for the authors to discuss what these dosages or concentrations mean? If NANOG is oligomerizing at 5 nM and regular cell expression is 70-80 nM, is the low dose below or near 5 nM while the high dose is 70-80 nM? It would be helpful to quantitatively characterize the dose dependency considering the authors observations.

Indeed, these are questions we eventually want to answer. Currently, even at low 5 nM, we observe oligomerization of FL-NANOG (oligomers composed of 5 or more monomeric units, Fig. 3). We are unsure whether the resolution of current Hi-C and other in-cells functional assays can distinguish between 5 nM and 70 nM. Meanwhile, we do know from the literature that amyloid aggregation is a dose-dependent event. Higher concentration leads to faster assembly kinetics with heterogeneous high MW oligomeric sizes. We have also shown this in the in vitro EMSA assay (Figure 4) that higher NANOG concentrations (>60 nM) results in greater population of high MW complexes (bands in the wells). It would be interesting to characterize NANOG concentrations of the primed stem cells versus naïve stem cells¹⁶, monoallelic states and bi-allelic states of NANOG¹⁷ to truly understand the mechanism and why dosage is very important to NANOG function. These are difficult experiments (not within our current expertise nor scope of the paper). Furthermore, to date, most literature studies are performed on mouse ESCs, rather than human ESCs.

Reviewer #3:

Remarks to the Author:

Choi et al characterize different domains of the human pluripotency factor NANOG using a combination of biochemical and structural assays. They conclude that NANOG is a disordered protein with an unstructured NTD and a prion-like CTD. Only the prion-like domain can form phase-separated condensates. Moreover, they show that full-length NANOG oligomerizes in cells and extracts, and it has the potential to bridge DNA elements using fEMSA and FRET assays.

While this study makes potentially interesting observations, it is somewhat difficult to ascertain their relevance and fit for a cell biology audience. For example, I would find it important to show at least some functional pluripotency assays using the mutants the authors generated, specifically the W8A version of NANOG (either via overexpression or knock-in in PSCs). Similarly, the impact of this study for a cell biology audience would be elevated if the authors validated some of their predictions using genomic assays such as Hi-C in cells expressing WT vs mutant NANOG. In the absence of such additional experiments, this manuscript may be a better candidate for a more specialized journal.

We appreciate the comments and suggestions. In response, we have included Hi-C and pluripotency assays to expand our cell biology experiments (Figures 1-4 above) Lastly, please also review our responses to Reviewers 1 and 2.

Specific comments:

1. The authors claim that NANOG's ability to oligomerize and phase-separate may explain its unique dose-sensitivity in PSCs. However, the authors also state that they have unpublished data on SOX2 and KLF4 undergoing phase separation, raising questions about specificity. I'd find it important to repeat at least some of the assays with a well-known pluripotency factor that does not form condensates or phase-separates, otherwise the specificity of this observation and its functional consequences remain unclear.

We would like to emphasize the distinctions among the different protein systems. We have demonstrated that KLF4 undergoes liquid-liquid phase separation (LLPS) readily in vitro and in cells, due to the multivalency of the zinc fingers DNA binding domain and KLF4 recognition of the partial motifs of its cognate sequence¹⁸. In addition, the condensates observed in cells and in vitro are at high nM to μ M concentrations. At similar concentrations in cells where KLF4 forms distinct puncta or droplet condensates, OCT4 and SOX2 do not undergo LLPS in cells. It is important to note that these KLF4 condensates are at the micro-mesoscale level (μ m range). Our unpublished observations of SOX2 LLPS in vitro was only induced at high nM to μ M concentrations and in the presence of crowding agents. Boija et al ⁶ also showed SOX2-GFP LLPS at 40 μ M with 10% PEG-8K. Yes, we agree that future studies should resolve the relevance of every observed

condensates. Sabari et al⁵ show that the mediator coactivator undergoes condensation in cells, and these are more in the nanoscale level (i.e., nanocondensates). We cannot directly compare NANOG's with KLF4's condensates in cells because NANOG cannot be overexpressed at high concentration levels. Moreover, NANOG's phase transition behavior is different from that of KLF4. First, NANOG aggregation/condensation is due to the oligomerization domain and not the DNA-binding domains/DNA recognition (heterotypic assembly) by KLF4. Second, NANOG readily oligomerizes at very low nM concentrations (~5 nM) on its own (homotypic assembly). At higher μM concentrations, purified NANOG (especially C-terminal fragments) readily aggregates into more solid precipitates (indicative of liquid to solid phase transition). As pointed out by Alberti et al.³, liquids, solids, and gels can arise from LLPS, depending on protein sequence and material properties. Despite via different mechanisms, the role of KLF4 and NANOG condensation might have overlapping functions. We showed that they might be relevant for chromatin looping and the establishment of pluripotency contacts. Future studies are needed to dissect their distinct contributions. It is possible that KLF4 condensation is more critical in *early* stages of *induced* reprogramming where it facilitates chromatin opening and OCT4 and SOX2 cooperative recruitment while NANOG oligomerization or assembly is necessary for the *later* stages in reprogramming where essential stable interactions are necessary to achieve pluripotency. These hypotheses are also consistent with KLF4 playing an early critical role in activating NANOG expression^{15, 19} as well as other literature studies on the distinct roles of KLF4 and NANOG in different stages of reprogramming^{20, 21}. We have added some of these discussions in the manuscript text.

2. To be a contender for NCB, the authors should at least provide some basic pluripotency assays of the mutants they've generated, e.g. overexpression of WT vs W8A NANOG in PSCs under self-renewal vs differentiation conditions.

Thank you for the suggestions. We have now carried out pluripotency assays (see Figure 2 above). Overexpression of NANOG W8A mutant results in dramatic differentiation of H9 ES cells. In contrast, H9 ES cells overexpressing WT maintain the round ES colony morphology and display more AP+ colonies.

3. It remains unclear what impact the ability of NANOG to oligomerize, phase-separate and establish DNA-bridges has on chromatin structure in PSCs. The authors should consider performing Hi-C or minimally 3C assays for select genes in PSCs expressing WT and mutant NANOG to assess their effects on 3D chromatin architecture.

Thanks for the suggestions. We have made significant efforts in performing Hi-C and ChIP-seq assays (see Figures 3-4 above; Figures 5-6 in revised paper; additional supplementary figures 18-20 were also added) and revised the paper to incorporate these new findings. We performed these experiments in HEK 293T cells (without endogenous NANOG expression) that overexpressed GFP-tagged NANOG WT or W8A mutant. We avoided human ESCs because of

complications that can arise from endogenous NANOG and changes in the chromatin architecture that are indirectly linked to NANOG WT/W8A overexpression and/or cell identity changes. To directly investigate effects in human ESCs would require studies similar to that of de Wit et al²², which took advantage of established mouse ESC cell lines with endogenous NANOG knockout, as well as many literature studies on mouse ESCs. Such established cell lines are yet to be developed for human ESCs especially because in vitro hESCs are mostly primed stem cells and not naïve stem cells¹⁶. Generating hESCs similar to mESCs is still currently challenging and not yet standardized. Regardless, our new results using Hi-C 3.0 and ChIP-seq strongly supported that NANOG promotes DNA contacts in cells, and this function is dependent on the C-terminal PrD domain (as W8A inhibited such roles).

1. Kato, M. *et al.* Cell-free formation of RNA granules: low complexity sequence domains form dynamic fibers within hydrogels. *Cell* **149**, 753-767 (2012).
2. Patel, A. *et al.* A Liquid-to-Solid Phase Transition of the ALS Protein FUS Accelerated by Disease Mutation. *Cell* **162**, 1066-1077 (2015).
3. Alberti, S., Gladfelter, A. & Mittag, T. Considerations and Challenges in Studying Liquid-Liquid Phase Separation and Biomolecular Condensates. *Cell* **176**, 419-434 (2019).
4. Zhang, Q. *et al.* Visualizing Dynamics of Cell Signaling In Vivo with a Phase Separation-Based Kinase Reporter. *Mol Cell* **69**, 334-346 e334 (2018).
5. Sabari, B.R. *et al.* Coactivator condensation at super-enhancers links phase separation and gene control. *Science* **361** (2018).
6. Boija, A. *et al.* Transcription Factors Activate Genes through the Phase-Separation Capacity of Their Activation Domains. *Cell* **175**, 1842-1855 e1816 (2018).
7. Maharana, S. *et al.* RNA buffers the phase separation behavior of prion-like RNA binding proteins. *Science* **360**, 918-921 (2018).
8. Banani, S.F., Lee, H.O., Hyman, A.A. & Rosen, M.K. Biomolecular condensates: organizers of cellular biochemistry. *Nature reviews. Molecular cell biology* **18**, 285-298 (2017).
9. Wang, J. *et al.* A Molecular Grammar Governing the Driving Forces for Phase Separation of Prion-like RNA Binding Proteins. *Cell* **174**, 688-699 e616 (2018).
10. Gagliardi, A. *et al.* A direct physical interaction between Nanog and Sox2 regulates embryonic stem cell self-renewal. *EMBO J* **32**, 2231-2247 (2013).
11. Fidalgo, M. *et al.* Zfp281 mediates Nanog autorepression through recruitment of the NuRD complex and inhibits somatic cell reprogramming. *Proc Natl Acad Sci U S A* **109**, 16202-16207 (2012).
12. Liang, J. *et al.* Nanog and Oct4 associate with unique transcriptional repression complexes in embryonic stem cells. *Nat Cell Biol* **10**, 731-739 (2008).
13. Wang, J. *et al.* A protein interaction network for pluripotency of embryonic stem cells. *Nature* **444**, 364-368 (2006).
14. Zbinden, M. *et al.* NANOG regulates glioma stem cells and is essential in vivo acting in a cross-functional network with GLI1 and p53. *EMBO J* **29**, 2659-2674 (2010).
15. Apostolou, E. *et al.* Genome-wide chromatin interactions of the Nanog locus in pluripotency, differentiation, and reprogramming. *Cell Stem Cell* **12**, 699-712 (2013).

16. Takashima, Y. *et al.* Resetting transcription factor control circuitry toward ground-state pluripotency in human. *Cell* **158**, 1254-1269 (2014).
17. Miyanari, Y. & Torres-Padilla, M.E. Control of ground-state pluripotency by allelic regulation of Nanog. *Nature* **483**, 470-473 (2012).
18. Sharma, R. *et al.* Liquid condensation of reprogramming factor KLF4 with DNA provides a mechanism for chromatin organization. *Nature communications* **12**, 5579 (2021).
19. Chan, K.K. *et al.* KLF4 and PBX1 directly regulate NANOG expression in human embryonic stem cells. *Stem Cells* **27**, 2114-2125 (2009).
20. Minkovsky, A., Patel, S. & Plath, K. Concise review: Pluripotency and the transcriptional inactivation of the female Mammalian X chromosome. *Stem Cells* **30**, 48-54 (2012).
21. Chronis, C. *et al.* Cooperative Binding of Transcription Factors Orchestrates Reprogramming. *Cell* **168**, 442-459 e420 (2017).
22. de Wit, E. *et al.* The pluripotent genome in three dimensions is shaped around pluripotency factors. *Nature* **501**, 227-231 (2013).

Decision Letter, first revision:

Subject: Your manuscript, NCB-F46094A
Message: Our ref: NCB-F46094A

18th January 2022

Dear Dr. Ferreon,

Thank you for submitting your revised manuscript "NANOG prion-like assembly mediates DNA bridging" (NCB-F46094A). It has now been seen by the original referees and their comments are below. The reviewers find that the paper has improved in revision, and therefore we'll be happy in principle to publish it in Nature Cell Biology, pending minor revisions to satisfy the referees' final requests and to comply with our editorial and formatting guidelines.

Thank you again for your interest in Nature Cell Biology. Please do not hesitate to contact me if you have any questions.

Sincerely,

Jie Wang, PhD
Senior Editor
Nature Cell Biology

Tel: +44 (0) 207 843 4924
email: jie.wang@nature.com

Reviewer #1 (Remarks to the Author):

The authors have made extensive revisions that have significantly strengthened this already-strong paper, and revealed interesting connections and correlations between NANOG's oligomerization propensity and NANOG's ability to bridge DNA and affect pluripotency. The manuscript is a tour de force having used a wide breadth of state-of-the-art biophysical and cell biology techniques to examine the structure/function link of oligomerization to NANOG functionality. I also appreciate the author's careful wording when discussing phase transitions and different types of systems. In many ways, this work carves out a unique niche in how to investigate difficult oligomerizing protein systems (of which there are many, and these are underexplored but of intense interest given their propensity to include prion-like or other low-complexity regions). The use of fluorescence fluctuation and smFRET techniques allowed the authors to probe low nM-based protein oligomerization. In addition, the authors performed Hi-C experiments in cells to measure pairwise contact changes in genomic loci in the presence of either WT or W8A NANOG. These latter experiments suggest a link between the Trp-containing C-terminal oligomerization domain and NANOG functionality in bridging DNA. There are still MANY interesting questions for followup, particularly in relationship to other transcription factors and interaction partners with NANOG, but these are outside the scope of the current work. I now believe the manuscript is ready for publication in Nature Cell Biology.

Minor question/correction:

Figure 4 - Could the authors clarify what does it mean that the residuals don't line up in the FCCS traces, e.g. DNA-AF488 in panel (iii) of Figure 4e?

Reviewer #2 (Remarks to the Author):

The authors have adequately addressed the concerns of this reviewer and this manuscript is a strong candidate for publication in Nature Cell Biology. The additional experiments and data included in the manuscript enhance the quality and depth of this study and provide an excellent foundation upon which future studies can be built.

Minor comments:

1) In extended data Figure 12, is it possible to alter the color of the donor and acceptor channel traces and the fluorophores in the model to green and magenta? The same goes for the traces in extended data Figures 13 – 15. The red and green will appear as the same color for red/green color blind readers.

Reviewer #3 (Remarks to the Author):

The authors have made a significant effort to assess the functional roles of the W8A mutant in ESC biology, uncovering an intriguing differentiation phenotype. In addition, they determined the consequences of the mutants on genome-wide DNA binding (ChIP-Seq) and 3D chromatin architecture (3C), showing differences in binding preferences and looping strength. I am satisfied with the authors' responses and revisions and recommend publication.

Decision letter, final requests:

Subject: NCB: Your manuscript, NCB-F46094A
Message: Our ref: NCB-F46094A

13th February 2022

Dear Dr. Ferreon,

Thank you for your patience as we've prepared the guidelines for final submission of your Nature Cell Biology manuscript, "NANOG prion-like assembly mediates DNA bridging" (NCB-F46094A). Please carefully follow the step-by-step instructions provided in the attached file, and add a response in each row of the table to indicate the changes that you have made. Ensuring that each point is addressed will help to ensure that your revised manuscript can be swiftly handed over to our production team.

We would like to start working on your revised paper, with all of the requested files and forms, as soon as possible (preferably within one week). Please get in contact with us if you anticipate delays.

If you have not done so already, please alert us to any related manuscripts from your group that are under consideration or in press at other journals, or are being written up for submission to other

journals (see: <https://www.nature.com/nature-research/editorial-policies/plagiarism#policy-on-duplicate-publication> for details).

In recognition of the time and expertise our reviewers provide to Nature Cell Biology's editorial process, we would like to formally acknowledge their contribution to the external peer review of your manuscript entitled "NANOG prion-like assembly mediates DNA bridging". For those reviewers who give their assent, we will be publishing their names alongside the published article.

Nature Cell Biology offers a Transparent Peer Review option for new original research manuscripts submitted after December 1st, 2019. As part of this initiative, we encourage our authors to support increased transparency into the peer review process by agreeing to have the reviewer comments, author rebuttal letters, and editorial decision letters published as a Supplementary item. When you submit your final files please clearly state in your cover letter whether or not you would like to participate in this initiative. Please note that failure to state your preference will result in delays in accepting your manuscript for publication.

Cover suggestions

As you prepare your final files we encourage you to consider whether you have any images or illustrations that may be appropriate for use on the cover of Nature Cell Biology.

Nature Cell Biology has now transitioned to a unified Rights Collection system which will allow our Author Services team to quickly and easily collect the rights and permissions required to publish your work. Approximately 10 days after your paper is formally accepted, you will receive an email in providing you with a link to complete the grant of rights. If your paper is eligible for Open Access, our

Author Services team will also be in touch regarding any additional information that may be required to arrange payment for your article.

Please note that Nature Cell Biology is a Transformative Journal (TJ). Authors may publish their research with us through the traditional subscription access route or make their paper immediately open access through payment of an article-processing charge (APC). Authors will not be required to make a final decision about access to their article until it has been accepted. Find out more about Transformative Journals

Authors may need to take specific actions to achieve compliance with funder and institutional open access mandates. For submissions from January 2021, if your research is supported by a funder that requires immediate open access (e.g. according to Plan S principles) then you should select the gold OA route, and we will direct you to the compliant route where possible. For authors selecting the subscription publication route our standard licensing terms will need to be accepted, including our self-archiving policies. Those standard licensing terms will supersede any other terms that the author or any third party may assert apply to any version of the manuscript.

For information regarding our different publishing models please see our Transformative Journals page. If you have any questions about costs, Open Access requirements, or our legal forms, please contact ASJournals@springernature.com.

[REDACTED]

If you have any further questions, please feel free to contact us. Many thanks!

Best regards,

Ziqian Li
Editorial Assistant
Nature Cell Biology

On behalf of

Jie Wang, PhD
Senior Editor
Nature Cell Biology

Tel: +44 (0) 207 843 4924
email: jie.wang@nature.com

Reviewer #1:

Remarks to the Author:

The authors have made extensive revisions that have significantly strengthened this already-strong paper, and revealed interesting connections and correlations between NANOG's oligomerization propensity and NANOG's ability to bridge DNA and affect pluripotency. The manuscript is a tour de force having used a wide breadth of state-of-the-art biophysical and cell biology techniques to examine the structure/function link of oligomerization to NANOG functionality. I also appreciate the author's careful wording when discussing phase transitions and different types of systems. In many ways, this work carves out a unique niche in how to investigate difficult oligomerizing protein systems (of which there are many, and these are underexplored but of intense interest given their propensity to include prion-like or other low-complexity regions). The use of fluorescence fluctuation and smFRET techniques allowed the authors to probe low nM-based protein oligomerization. In addition, the authors performed Hi-C experiments in cells to measure pairwise contact changes in genomic loci in the presence of either WT or W8A NANOG. These latter experiments suggest a link between the Trp-containing C-terminal oligomerization domain and NANOG functionality in bridging DNA. There are still MANY interesting questions for followup, particularly in relationship to other transcription factors and interaction partners with NANOG, but these are outside the scope of the current work. I now believe the manuscript is ready for publication in Nature Cell Biology.

Minor question/correction:

Figure 4 - Could the authors clarify what does it mean that the residuals don't line up in the FCCS traces, e.g. DNA-AF488 in panel (iii) of Figure 4e?

Reviewer #2:

Remarks to the Author:

The authors have adequately addressed the concerns of this reviewer and this manuscript is a strong candidate for publication in Nature Cell Biology. The additional experiments and data included in the manuscript enhance the quality and depth of this study and provide an excellent foundation upon which future studies can be built.

Minor comments:

1) In extended data Figure 12, is it possible to alter the color of the donor and acceptor channel traces and the fluorophores in the model to green and magenta? The same goes for the traces in extended data Figures 13 – 15. The red and green will appear as the same color for red/green color blind readers.

Reviewer #3:

Remarks to the Author:

The authors have made a significant effort to assess the functional roles of the W8A mutant in ESC biology, uncovering an intriguing differentiation phenotype. In addition, they determined the consequences of the mutants on genome-wide DNA binding (ChIP-Seq) and 3D chromatin architecture (3C), showing differences in binding preferences and looping strength. I am satisfied with the authors' responses and revisions and recommend publication.

Author Rebuttal, first revision:

Reviewers' Comments

We thank the reviewers for their great comments. Pls. see our comments below.

Reviewer #1 (Remarks to the Author):

The authors have made extensive revisions that have significantly strengthened this already-strong paper, and revealed interesting connections and correlations between NANOG's oligomerization propensity and NANOG's ability to bridge DNA and affect pluripotency. The manuscript is a tour de force having used a wide breadth of state-of-the-art biophysical and cell biology techniques to examine the structure/function link of oligomerization to NANOG functionality. I also appreciate the author's careful wording when discussing phase transitions and different types of systems. In many ways, this work carves out a unique niche in how to investigate difficult oligomerizing protein systems (of which there are many, and these are underexplored but of intense interest given their propensity to include prion-like or other low-complexity regions). The use of fluorescence fluctuation and smFRET techniques allowed the authors to probe low nM-based protein oligomerization. In addition, the authors performed Hi-C experiments in cells to measure pairwise contact changes in genomic loci in the presence of either WT or W8A NANOG. These latter experiments suggest a link between the Trp-containing C-terminal oligomerization domain and NANOG functionality in bridging DNA. There are still MANY interesting questions for followup, particularly in relationship to other transcription factors and interaction partners with NANOG, but these are outside the scope of the current work. I now believe the manuscript is ready for publication in Nature Cell Biology.

Minor question/correction:

Figure 4 - Could the authors clarify what does it mean that the residuals don't line up in the FCCS traces, e.g. DNA-AF488 in panel (iii) of Figure 4e?

We could fit the data with less deviation in DNA-AF488 auto-correlation curve if we use parameters of more species (multiple diffusion times and concentrations) or of including a triplet state model only for that donor channel, but we didn't want to overparameterize (Occam's razor) and we want the model that best fit *all* data. The deviation could be caused by triplet state fast fluctuations contribution of the AF488 dye and also if the NANOG oligomers are heterogeneous (characterized by more than one diffusion coefficient, which is most likely the case). Regardless, this shouldn't affect the observed NANOG-mediated cross-correlation of DNA-AF488/DNA-AF647.

Reviewer #2 (Remarks to the Author):

The authors have adequately addressed the concerns of this reviewer and this manuscript is a strong candidate for publication in Nature Cell Biology. The additional experiments and data included in the manuscript enhance the quality and depth of this study and provide an excellent foundation upon which future studies can be built.

Minor comments:

1) In extended data Figure 12, is it possible to alter the color of the donor and acceptor channel traces and the fluorophores in the model to green and magenta? The same goes for the traces in extended data Figures 13 – 15. The red and green will appear as the same color for red/green color blind readers.

We thank the reviewer for making us aware of this. We have now changed the colors as suggested, Fig.4, Extended Data Fig. 7 and Supplementary Information.

Reviewer #3 (Remarks to the Author):

The authors have made a significant effort to assess the functional roles of the W8A mutant in ESC biology, uncovering an intriguing differentiation phenotype. In addition, they determined the consequences of the mutants on genome-wide DNA binding (ChIP-Seq) and 3D chromatin architecture (3C), showing differences in binding preferences and looping strength. I am satisfied with the authors' responses and revisions and recommend publication.

We thank you.

Final Decision Letter:

Subject: Decision on Nature Cell Biology submission NCB-F46094B

Message:

Dear Dr Ferreon,

I am pleased to inform you that your manuscript, "NANOG prion-like assembly mediates DNA bridging to facilitate chromatin reorganization and activation of pluripotency", has now been accepted for publication in Nature Cell Biology.

Please note that Nature Cell Biology is a Transformative Journal (TJ). Authors may publish their research with us through the traditional subscription access route or make their paper immediately open access through payment of an article-processing charge (APC). Authors will not be required to make a final decision about access to their article until it has been accepted. Find out more about Transformative Journals

Authors may need to take specific actions to achieve compliance with funder and institutional open access mandates. If your research is supported by a funder that requires immediate open access (e.g. according to Plan S principles) then you should select the gold OA route, and we will direct you to the compliant route where possible. For authors selecting the subscription publication route, the journal's standard licensing terms will need to be accepted, including self-archiving policies. Those licensing terms will supersede any other terms that the author or any third party may assert apply to any version of the manuscript.

If your paper includes color figures, please be aware that in order to help cover some of the additional cost of four-color reproduction, Nature Research charges our authors a fee for the printing of their color figures. Please contact our offices for exact pricing and details.

If you have not already done so, we strongly recommend that you upload the step-by-step protocols used in this manuscript to the Protocol Exchange (www.nature.com/protocolexchange), an open online resource established by Nature Protocols that allows researchers to share their detailed experimental know-how. All uploaded protocols are made freely available, assigned DOIs for ease of citation and are fully searchable through nature.com. Protocols and the Nature and Nature research journal papers in which they are used can be linked to one another, and this link is clearly and prominently visible in the online versions of both papers. Authors who performed the specific experiments can act as primary authors for the Protocol as they will be best placed to share the methodology details, but the Corresponding Author of the present research paper should be included as one of the authors. By uploading your Protocols to Protocol Exchange, you are enabling researchers to more readily reproduce or adapt the methodology you use, as well as increasing the visibility of your protocols and papers. You can also establish a dedicated page to collect your lab Protocols. Further information can be found at www.nature.com/protocolexchange/about

You can use a single sign-on for all your accounts, view the status of all your manuscript submissions and reviews, access usage statistics for your published articles and download a record of your refereeing activity for the Nature journals.

With kind regards,

Jie Wang, PhD
Senior Editor
Nature Cell Biology

Tel: +44 (0) 207 843 4924
email: jie.wang@nature.com